# UV-induced G4 DNA structures recruit ZRF1 which prevents UV-induced senescence

Alessio De Magis [1,2], Michaela Limmer[1,2], Venkat Mudiyam [1], David Monchaud [3], Stefan Juranek [2] & Katrin Paeschke [1,2] ✉

Senescence has two roles in oncology: it is known as a potent tumor-suppressive mechanism, which also supports tissue regeneration and repair, it is also known to contribute to reduced patient resilience, which might lead to cancer recurrence and resistance after therapy. Senescence can be activated in a DNA damage-dependent and -independent manner. It is not clear which type of genomic lesions induces senescence, but it is known that UV irradiation can activate cellular senescence in photoaged skin. Proteins that support the repair of DNA damage are linked to senescence but how they contribute to senescence after UV irradiation is still unknown. Here, we unraveled a mechanism showing that upon UV irradiation multiple G-quadruplex (G4) DNA structures accumulate in cell nuclei, which leads to the recruitment of ZRF1 to these G4 sites. ZRF1 binding to G4s ensures genome stability. The absence of ZRF1 triggers an accumulation of G4 structures, improper UV lesion repair, and entry into senescence. On the molecular level loss of ZRF1 as well as high G4 levels lead to the upregulation of DDB2, a protein associated with the UV-damage repair pathway, which drives cells into senescence.

Genome stability is constantly challenged by exogenous and endogenous factors such as genotoxic agents (e.g., cigarette smoke), environmental factors (e.g., ultraviolet (UV) light, irradiation), metabolic activity or the formation of alternative DNA structures (e.g., G-quadruplex-DNA or G4)[1,2]. To counteract these challenges different DNA damage repair (DDR) pathways are activated to recognize and repair DNA lesions and maintain genome stability. When DNA damage is particularly severe or the repair machinery is not working properly[3,4], cells enter apoptosis[5] or senescence[6,7].

The irreversible growth arrest senescence is seen, at least in part, as a mechanism that can suppress the development of cancer[8]. Different cellular changes such as telomere defects, DNA double-strand breaks (DSBs), UV irradiation as well as the stable formation of G4 structures were shown to induce senescence[9–12].

G4s are stable DNA structures that form within specific guanine (G)-rich regions in all tested genomes. In humans, over 1 million potential sites exist that can fold into G4s[13–15]. Based on their position in the genome (notably being enriched at promoters, transcription start sites and telomeres) as well as their evolutionary conservation, the current model is that folded G4s support and fine-tune biological processes such as expression of specific genes, telomere maintenance and transcription factor binding to specific sites[16]. G4 formation changes dynamically in the cell[17,18]. Together with this, current data suggests that specific cell types, cell stages and exogenous stimuli influence the formation of specific G4s that can affect cellular pathways/functions[14,19]. G4s support cellular functions, therefore their formation and unfolding must occur in a controlled and regulated manner[20–25].

In contrast to these positive aspects are the findings that G4s can induce DNA damage[26,27] and directly affect the efficiency of DNA repair[28]. G4s can block DNA replication, stimulate DNA recombination events and affect transcription and telomere maintenance in a negative manner[17,18]. G4 formation can also drive apoptosis and is linked to increased senescence[10,11,29,30] by either triggering unsustainable genetic

[1]Institute of Clinical Chemistry and Clinical Pharmacology, University Hospital Bonn, Bonn, Germany. [2]Department of Oncology, Hematology and Rheumatology, University Hospital Bonn, Bonn, Germany. [3]Institut de Chimie Moléculaire de l'Université de Bourgogne (ICMUB), CNRS UMR 6302, Université de Bourgogne, Dijon, France. ✉e-mail: katrin.paeschke@ukbonn.de

instability or altering gene expression due to their formation within specific promoters, or both, but the fine mechanistic details are not known yet.

We have recently identified a unique function of G4s for genome stability, revealing that UV irradiation induces the formation of G4s in *S. cerevisiae*[21]. These folded G4s are bound by the protein Zuo1 that stabilizes G4s and leads to the recruitment of nucleotide excision repair (NER) proteins to the UV damage site, which is essential for an efficient repair. Further molecular experiments revealed that the G4 formation itself is key for NER recruitment to the lesion and for an efficient repair[21].

NER is a highly conserved pathway and many proteins that act in yeast are also relevant for human NER activity[31]. The major proteins that contribute to NER belong to the XP family. Mutations in most of these proteins are linked to genetic disorders such as Xeroderma pigmentosum (XP) and a high predisposition to skin cancer[32]. In the NER pathway one of the most important proteins involved in the recognition of DNA lesions is XPC[33,34] that specifically recognizes bulky DNA adducts. XPC ubiquitination is supported by ZRF1 and the CLR4-DDB2 complex, a heterodimer of DDB1-DDB2 and the E3 ligase components, RBX1 and CUL4A[34].

ZRF1 (also known as DnaJC2) is the human ortholog of Zuo1 in yeast and is a member of the M-phase phosphoprotein family that contains a DNA binding domain[35]. ZRF1 was originally characterized as a ribosome-associated factor but accumulating data now points towards relevant functions for gene activation, senescence and cancer progression[35–37]. Similar to its yeast ortholog, ZRF1 function is linked to NER by supporting XPC ubiquitination[36]. However, its function and relevance for NER and genome stability is not studied in detail yet. The findings that changes in ZRF1 activity are linked to different types of cancer such as acute myeloid leukemia (AML)[38], gastric carcinoma[39], breast carcinoma[40] and head and neck squamous cell carcinoma (HNSCC)[41] highlights the connection of ZRF1 to genome stability.

Here, we investigated this connection, in particular after UV irradiation. Similar to yeast, G4s form rapidly in human cells after UV irradiation. We show that ZRF1 specifically targets DNA G4s that form upon UV irradiation and provide evidence that supports its role in safeguarding genome stability as its absence leads to the first signs of genome instability (micronuclei formation). Our data reveals a cellular mechanism involving G4s, showing that their formation controls ZRF1 binding to chromatin, supports genome stability and prevents senescence.

## Results

### G4s form after UV irradiation and affect cellular functions

In higher eukaryotes, folded G4s can, under specific conditions, serve as hotspots for DNA damage and negatively impact genome stability[26,27] by either acting as roadblocks to polymerase motion[2,42] or by trapping DNA-related enzymes (e.g., TOP2A) on their substrates in a structure-dependent manner[43,44]. In yeast it has been shown that folded G4s positively support DNA repair after UV irradiation[21].

Here we tested the impact of G4 stabilization on cell viability after UV irradiation in human cells. HeLa cells were treated with pyridostatin (PDS, 2 µM, 48 h), a well-studied G4 ligand that increases G4 formation by at least two-fold[26]. PDS treated as well as untreated cells were subsequently irradiated with different UV doses (0–50 J m$^{-2}$) and the vitality was checked 24 h after UV irradiation (recovery) (Fig. 1a). First, we used the MTT assay (3-(4,5-dimethylthiazolyl-2)−2,5-diphenyl-tetrazolium bromide), a colorimetric assay that measures cell metabolic rate, as a readout for cellular vitality[45]. This assay demonstrated that enhanced G4 formation resulted in a gain in vitality after UV irradiation (Fig. 1a). At 10 J m$^{-2}$ cells had twice the metabolic activity (60% metabolically active) upon PDS treatment compared to untreated cells (30% metabolically active). Non-irradiated HeLa cells were used for normalization and the average of the replicates represents 100% vitality.

To assess whether these metabolic changes correlate also with enhanced growth after UV irradiation (10 J m$^{-2}$), colony formation assays were performed with untreated and PDS-treated cells. As a control, we used HeLa cells that lacked a functional NER machinery after transfecting them with short interference RNA (siRNA) against XPC (western blot with XPC protein levels after siRNAs transfection is reported in Fig. 1a of the data source folder). Contrary to MTT assays, colony formation assays showed a complete lack of cell growth after treatment with PDS (2 µM, 48 h) and UV irradiation. As expected, XPC downregulation led to a loss in cell growth in colony formation assays after UV irradiation (Supplementary Fig. 1a). These results revealed that G4 stabilization leads to an increased metabolic activity but also to the loss of colony formation after UV irradiation (Supplementary Fig. 1a).

To assess if enhanced G4 formation directly caused the observed increased metabolic rate after UV irradiation, we used a newly identified compound, PhpC, which destabilizes G4s in vitro[46]. In order to visualize G4s in human cells, immunofluorescence (IF) analyses were performed using the well-characterized G4-specific antibody BG4[47]. IF analyses demonstrated that 100 µM PhpC reduced G4 levels significantly (Supplementary Fig. 1b) while it had no effect on the metabolic activity (Fig. 1b). These results confirmed that PhpC can modulate G4 formation in cells. To assess whether high G4 levels caused the high MTT rates after UV irradiation, we reduced G4s by PhpC and determined the metabolic activity after UV irradiation. 100 µM PhpC reduced the metabolic activity compared to untreated control cells (25% reduction in metabolic activity cells when the PhpC treatment was performed before or after UV irradiation) (Fig. 1b). These results strengthened our hypothesis that G4 formation positively correlates with an increase in metabolic activity after UV irradiation.

To further investigate whether more G4 structures are present in cells in response to UV irradiation, we monitored G4 levels by IF over time in cells after UV irradiation (0 to 8 h after UV irradiation) (Fig. 1c). One hour after UV irradiation 2-fold higher G4 levels were detected, which then slowly decayed over time (Fig. 1c). A similar trend was obtained in PDS-treated cells (2 µM, 48 h) after UV irradiation (Supplementary Fig. 1c), showing that multiple newly G4s form upon UV irradiation.

We then decided to further investigate the observed decrease in colony formation when G4s are stabilized after UV irradiation. UV irradiation causes a variety of DNA damage products among which are 6–4 photoproducts (6–4PPs) and cyclobutane pyrimidine dimers (CPDs). Those products are responsible for many genetic mutations that can drive photocarcinogenesis and lead to the development of skin cancer[32,33]. In order to address how G4 formation influences the repair of CPDs and 6–4PPS after UV irradiation, we first determined the amount of 6–4PPs as a readout for UV-promoted DNA damage. PDS treated (2 µM, 48 h) and untreated cells were UV irradiated with 10 J m$^{-2}$ and 6-4PPs formation was monitored at 0, 1 and 4 h after UV irradiation by IF using an anti-6-4PPs antibody (untreated wild type (WT) staining is reported in Fig. 1b of the data source folder). HeLa cells transfected with siRNA against XPC were used as a control (Fig. 1d). 6-4PPs was significantly elevated after G4 stabilization with PDS (p-value < 0.0001, Fig. 1d). Using a similar experimental set up we also monitored CPD formation 0 and 24 h after 10 J m$^{-2}$ UV irradiation in PDS treated (2 µM, 48 h) and untreated cells. CPD formation was monitored both by IF (Supplementary Fig. 1d) and by dot blot analysis (Supplementary Fig. 1e). HeLa cells transfected with siRNA against XPC were used as a control (Supplementary Fig. 1d,e). Similar to 6-4PPs IF analysis showed a significant change in CPDs after G4 stabilization by PDS. A 1.4-fold increase in CPDs directly after UV irradiation and a 1.6-fold increase after 24 h (Supplementary Fig. 1d). Using dot blot we determined that G4 stabilization induced a slight but not significant increase in CPD directly after UV irradiation (1.6-fold), which is increasing after 24 h (3.2-fold, Supplementary Fig. 1e). SYBR Gold

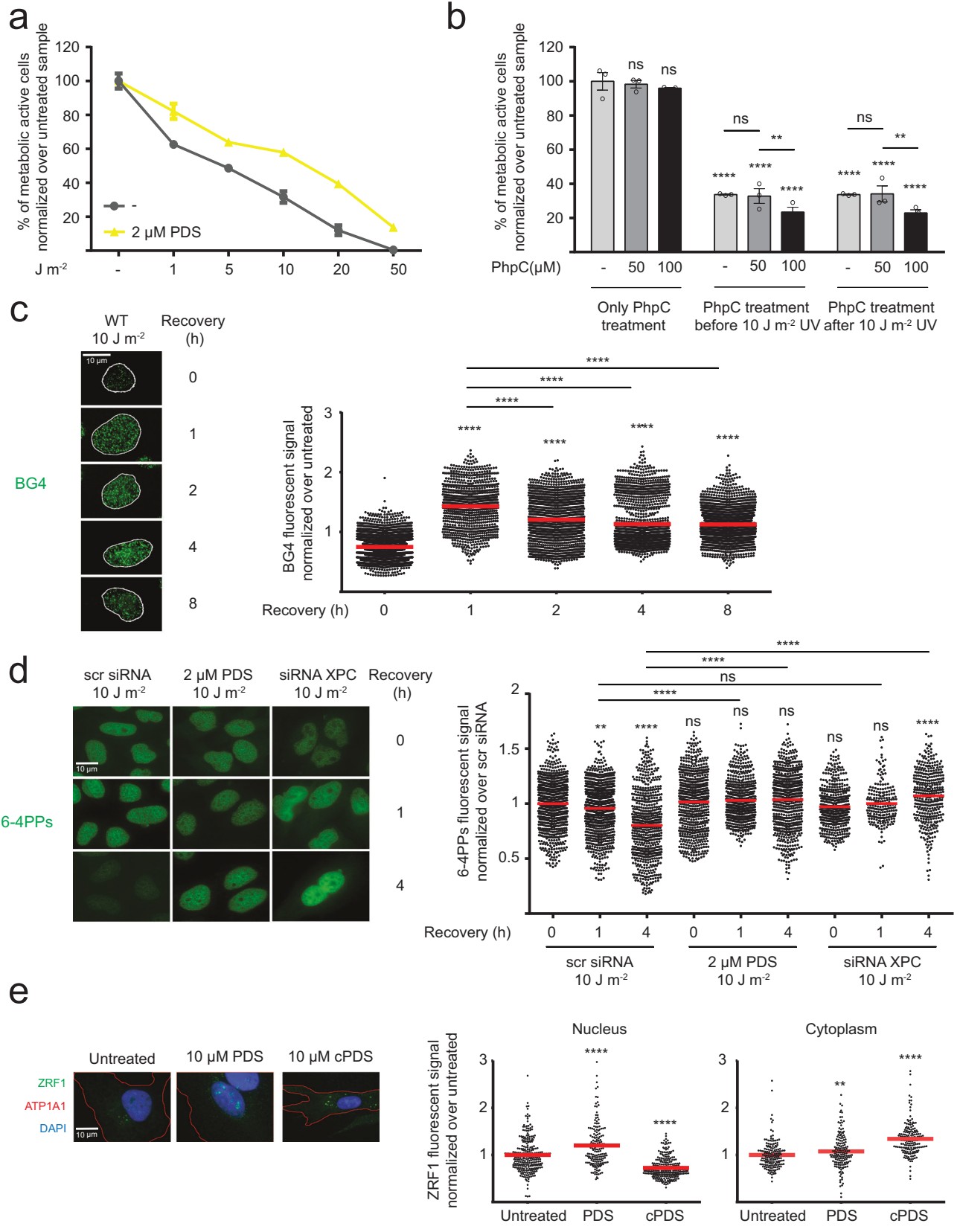

staining was used for normalization in dot blots as it labels total DNA (Supplementary Fig. 1e). These results indicated that cells treated with the G4-stabilizer PDS cannot response properly to UV-induced DNA damage (Fig. 1d, Supplementary Fig. 1d, e).

## ZRF1 binds to folded G4s and supports genome stability

We showed that G4s form after UV irradiation and that their formation reduced cellular growth but increased metabolic activity (Fig. 1). In cells, G4 folding and unfolding is regulated by specific proteins. In

**Fig. 1 | Consequences of G4 structures for the cellular fitness. a** Vitality assay (MTT) in HeLa cells treated/untreated for 48 h with 2 µM PDS and irradiated with different amounts of UV light and recovered 24 h in DMEM/10% FBS. The graph shows the mean of $n = 3$ biological independent experiments ± SEM. **b** Vitality assay (MTT) in HeLa cells that were untreated or irradiated with 10 J m$^{-2}$ UV light and treated for 24 h with different concentrations of PhpC and recovered 24 h in DMEM/10% FBS. The graph shows the mean of $n = 3$ biological independent experiments ± SEM. Significance was determined using a two-sided T test. Asterisks indicate statistical significance; in detail, $*p < 0.05$, $**p < 0.01$, $***p < 0.001$, $****p < 0.0001$. **c** Left, IF staining of HeLa cells, irradiated with 10 J m$^{-2}$ UV light, recovered in DMEM/10% FBS for 1 to 8 h and stained with BG4 antibody (green) and DAPI (signal was used to indicate the nuclear border as a white line). Scale bar: 10 µm. Right, quantification of the BG4 signal in the nucleus of the cells. **d** Left, IF staining of HeLa cells, pretreated 24 h with a scrambled siRNA, 2 µM PDS or a siRNA against XPC, irradiated with 10 J m$^{-2}$ UV light and recovered in DMEM/10% FBS for 0–4 h. The cells were stained with an anti-6-4PPs antibody (green). Scale bar: 10 µm. Right, Quantification of the 6–4PPs signal. **e** Left, IF staining of HeLa cells, treated/untreated 24 h with 10 µM of PDS or cPDS, stained with anti-ZRF1 antibody (green), anti-sodium potassium ATPase (ATP1A1 - red) and DAPI (blue). Scale bar: 10 µm. Right, quantification of ZRF1 signal in nucleus (left panel) and cytoplasm (right panel). The graphs in **c**–**e** show fluorescence intensity (FI) levels normalized over untreated cells of $n = 3$ biological independent experiments. Horizontal red line represents the mean value. Significance was determined using an ordinary one-side ANOVA multiple comparison using the Geisser-Greenhouse correction. Asterisks indicate statistical significance; in detail, $*p < 0.05$, $**p < 0.01$, $***p < 0.001$, $****p < 0.0001$. Significance compared to untreated-WT cells is indicated by asterisks, connecting lines are used when the significance was compared to other samples.

yeast, we identified that Zuo1 binds to G4s after UV irradiation and promotes the recruitment of NER proteins. We revealed that G4 formation is essential for NER activation at specific lesions sites in yeast[21]. In vitro data revealed that ZRF1, the human ortholog of Zuo1, contributes to NER by supporting the formation of the CLR4-DDB2 complex[34]. We thus postulated that, similarly to Zuo1, ZRF1 binds to G4s. To investigate this, we first checked whether ZRF1 localizes to cellular compartments that showed enhanced G4 levels. PDS treatment increases G4 levels in both, nucleus and cytoplasm[48], while carboxyPDS (cPDS) increases G4 levels mainly in the cytoplasm[23]. We demonstrated by IF that after G4 stabilization by both, PDS and cPDS, ZRF1 localized to the cell compartment where G4s have accumulated (Fig. 1e). This finding supported our hypothesis that ZRF1 is recruited to G4 structures.

We then monitored ZRF1 binding to DNA sites using chromatin immunoprecipitation followed by genome-wide sequencing analysis (ChIP-seq). No chromosomal binding sites (peaks) were identified for ZRF1 in untreated cells, suggesting that ZRF1 does not bind to chromatin in unchallenged conditions. In sharp contrast, 2195 specific ZRF1 peaks were detected upon incubation with PDS (2 µM, 48 h) (Fig. 2a, Supplementary Data 1). When analysing all ChIP-seq peaks, we found characteristic G-rich binding motifs of ZRF1 by MEME analysis, corresponding to G4 consensus motifs prone to fold into G4s (Fig. 2b). A correlation of obtained ZRF1 binding sites with published G4 sites[14] revealed a significant overlap (24%; 271 peaks on the reverse strand and 259 peaks on the forward strand (significance: p < 0.01)) (Supplementary Fig. 2a) demonstrating an interaction of ZRF1 with G4 sites.

To assess whether ZRF1 folds/unfolds G4s, we created a stable ZRF1-knockout (KO) in HeLa cells by CRISPR/Cas9. ZRF1-KO clones were confirmed by western blot (Supplementary Fig. 2b). First, we measured changes in the doubling time between wildtype (WT) and ZRF1-KO cells and found that ZRF1-KO cells only showed minor changes in their growth rate (doubling time of 18 h for WT and 19 h for ZRF1-KO cells) (Supplementary Fig. 2c). Second, we determined if in the absence of ZRF1 cells were sensitive to PDS treatment. ZRF1-KO cells did not grow after PDS (10 µM) treatment (Supplementary Fig. 2c) implying that ZRF1 is essential to support genome stability when cells exhibit enhanced G4 levels. Third, we monitored G4 levels in WT and ZRF1-KO cells with non-toxic concentrations of PDS (range 1–5 µM) for 24, 48 and 72 h by IF using BG4 (Fig. 2c, Supplementary Fig. 2d, e). Without PDS, G4 levels were significantly elevated in ZRF1-KO in comparison to WT cells and the addition of PDS triggered only minor changes of G4 levels in both, ZRF1-KO and WT cells.

G4 formation is connected to genome instability and their stabilization by PDS was shown to induce DNA damage and stimulate micronuclei formation[26,27,49], which indicates that stabilized G4s impair proper DNA damage repair[50,51]. We thus tracked micronuclei formation upon PDS incubation (range 1–5 µM) for 24, 48, and 72 h using DAPI fluorescence as a readout. A slight increase of micronuclei was observed in ZRF1-KO compared to WT cells (8% versus 2% of the cells,

Fig. 2d). Increasing PDS concentration led to a significant increase in micronuclei formation in ZRF1-KO compared to WT cells (Fig. 2d, Supplementary Fig. 3a, b) indicating that in ZRF1-KO cells genome instability is increased. To test if micronuclei originated from DNA single-strand breaks (SSBs) or DSBs, we monitored S139-phosphorylation of histone H2AX (γH2AX) foci, a known SSB/DSB marker[52,53]. No changes in γH2AX were detected between WT and ZRF1-KO cells with or without PDS treatment (Fig. 2e), indicating that micronuclei formation was not due to more SSBs or DSBs in ZRF1-KO cells.

## ZRF1 binds to G4 after UV irradiation

Zuo1, the yeast ortholog of ZRF1, binds to G4s after UV irradiation where it supports the binding and recruitment of NER proteins[21]. To elucidate if ZRF1 function is similar to Zuo1, we tested whether ZRF1 can rescue cellular defects in the absence of Zuo1. We used yeast plasmids (kindly provided by the Craig lab) containing either isoform 1 or isoform 2 of ZRF1[54]. Both isoforms are homolog to the N-terminus of Zuo1 and differ by a 445 bp region located at the N-terminus. Isoform 2 lacks two alternate in-frame exons compared to isoform 1, resulting in a shorter protein. We expressed ZRF1 in either wild type (WT) or Zuo1-deficient *(zuo1Δ)* yeast strains. Standard plating assays on rich media plate (yeast extract, peptone and dextrose (YPD)) before and after UV irradiation (15 J m$^{-2}$) were performed. In WT yeast cells, expression of ZRF1 did not affect growth or UV sensitivity but both isoforms of ZRF1 rescued the growth defect of *zuo1Δ* yeast cells (Fig. 3a). Of note, after UV irradiation only isoform 2 of ZRF1 led to a partial rescue of the UV sensitivity of *zuo1Δ* cells (Fig. 3b). These results showed that ZRF1 partly rescued *zuo1Δ* defects in yeast indicating a partly conserved function.

Our results showed that ZRF1 enters the nucleus, binds to G4s and mitigate G4-mediated genome instability (Figs. 2 and 3). As we also showed that UV irradiation induces G4 formation (Fig. 1), we speculated that ZRF1 binds to G4s that accumulate after UV irradiation. To test this assumption, we monitored ZRF1 binding after UV irradiation by ChIP-seq in human cell lines and found that, similar to the PDS treatment, UV irradiation recruits ZRF1 to DNA and 4159 ZRF1 binding sites were detected (Fig. 3c, Supplementary Data 2). MEME motif analysis[55] revealed that most ZRF1 bindings sites harbor a G-rich binding motif (Fig. 3d, top panel), which significantly overlaps published G4 motifs[13,14] (p-value 0.001). Interestingly, ZRF1 binding sites after UV irradiation significantly overlap with peaks detected after PDS treatment (Fig. 2a, Supplementary Fig. 4a, b) (p < 0.01), further supporting the hypothesis that ZRF1 is recruited to UV-induced G4s. Co-treatment of UV and PDS (2 µM, 48 h) accelerated ZRF1 recruitment to chromatin and resulted in 34934 ZRF1 binding sites (Fig. 3c, d bottom panel, Supplementary Data 3).

G4 stabilization by PDS prevented the repair of both 6-4PPs and CPDs after UV irradiation (Fig. 1d, Supplementary Fig. 1d, e). As ZRF1-KO cells had more G4s, we tested whether ZRF1-KO cells

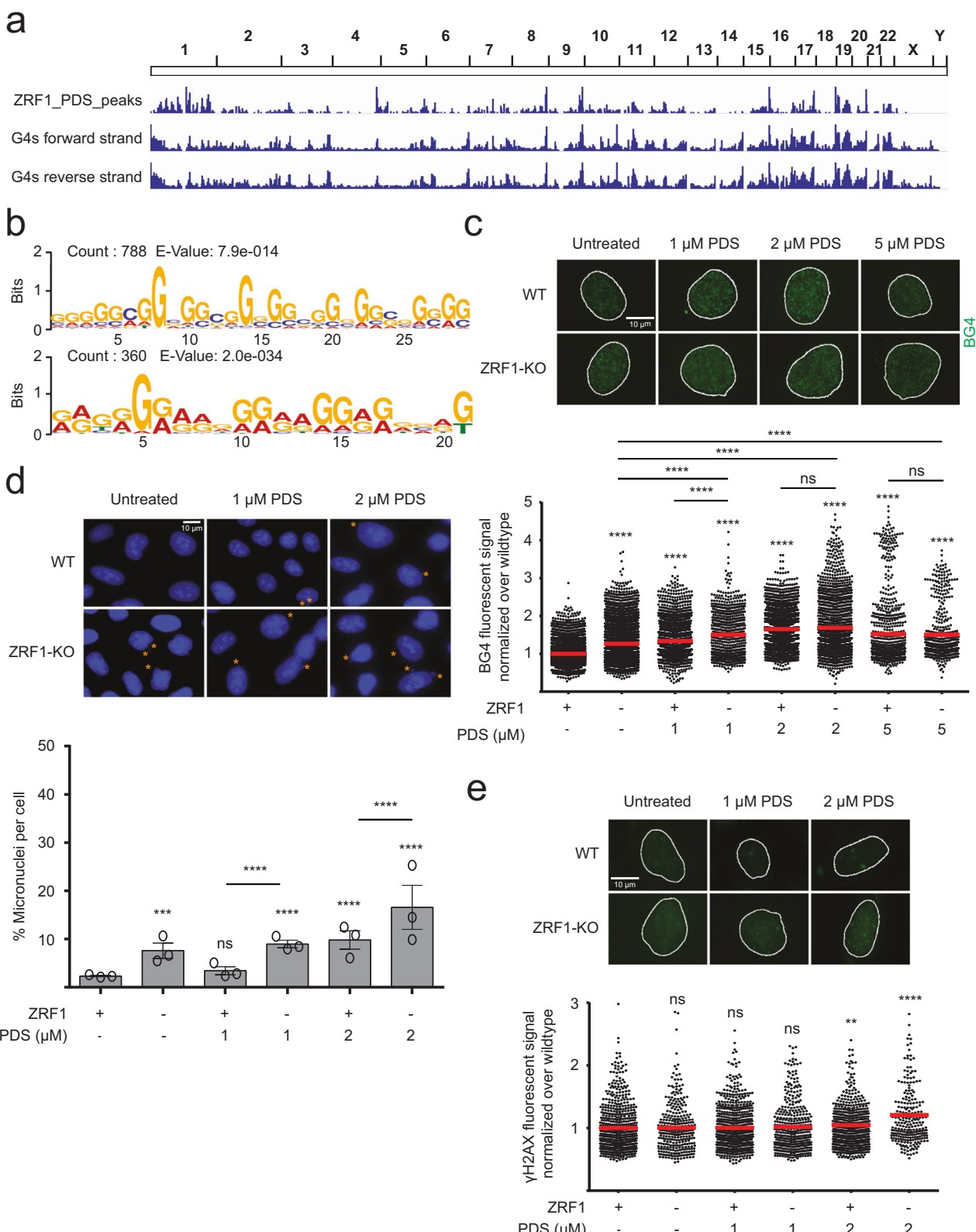

exhibited a similar defect in UV damage repair. We monitored both 6-4PPs and CPD levels by IF and in addition CPD products by dot blot. In these assays, cells transfected with a siRNA against XPC were used as a control (Fig. 3e). This analysis revealed that 6-4PPs were significantly elevated in ZRF1-KO cells (p-value < 0.001, Fig. 3e). These data were supported by the quantification of CPDs. IF analysis showed a significant change in CPDs in ZRF1-KO cells: a 1.6-fold

increase in CPDs directly at 0 and 24 h after UV irradiation (Supplementary Fig. 4c). A similar trend in CPDs was determined in ZRF1-KO cells using dot blot experiments. CPDs were enriched directly after UV irradiation (1.2-fold) and even increased after 24 h (1.9-fold, Supplementary Fig. 4d). However, dot blot analyses were not significant. SYBR Gold staining was used for normalization in the dot blots (Supplementary Fig. 4d).

**Fig. 2 | ZRF1 binds G4s and is involved in attenuating G4 mediated genome instability. a** IGV Genome Browser screenshot. Upper lane: Genome-wide ZRF1 binding sites in HeLa cells treated 48 h with 2 μM of PDS; bottom lanes: G4 regions obtained for forward and reverse strand by Polymerase Stop Assay[14]. **b** Motif discovery using MEME for ZRF1 binding sites in HeLa cells treated 48 h with 2 μM PDS. The motifs reveal the presence of possible G4 motifs (representation and e-value are reported in the figure). **c** IF staining of WT and ZRF1-KO cells, treated/untreated 48 h with 1, 2, or 5 μM of PDS, stained with the BG4 antibody (green), and DAPI (signal was used to indicate the nuclear border as a white line). Scale bar: 10 μm. Below, quantification of the BG4 signal in the nucleus of the cells. **d** IF staining of WT and ZRF1-KO cells, treated/untreated 48 h with 1 or 2 μM of PDS and stained with DAPI (blue). Orange asterisks identified micronuclei location. Scale bar: 10 μm. Below, micronuclei quantification as a percentage of micronuclei per cells. Bars show mean value of *n* = 3 biological independent experiments ± SEM. **e** IF staining of WT and ZRF1-KO cells, treated/untreated 48 h with 1 or 2 μM of PDS, stained with anti-γH2AX antibody (green), and DAPI (signal was used to indicate the nuclear border as a white line). Scale bar: 10 μm. Below, quantification of γH2AX signal in the nucleus of the cells. Graphs in **c**, **e** show fluorescence intensity (FI) levels normalized over untreated cells of, at least *n* = 3 biological independent experiments. Horizontal red line represents the mean value. Significance was determined using an ordinary one-sided ANOVA multiple comparison using the Geisser-Greenhouse correction. Asterisks indicate statistical significance; in detail, *$p < 0.05$, **$p < 0.01$, ***$p < 0.001$, ****$p < 0.0001$. Significance compared to untreated-WT cells is indicated by asterisks, connecting lines are used when the significance was compared to other samples.

G4 stabilization positively influenced metabolic activity after UV damage (Fig. 1a). ZRF1-KO cells showed elevated G4 levels (Fig. 2c). We thus tested whether the ZRF1-KO affects the metabolic rate after different UV irradiation doses. MTT assays were performed in WT and ZRF1-KO cells along with a ZRF1-KO cell line containing a ZRF1 expression plasmid (rescue condition, ZRF1-RE). Western analysis showed that ZRF1 was expressed in the rescue conditions (Supplementary Fig. 4e). Cell numbers were counted before and after UV irradiation to avoid cell number-specific changes. ZRF1-KO cells showed enhanced metabolic rates after UV irradiation compared to WT cells. After UV irradiation (10 J m$^{-2}$) ZRF1-KO cells were 70% metabolically active (Fig. 3f) while, as expected, the ZRF1-RE showed similar rates as WT cells (49% versus 37% for the WT) (Fig. 3f). We also checked cell viability with colony formation assays (WT cells treated with a siRNA against XPC were used as a control). The ZRF1-KO cells showed a complete lack of growth similar as the NER-deficient cell line (Supplementary Fig. 4f) whereas the ZRF1-RE cells showed an almost complete rescue. Taken together our results showed that I ZRF1 binds to G4s after UV irradiation where it prevents genome instability and II cells lacking ZRF1 showed increased metabolic activity after UV irradiation.

## DDB2 upregulation drives ZRF1-KO phenotype

Our data demonstrated that although ZRF1-KO cells stop dividing, their metabolic rate increases after UV irradiation (Fig. 3f and Supplementary Fig. 4f). To characterize which gene expression changes occurred in ZRF1-KO cells, we performed RNA-seq with both WT and ZRF1-KO cells. 1004 differently expressed genes (DEGs) were detected (Supplementary data 4). Gene ontology of DEGs confirmed that genes linked to metabolic processes were expressed differentially. Further genes belonging to the GO annotations negative regulators of apoptosis (*p*-value 1.08E-04), stress response (*p*-value 7.08E-06) and stimuli response (*p*-value 1.04E-05) were upregulated in ZRF1-KO cells. To gain more insights into specific UV irradiation responses we analyzed which of the NER-associated genes were differentially expressed and identified that DDB2, previously shown to colocalize with ZRF1[34], was upregulated in ZRF1-KO cells. Changes in DDB2 gene expression were confirmed by quantitative PCR (qPCR) in both ZRF1-KO and WT cells (Supplementary Fig. 5a). To exclude that the total NER machinery is differentially expressed, we monitored gene expression changes of two additional NER-associated genes, ERCC2 and ERCC3 (Supplementary Fig. 5a). These analyses showed that ZRF1 did not influence the gene expression of other tested NER-associated genes. Western blot analysis showed that in ZRF1-KO cells DDB2 protein expression was also elevated before and after UV irradiation (not significant, Supplementary Fig. 5b, c). To clarify which impact DDB2 has on the ZRF1-KO phenotype, in particular upon UV irradiation, we further analyzed our RNA-seq data. It is known that DDB2 acts as a heterodimer with DDB1, which binds strongly to DNA containing 6−4PPs and CPD (while weakly binding to undamaged DNA)[56]. The CLR4-DDB2 complex is required for efficient NER[57,58]. DDB2 acts upstream of XPC[32,33], which interacts with ZRF1[34]. We detected that XPC gene expression was also upregulated in ZRF1-KO cells by RNA-seq (*p*-value > 0.05) and qPCR analysis (Supplementary Fig. 5d). Previously it was shown that DDB2 levels directly impact XPC[59]. This led us to postulate that XPC is upregulated as a consequence of elevated DDB2 expression. To confirm this, we downregulated DDB2 by siRNA and found that XPC mRNA levels decreased significantly (Supplementary Fig. 5d). To further assess the contribution of DDB2 and XPC on metabolic activity of ZRF1-KO cells, MTT assays were performed. DDB2 and XPC were downregulated by siRNA followed by a 24 h recovery period in siRNA-free media (downregulation was confirmed by immunoblotting, Supplementary Fig. 5e). Then, cells were UV-treated and MTT analyses were performed. The downregulation of either DDB2 or XPC decreased the metabolic activity of ZRF1-KO after UV-irradiation (Fig. 4a), confirming that the increased metabolic rate of ZRF1-KO cells, after UV irradiation, was due to the up-regulation of DDB2.

Next, we investigated if G4 formation was influenced by DDB2 levels. We tested G4 formation after DDB2 downregulation in both WT and ZRF1-KO cells, before and after UV irradiation (0−8 h after treatment) (Fig. 4b, c). In WT cells, the downregulation of DDB2 did not affect G4 levels significantly compared to untreated or a scrambled siRNA control. Contrary, in ZRF1-KO cells the downregulation of DDB2 prevented the ZRF1-KO specific increase of G4s in response to UV irradiation (Fig. 4b, c). This data suggested that in the absence of ZRF1, DDB2 positively contributes to G4 accumulation after UV irradiation. To support this hypothesis, we tested if DDB2 binds to G4 structures in cells. Specifically, we used the G4-specific antibody BG4 to pull down DNA regions with G4 structures and their associated protein from cells. Briefly, chromatin was sheared and incubated with the BG4 antibody and after stringent washes the presence of DDB2 in the pull-down fraction was monitored by western blot analysis. Western blot analysis showed a specific immunoprecipitation of DDB2 before and after UV irradiation to G4s (Fig. 4d, Ponceau staining in Supplementary Fig. 5f). This data was complemented by binding assays that also demonstrated the binding of the heterodimer DDB1-DDB2 to a G4 structure in vitro (Supplementary Fig. 5g). These data indicated that DDB2 may bind to G4s. Without both DDB2 and ZRF1, G4s did not accumulate after UV irradiation (Fig. 4b,c) and cells did not display enhanced metabolic activity after UV irradiation (Fig. 4a).

## ZRF1 prevents upregulation of senescence markers and DDB2-driven senescence

Elevated G4 levels caused by either the absence of ZRF1 or PDS treatment, led to an upregulation of cellular metabolism after UV irradiation, likely involving the upregulation of DDB2. As this upregulation did not lead to an efficient NER (as determined by the quantification of 6−4PPs and CPDs, Figs. 1d and 3e, Supplementary Figs. 1e and 4d), we performed an RNA-seq analysis after UV irradiation to check for DEGs that correlate to changes in cellular fitness/metabolism. We identified 1132 DEGs between WT and ZRF1-KO cells after UV irradiation

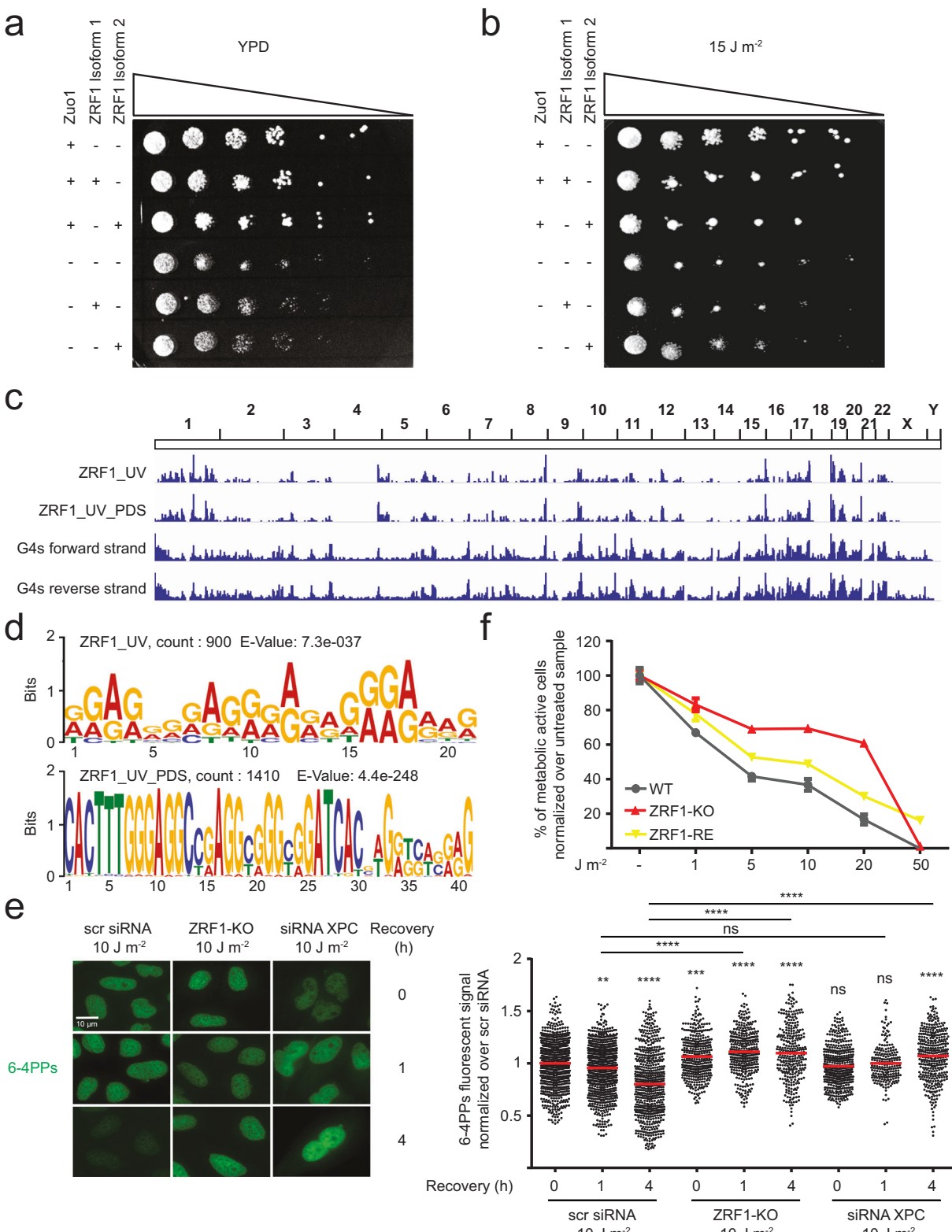

(Supplementary Data 5). When subtracting those DEGs from those already identified before UV irradiation, 548 DEGs could be specifically identified under UV conditions (Supplementary Data 6). Most DEGs corresponded to genes that were upregulated in UV conditions (400/548), among which were genes involved in cellular senescence (10 DEGs, Supplementary Table 1). The upregulation of the senescence genes CDKN2A (p16$^{INK4a}$ and p14$^{ARF}$), CDKN1A and senescence-

associated secretory phenotype (SASP) genes in ZRF1-KO cells after UV irradiation was confirmed by qPCR analysis (Fig. 5a).

These findings indicated that senescence might be the trigger which affects cellular metabolism after UV irradiation in ZRF1-KO cells. This assumption agrees with a previous publication showing that ZRF1 supports oncogene-induced senescence by modulating the expression of the tumor suppressor INK4-ARF[37] by binding to the promoters of all

**Fig. 3 | ZRF1 activity is evolutionary conserved but with an opposite phenotype compared to the yeast orthologue ZUO1. a, b** A 1:5 serial dilution of yeast cells was spotted on YPD media and irradiated with 15 J m$^{-2}$ UV light. Growth changes of $n = 3$ biologically independent experiments were monitored by colonies formation. **c** IGV Genome Browser screenshot. Upper lanes: Genome-wide ZRF1 binding sites in HeLa cells pre-treated/untreated 48 h with 2 µM of PDS and treated/untreated with 10 J m$^{-2}$ UV light; bottom lanes: G4 regions obtained from forward and reverse strand by Polymerase Stop Assay[14]. **d** Motif discovery using MEME for ZRF1 binding sites in HeLa cells irradiated with 10 J m$^{-2}$ UV light (top panel) and pre-treated 48 h with 2 µM of PDS and irradiated with 10 J m$^{-2}$ UV light (bottom panel). The motifs reveal the presence of possible G4 motifs (representation and e-value are reported in the figure). **e** Left, IF staining of WT and ZRF1-KO treated with 10 J m$^{-2}$ UV light and recovered in DMEM/10% FBS for 0–4 h. The cells were stained with an anti-6-

4pps antibody (green), and DAPI (blue). Scale bar: 10 µm. Right, quantification of 6–4pps signal in the nucleus of the cells. Graph show fluorescence intensity (FI) levels normalized over WT treated cells of $n = 3$ biological independent experiments ± SEM. Horizontal red line represents the mean value. Significance was determined using an ordinary one-sided ANOVA multiple comparison using the Geisser-Greenhouse correction. Asterisks indicate statistical significance; in detail, *$p < 0.05$, **$p < 0.01$, ***$p < 0.001$, ****$p < 0.0001$. Significance compared to untreated-WT cells is indicated by asterisks, connecting lines are used when the significance was compared to other samples. **f** Vitality assay (MTT) in HeLa WT, ZRF1-KO and ZRF1-KO cells with constitutive ZRF1 exogenous protein expression (ZRF1-RE). Cells were irradiated with different amounts of UV light. The graph show means of $n = 3$ biological independent experiments ± SEM.

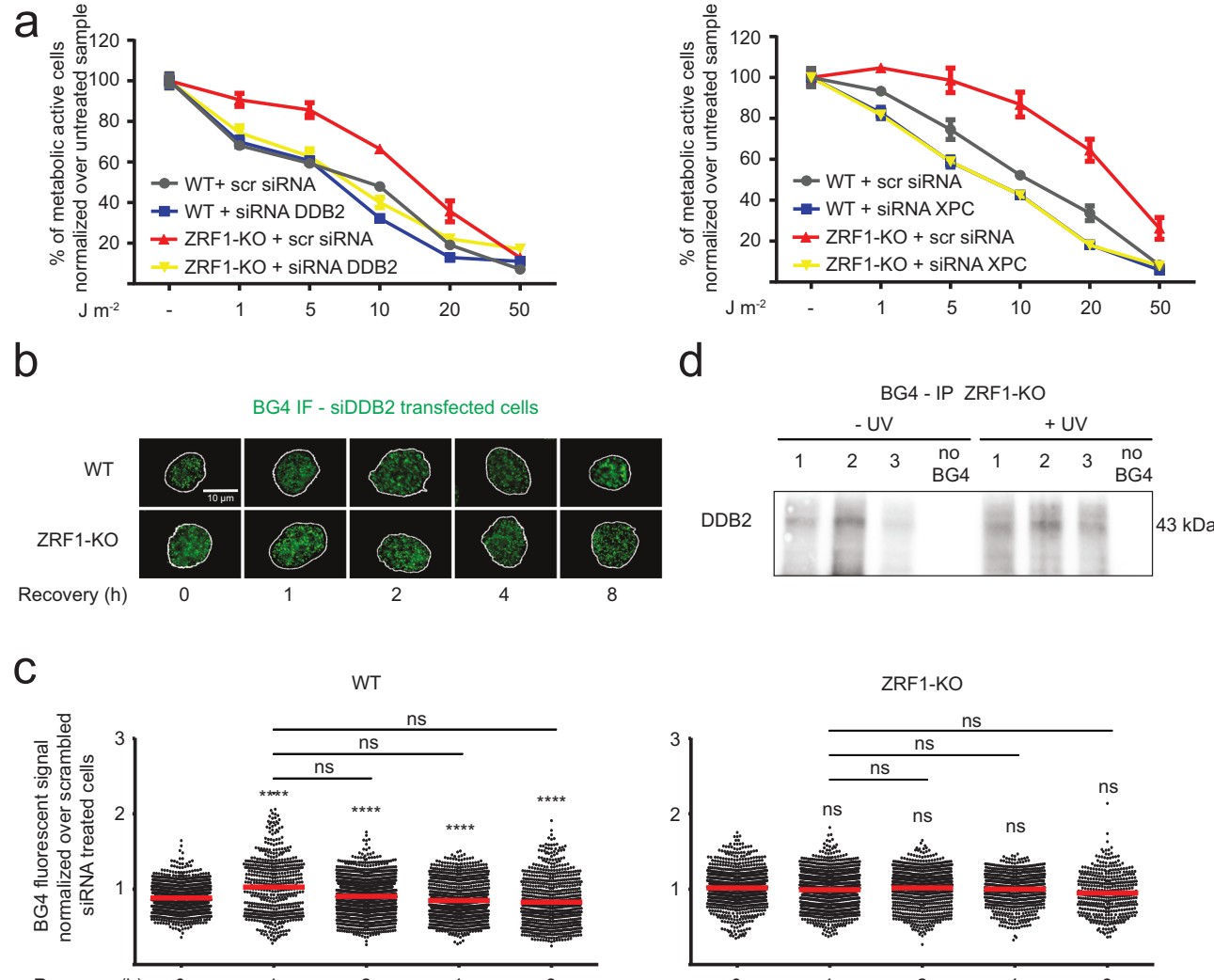

**Fig. 4 | DDB2 upregulation mediated GG-NER activity at G4s. a** MTT assay of WT and ZRF1-KO cells, transfected/untransfected for 48 h with DDB2 (left graph), XPC (right graph) or scramble siRNA (scr siRNA). Cells were treated with different amounts of UV light and recovered 24 h in DMEM/10% FBS. The graph show means of $n = 3$ biological independent experiments ± SEM. Note, ZRF1-KO + siRNA XPC values (yellow lane) are masked by WT + siRNA XPC values (blue lane). **b** IF staining of WT and ZRF1-KO cells, pre-treated 48 h with siRNA against DDB2 and treated/untreated with 10 J m$^{-2}$ UV light, recovered in DMEM/10% FBS for 1 to 8 h and stained with BG4 antibody (green) and DAPI (signal was used to indicate the nuclear border as a white line). Scale bar: 10 µm. **c** Quantification of BG4 signal in the

nucleus of the cells from **c**. Graph shows fluorescence intensity (FI) levels normalized over untreated cells of at least $n = 3$ biological independent experiments. Horizontal red line represents the mean value. Significance was determined using an ordinary one-sided ANOVA multiple comparison using the Geisser-Greenhouse correction. Asterisks indicate statistical significance; in detail, *$p < 0.05$, **$p < 0.01$, ***$p < 0.001$, ****$p < 0.0001$. Significance compared to untreated-WT cells is indicated by asterisks, connecting lines are used when the significance was compared to other samples. **d** BG4 immunoprecipitation performed in three independent replicates (marked 1, 2, and 3) of ZRF1-KO cells untreated or treated with 10 J m$^{-2}$ UV light and stained with antibody against DDB2.

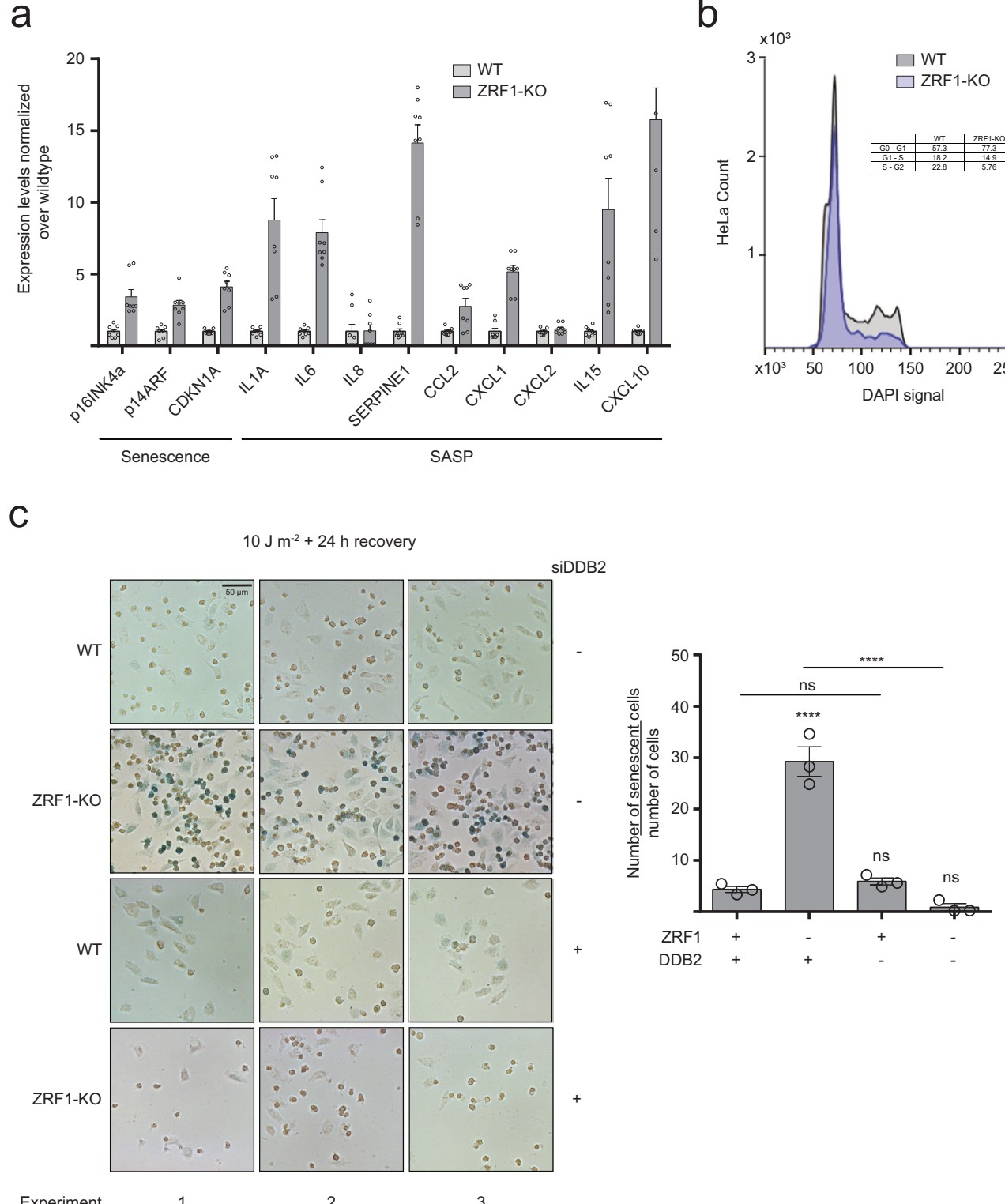

**Fig. 5 | ZRF1 prevents CDKN2A upregulation and UV-induced senescence.**
**a** Expression levels of senescence marker genes. mRNA levels were normalized to the level of U6 snRNA and GAPDH. WT mRNA levels were scaled to 1. Error bars represent SEM of $n = 8$ biological independent experiments. **b** Histogram plot of the DAPI signal in WT and ZRF1-KO cells, treated with 10 J m$^{-2}$ UV light and recovered 24 h in DMEM/10% FBS. The cell cycle distribution was obtained dividing the cells per DAPI amount. The table states the quantification (% of total) of the cells in the three cell cycle phases. **c** β-galactosidase staining of WT and ZRF1-KO cells, pretransfected with or without siRNA against DDB2, treated with 10 J m$^{-2}$ UV light and recovered 24 h in DMEM/10% FBS. Scale bar: 50 μm. Bottom part quantification of $n = 3$ biological independent experiments ± SEM. Significance was determined using an ordinary one-sided ANOVA multiple comparison using the Geisser-Greenhouse correction. Asterisks indicate statistical significance; in detail, $*p < 0.05$, $**p < 0.01$, $***p < 0.001$, $****p < 0.0001$.

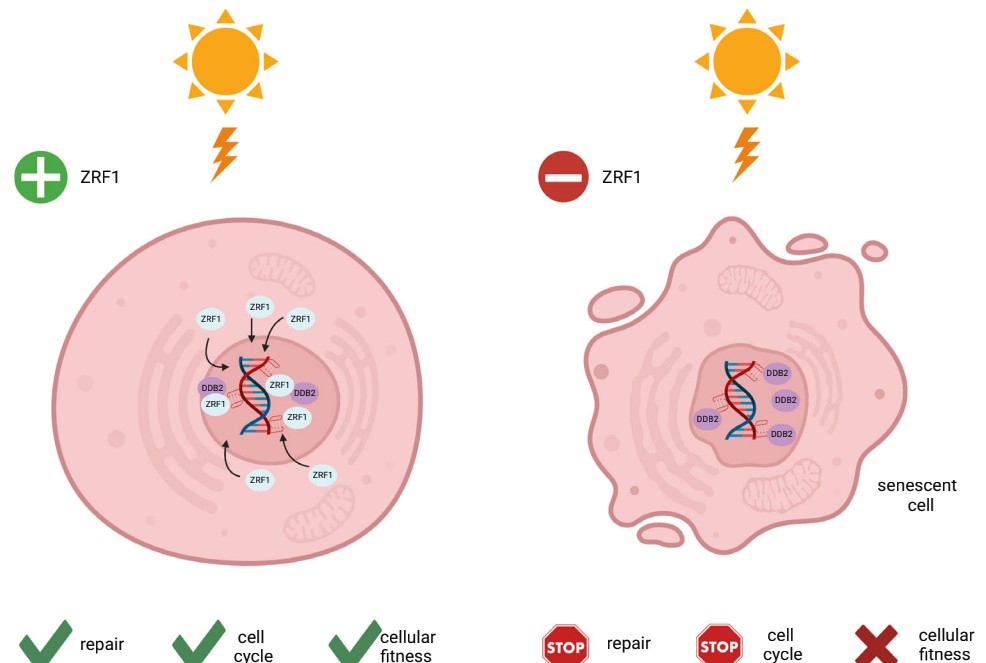

**Fig. 6 | Model.** UV irradiation induces G4s formation in the double helix. In wild-type cells ZRF1 enters the nucleus where it recognizes newly formed G4s and recruits DDB2. DNA is repaired and genome stability is maintained. Without ZRF1, newly formed G4s are recognized by DDB2, cells stall in G0/G1 phase and undergo senescence. Cell cycle blockage and senescence activation prevent activation of the repair machinery. These cells did not exhibit mortality in the first 3–4 days, but a complete mortality followed 1–2 weeks after treatment. The image was created with BioRender.com.

INK4-ARF genes (p15INK4b, p16ARF and p16INK4a). To further investigate this finding, we characterized the cell state of ZRF1-KO cells. It is known that CDKN2A activation induces an irreversible cell cycle arrest[6]. Therefore, we monitored cell cycle changes by flow cytometry using DAPI, which allowed the detection of distinct cell cycle populations such as G0/G1 phases, cells poised to enter replication at the G1/S phases and replicating cells in the S/G2 phases. A higher accumulation of G0/G1 cells (77%) was determined in ZRF1-KO compared to WT cells (57%) along with a significant decrease in cells at the S/G2 phases (6% in ZRF1-KO versus 23% in WT cells, Fig. 5b). We next assessed the cell growth over time. 3 days after UV irradiation WT cells returned to exponential growth, whereas ZRF1-KO cells did not show any growth and remained quiescent (Supplementary Fig. 6a). Flow cytometry analyses supported these findings as 5 days after UV irradiation ZRF1-KO cells still had a higher G1/G0 peak as WT cells (Supplementary Fig. 6b). These data agreed with the lack of colony formation of ZRF1-KO cells after UV irradiation (Supplementary Fig. 4f). We then quantified senescence in both WT and ZRF-KO cells by monitoring lysosomal β-galactosidase levels[9]. After UV irradiation ZRF1-KO cells displayed a strong increase in senescent cells compared to WT cells (Fig. 5c). Next, we aimed to correlate if the upregulation of DDB2 drives ZRF1-KO cells into senescence. After depletion of DDB2 by siRNA the β-galactosidase levels showed that ZRF1-KO cells did not undergo senescence anymore as indicated by reduced β-galactosidase staining. This result was supported by qPCR analysis using primer combinations targeting known senescence markers and SASP (Supplementary Fig. 6c). Taken together, our data showed that G4s accumulated upon UV irradiation and led to ZRF1 recruitment to chromatin which prevented DDB2-mediated UV-induced senescence.

## Discussion

It is of great importance to understand the molecular mechanisms and drivers that impact genome stability due to its relevance to premature aging and cancer. During daily life, multiple exogenous threats constantly and massively challenge genome stability, the most common risks being, among others, inhalable chemicals and UV irradiation. UV-mediated oncogenic mutations increase the incidence of malignant melanoma[60], which correlates with age and accumulation of sun exposure[61]. In this work, we aimed to deepen our understanding of UV-mediated processes that are connected to genome stability. We determined that upon UV irradiation, G4 structures rapidly accumulated (Fig. 1c) and ZRF1 was recruited to the nucleus where it bound genome-wide to G4s (Fig. 3c). This binding event was essential for an efficient UV damage repair because in the absence of ZRF1, 6-4PPs and CPDs formed rapidly and were not resolved within the first 24 h after UV irradiation (Fig. 3e, Supplementary Fig 4d). Loss of ZRF1 led to elevated G4 levels, indicating that without ZRF1 more G4s formed or were available to be detected by the G4 specific antibody BG4 (Fig. 2c, Supplementary Fig. 2d, e). We thus correlated the fast formation of 6-4PPs and CPDs and the delay in repair to the widespread accumulation of G4s, because similar changes were observed after treating cells with the G4-stabilizing ligand PDS (Fig. 1d).

After UV irradiation, ZRF1-KO cells showed no growth in colony formation assays (Supplementary Fig. 4f) but displayed a high metabolic activity as assessed by MTT assays (Fig. 3f), similar to cells with more and/or stabilized G4s (Fig. 1a, Supplementary Fig. 1a). Previously, it has been shown that ZRF1 function at DNA damage sites is linked to the NER factor XPC[34], where it supports the assembly of the CLR4-DDB2 complex[34]. Our data showed that without ZRF1, UV lesions (CPDs and 6-4PPS) rapidly form and accumulate (Supplementary Data 4). This is an indication that UV lesions are nor repaired. However, the lack in repair of UV lesions in the ZRF1-KO cells cannot be explained by the downregulation of repair proteins, as most repair proteins are not differentially expressed in ZRF1-KO cells. Rather something must prevent the repair machinery from binding and repairing the lesion. The ZRF1-KO cells arrested in G0/G1 phase, did no longer grow and genes related to senescence were upregulated. Therefore, we concluded that ZRF1-KO cells entered senescence (Fig. 6). We hypothesize that the repair machinery is functioning properly in ZRF1-KO cells but something prevents an efficient repair. One possibility could be the

entry of the cells into senescence, because upon entry the cells are no longer in the regular cell cycle phases. Most repair pathways depend on a specific cell cycle phase and proteins with a specific spatio-temporal expression in the cell cycle.

Although the impact of G4 structures and ZRF1 on NER function may be indirect, it is known that other DNA repair pathways are impacted by G4 formation using ad hoc ligands including PDS[28]. Indeed, some G4 ligands are cytotoxic in cells lacking homologous recombination (e.g., BRCA1- and BRCA2-deficient cells)[27,62,63]. To date, no protein that contributes to genome stability by binding to and not by unwinding G4s during repair processes has been identified. ZRF1 is thus, to our knowledge, the first protein that positively influences a DNA repair process by binding to G4 structures. We showed that ZRF1 binding to G4s prevented a strong upregulation of DDB2 (Supplementary Fig. 5a) and may also affect DDB2 binding to G4s themselves (Supplementary Fig. 5g). Our data proposes that DDB2 itself binds to G4s, but how strong this interaction is and if other proteins are part of this interaction remains elusive.

Like ZRF1, its yeast ortholog Zuo1 binds to G4s that accumulate after UV damage, which is found to be essential for the recruitment of NER proteins[21]. Supported by the evolutionary conservation (Fig. 3a, b) we anticipated that ZRF1 similarly to Zuo1 protects folded G4s and supports genome stability. However, we believe that ZRF1 does not recruit NER proteins itself but modulates expression and may be the binding of DDB2, thereby supporting indirectly NER by preventing the entry of cells into senescence.

Although ZRF1-KO cells showed enhanced metabolic activity as compared to WT cells (Fig. 3f), these cells were not dividing properly (lack of colony formation (Supplementary Fig. 4f) and exhibited growth defects (Supplementary Fig. 6b)). These results, combined with the observation that ZRF1-KO cells accumulated in G1 phase and had a distinct gene expression pattern of senescing cells (Fig. 5b, c, Supplementary Fig. 6c), supported our conclusion that ZRF1-KO cells entered senescence after UV irradiation. This hypothesis was further strengthened by the fact that cells that entered senescence experienced a metabolic reprogramming[64,65] in line with the elevated metabolic activity of ZRF1-KO cells after UV irradiation (Fig. 3f). Senescence, which is activated in response to DNA damage (e.g., telomere defects, oncogenic mutations, genotoxic stress) depends on the activation of p16$^{INK4a}$/pRB and p53/p21$^{WAF1}$ pathways[66] and acts as an anti-tumorigenic mechanism[67–69]. Many additional proteins act within this pathway and modulate the entry into senescence. We determined that the upregulation of DDB2 is essential to drive cells into senescence in ZRF1-KO conditions (Fig. 5a, c, Supplementary Fig. 6c). DDB2 was upregulated immediately after UV irradiation (Supplementary Fig. 5c). Elevated levels of DDB2 can be explained by changes in either transcription, mRNA decay, translation or protein degradation. All these pathways have been previously shown to be affected by G4s. For example, G4s in UTRs prevents mRNA decay[23]. As DDB2 levels changed immediately after UV irradiation it is unlikely that only transcriptional or translational changes were the cause of this decrease. But how ZRF1 affects DDB2 levels and how it influences mRNA decay or protein degradation is not clear, yet. DDB2 function is not fully understood but it acts in a complex where it affects the ubiquitination of histones and XPC, which directly correlates to NER efficiency[58]. DDB2 was shown to act downstream of p21 during senescence and cells lacking DDB2 fail to enter premature senescence[70]. As ZRF1 and DDB2 directly colocalize[34] we hypothesize that, in the absence of ZRF1, cells compensate for ZRF1 loss by upregulating DDB2 (Fig. 4a), which causes the entry into senescence, likely via the upregulation of p21, CDKN2A and CDKN1A (Fig. 5a).

Previously it was shown that enhanced G4 formation by G4 ligands leads to the formation of micronuclei[27,49] and affects the transcription of genes implicated in the regulation of DDR, senescence and cell death[71–73]. It was also reported that treatment with the G4 ligand

20 A does not induce replicative senescence but rather stress-induced premature senescence in HeLa cells[29]. These findings agree with our hypothesis that both the absence of ZRF1 and the presence of unprotected/unbound G4s drives cells into senescence. In summary, we conclude that G4s accumulate after UV irradiation, ZRF1 binds to and protects them and by this, prevents genome instability. Without ZRF1 G4s are unbound/unprotected, genome stability is challenged, DDB2 levels are elevated and cells enter senescence (Fig. 6).

Senescence activation is a double edge sword during cancer, because it is preventing cancer development on one hand, but likely contributes to reduced patient resilience on the other hand, which leads to cancer recurrence after therapy and cancer adaptation to therapy[74,75]. Mutation and/or deletion of ZRF1 was identified in AML, glioblastoma and different carcinomas. In contrast, amplifications of ZRF1 are linked to different subtype of cancer, e.g., prostate cancer and myeloid neoplasm (https://www.cbioportal.org/). We speculate that during cancer, which also correlates to high levels of G4s[76], ZRF1 is likely recruited to chromatin where it acts to prevent the entry into senescence, making ZRF1 a new player during senescence activation and modulation of cancer aggressiveness.

## Methods

### Cell lines and cell culture
Wildtype HeLa T-Rex Flp-In cells (Thermo Fischer Scientific) were grown in Dulbecco's modified Eagle's medium (DMEM) (Gibco) supplemented with 10% (v/v) fetal bovine serum (FBS, Gibco), 100 U ml$^{-1}$ Penicillin–Streptomycin (Gibco). Originating from these cells, transgenic cell lines with stable integration of constructs were generated by co-transfection of pcDNA3.1(+)-C-HA containing the gene of interest (GenScript). Positive clones of these cell lines were selected and cultured in the same media as described above supplemented with 500 μg ml$^{-1}$ G-418 (Sigma) reagent for the selection. All cell lines were passaged 2–3 times a week and incubated at 37 °C in 5% $CO_2$.

### CRISPR/Cas9 gene editing for ZRF1 knockout cells
crRNAs were designed using Benchling (https://benchling.com). Alt-R crRNA specific for ZRF1 was ordered (IDT). Gene editing was performed using the IDT guidelines. Briefly, 100 pmol Alt-R crRNA and 100 pmol Alt-R tracrRNA-ATTO 550 were denatured at 95 °C for 5 min and incubated at RT for 15 min to anneal both strands in a total volume of 100 μl in Nuclease-Free Duplex Buffer (IDT). 15 pmol annealed RNA were combined with 15 pmol Cas9 (IDT) and 5 μl Cas9 + reagent (Thermo Fisher Scientific) in Opti-MEM (Gibco) in a total volume of 150 μl and mixed well. In a second tube 125 μl Opti-MEM was combined with 7.5 μl CRISPRMAX (Thermo Fisher Scientific). After incubation at RT for 5 min the contents of the two tubes were combined, mixed well and transferred to a 6-well plate containing wildtype HeLa T-Rex Flp-In cells. The cells were seeded the previous day at a density of $3 \times 10^5$ cells ml$^{-1}$. After 1 h 1.5 ml of standard medium was added. 48 h after transfection ATTO 550 positive cells were FACS-sorted and seeded at the density of 1 cell per well in a 96-well plate using standard medium described above. Single clones were expanded and analyzed for loss of ZRF1 protein by western blot using an anti-ZRF1 antibody (Novus Biologicals ref #NBP2-12802) and confirmed by sequencing.

### Cell counting
A total of $2.5 \times 10^4$ cells were seeded in 24-well plates. 24 h post PDS or UV irradiation treatment cells were trypsinized and counted using a Bürker chamber. All analyses were performed in triplicate.

### MTT assay
Cytotoxicity of PDS, PhpC and UV irradiation was determined with a MTT assay. Seeding was performed in 96-wells plates. After the different treatments the medium was aspirated and cells were irradiated with 10 J m$^{-2}$ UV light (254 nm wavelength - Stratagene UV Stratalinker

1800 Crosslinker) and left to recover for 24 h in DMEM/ 10% FBS. Cells were washed with PBS, counted and fresh medium containing 500 μg ml⁻¹ of Thiazolyl Blue Tetrazolium Bromide solution (Sigma) was added to each well and incubated for 4 h in an incubator at 37 °C in 5% $CO_2$. Medium was subsequently removed and precipitated formazan crystals were solubilized in 100 μl dimethyl sulfoxide (DMSO). Absorbance at 570 nm was measured using a multiplate reader. Cell survival directly correlated with the absorbance values at 570 nm. Absorbance was normalized against untreated cells (negative control) and used to obtain a compound concentration with a cell viability ≥ 80%.

### Colony formation assay

The experiment was performed as previously described[77]. Briefly, cells were seeded in 6-well plates. 24 h after seeding the PDS treatment or the siRNA transfection was performed. After 24 h of PDS treatment or siRNA transfection the cells were trypsinized and 1000 cells were re-seeded in 10 cm dishes. 24 h later cells were washed with PBS and UV irradiated (10 J m⁻²). Two weeks later cells were fixed with 6% glutaraldehyde solution (Sigma) containing 0.5% crystal violet (Sigma). After 30 min incubation at RT cells were washed with ddH₂0 and dried at RT. Colorimetric pictures were acquired on a ChemiDoc Imaging System (Biorad).

### 6−4PPs − CPDs immunofluorescence analyses

A total of $3 \times 10^4$ cells were seeded in 24-well plates. 24 h post seeding cells were incubated with 2 μM PDS for 48 h. Cells were irradiated with 10 J m⁻² UV light and recovered from 0 to 4 h (6-4PPs IF) or from 0 to 24 h (CPD IF) and fixed with a 4% PFA solution (diluted in PBS) for 10 min at RT. After 3 washes with PBS cells were incubated 5 min at RT with 0.5% (v/v) Triton X-100 in PBS. DNA was then denatured with a 2 M HCL solution 30 min at RT. After 5 washes with PBS cells were incubated for 30 min at 37 °C with a 20% FBS blocking solution (in PBS). Cells were incubated with 1:1000 anti 6-4PPs antibody (Cosmo-Bio #CAC-NM-DND-002) or 0.5 μg ml⁻¹ anti CPD-antibody (Kamiya Biomedical #MC-062) diluted in 5% FBS (PBS) and incubated for 1 h at 37 °C. Cells were then incubated with fluorescent secondary anti-rabbit IgG (Life technologies #A10520) diluted 1:1000 in blocking solution for 1 h at RT with gentle rocking. After each step the cells were washed three times for 10 min with PBS. Cover glasses were mounted with Fluoroshield mounting media containing DAPI (Merck). Slides were visualized on a fluorescence microscope (Zeiss Axio Observer Z1). Fluorescence signal was determined using Fiji[78]. The plots were prepared using GraphPad Prism 6.2.

### CPDs quantification

gDNA was extracted using a DNA extraction kit (Qiagen). A total of 500 ng of DNA was diluted in 100 μl of PBS and heated at 95 °C for 10 min to denature the DNA. Two Whatman paper and a Hybond+ membrane (Amersham) was placed in a dot blot apparatus and washed twice with PBS. Samples were immediately placed on ice for 5 min after denaturation and were loaded on the Hybond+ membrane. Membrane was allowed to air dry for 15 min. DNA was then crosslinked in an oven at 80 °C for 1.5 h. The membrane was blocked for 1 h with blocking solution (5% non-fat milk in TBS with 0.05% Tween 20 (TBS-T)). Membranes was incubated overnight at 4 °C under rotation, with 0.5 μg ml⁻¹ anti-CPD antibody (Kamiya Biomedical #MC-062) diluted in blocking buffer. After three 10 min washes with TBS-T the membrane was incubated with matching HRP-coupled anti-mouse secondary antibody (Santa Cruz Biotechnology sc-516102) 1:5000 diluted for 1 h at RT followed by another three washing steps. Signal was detected by chemiluminescence of HRP-coupled secondary antibodies with a ChemiDoc Imaging System. The concentration of the crosslinked DNA was determined by SYBR Gold staining (Thermo Fisher Scientific). CPD-IF was performed following the kit instruction (Kamiya Biomedical).

### BG4 purification

The plasmid expressing an engineered antibody specific to G4 (BG4)[47] was kindly provided by S. Balasubramanian (University of Cambridge, UK). The plasmid was transformed into BL21(DE3) competent cells. BG4 antibody was purified as described previously[21]. BG4 antibody was quantified on a NanoDrop spectrophotometer (Thermo Fisher Scientific) and stored at −80 °C. Purity of the BG4 preparation was monitored by SDS-PAGE and ELISA.

### BG4 immunofluorescence

BG4 immunofluorescence was performed as previously described[79]. Briefly, cells were seeded in 6- or 24-well plates. 24 h post seeding the cells were treated with 2 μM PDS for 48 h and pre-fixed with a 50/ 50 solution of DMEM and methanol/acetic acid (3:1) at RT for 5 min. After a brief wash with methanol/acetic acid (3:1) the cells were fixed with methanol/acetic acid (3:1) at RT for 10 min. Cells were then permeabilized with 0.1% (v/v) Triton X-100 in PBS at RT for 3 min with gentle rocking and incubated with a blocking solution (2% (w/v) dry milk in PBS, pH 7.4) for 1 h at RT with gentle rocking. Afterwards cells were incubated in blocking solution containing 0.5–1 μg of BG4 antibody per slide and incubated for 2 h at RT. Cells were then incubated with blocking solution containing 1:800 rabbit polyclonal antibody against the DYKDDDDK epitope (Cell Signaling #2368) for 1 h at RT with gentle rocking. Next, cells were incubated at RT with blocking solution containing 1:1000 fluorescent secondary anti-rabbit IgG (Life technologies #A10520) for 1 h at RT with gentle rocking. After each step, cells were washed three times with 0.1%(v/v) Tween-20 in PBS for 10 min. The cover glasses were mounted with a drop of Fluoroshield mounting media solution (Merck) containing the DNA staining fluorophore DAPI.

### ZRF1 immunofluorescence

A total of $3 \times 10^4$ cells were seeded in 24-well plates. 24 h post seeding cells were incubated with 10 μM PDS or cPDS for 24 h. Pre-extraction with CSK buffer (10 mM PIPES pH 7.4, 100 mM NaCl, 300 mM sucrose, 3 mM MgCl₂) containing 0.2% Triton X-100 was performed for 5 min on ice before fixation. Cells were washed twice with PBS and fixed with a 4% PFA solution (diluted in PBS) for 10 min at RT. After 3 washes with PBS cells were incubated 5 min at RT with 0.2% (v/v) Triton X-100 in PBS. After 1 h incubation with blocking solution (5% FBS, 0.1% Triton X-100 in PBS) cells were incubated with 1:500 ZRF1 antibody (Novus Biologicals #NBP2-12802) diluted in blocking solution overnight. On the next day the cells were incubated at RT with fluorescent secondary anti-rabbit IgG (Life technologies #A10520) diluted 1:1000 in blocking solution for 1 h at RT with gentle rocking. After each step, cells were washed three times for 10 min with PBS. Cover glasses were mounted with Fluoroshield mounting media containing DAPI. Slides were visualized on a fluorescence microscope. Fluorescence signal was determined using Fiji[78]. The plots were prepared using GraphPad Prism 6.2.

### Chromatin immunoprecipitation sequencing (ChIP-seq)

A total of 1 million cells were seeded in 15 cm dishes. 24 h post seeding the cells were incubated with 2 μM PDS. 48 h and 72 h post seeding the cells were irradiated with 10 J m⁻² UV light. Cells were crosslinked with a 4% PFA solution for 10 min at RT with gentle rocking followed by 10 min incubation with 0.125 mM glycine. Nuclei were isolated and lysed with the Chromatrap Hypotonic and Lysis buffer and DNA was sheared using a E220 Focused-Ultrasonicator (Covaris). To the sheared chromatin 2 μg of ZRF1 antibody (Novus Biologicals #NBP2-12802) was added and incubated for 16 h at 4 °C under rotation. The next day samples were incubated with 80 μl Dynabeads-Protein G (Thermo Fisher Scientific) for 2 h at 4 °C. After washing two times with washing buffer (100 mM KCl, 0.1% (v/v) Tween-20 and 10 mM Tris-HCl pH 7.4) immunoprecipitated DNA was treated with Proteinase K for 1 h at 37 °C

and the crosslinking was reversed by overnight incubation at 65 °C. Immunoprecipitated DNA was purified using the QIAquick PCR Purification Kit (Qiagen). For genome-wide sequencing DNA was treated according to manufacturer's instructions (NEBNext® Ultra™ II DNA Library Prep Kit) and submitted to sequencing on a HiSeq 2500 sequencer (Illumina). Obtained sequence reads were aligned to the human reference genome with Bowtie[80]. Binding regions were identified by using MACS 2.0 with default settings for narrow peaks[81]. IGV genome browser was used to visualize genome annotation of sequencing reads. The ChIP input was used as a control data set. Overlap of binding sites with other genomic features and binding regions were determined using a PERL script based on a permutation analysis between the query and subject features.

## BG4 IP

Immunoprecipitation was performed as described in[82] with minor changes. $1 \times 10^6$ cells were seeded in 15 cm dishes. 24 h post seeding cells were irradiated with 10 J m$^{-2}$ UV light. Cells were crosslinked with a 4% PFA solution for 10 min at RT with gentle rocking followed by 10 min incubation with 0.125 mM glycine. Nuclei were isolated and lysed with the Chromatrap Hypotonic and Lysis buffer and DNA was sheared using a E220 Focused-Ultrasonicator. To the sheared chromatin 10 μg of BG4 antibody was added and incubated for 2 h at 4 °C under rotation. 2 μl of mouse monoclonal antibody (Sigma) against the DYKDDDDK epitope of BG4 was added to the solution and kept for 1 h at 4 °C under rotation. Samples were incubated with 50 μl Dynabeads-Protein G (Thermo Fisher Scientific) for 2 h at 4 °C. After washing two times with washing buffer (100 mM KCl, 0.1% (v/v) Tween-20 and 10 mM Tris-HCl pH 7.4) immunoprecipitated proteins bound to DNA were de-crosslinking by boiling the samples 10 min at 95 °C in 1× Laemmli buffer. The samples were then separated by SDS-PAGE and immunoblotted (as describe below).

## Western blot analysis

For standard protein analysis protein lysates were obtained by lysing the cells in NP-40 lysis buffer or 1× Laemmli buffer supplemented with 100 units of Benzonase Nuclease (Sigma). Proteins were separated by SDS-PAGE (8–15%) and blotted on a nitrocellulose membrane (GE Healthcare). For Ponceau staining the membrane was stained with Ponceau solution (Sigma) for 5 min at RT with gentle rocking. The Ponceau solution was removed through several washes with ddH$_2$0. After saturating free binding sites with 5% non-fat milk powder in TBS-T the membrane was incubated with suitable primary antibodies, anti-ZRF1 antibody 1:1000 (Novus Biologicals #NBP2-12802), anti-DDB2 antibody 1:500 (Santa Cruz sc-81246) or anti-XPC antibody 1:500 (Santa Cruz sc-74410)for 16 h at 4 °C under rotation. After three times 10 min washing with TBS-T the membrane was incubated with matching HRP-coupled secondary antibodies (anti-mouse sc-516102 or anti-rabbit sc-2357; Santa Cruz Biotechnology) 1:5000 diluted for 1 h at RT followed by another three washing steps. Signals were detected by chemiluminescence of HRP-coupled secondary antibodies (Santa Cruz Biotechnology) on a ChemiDoc Imaging System. Uncropped blots are provided in the Source Data file.

## Yeast strains and growth analysis

Plasmids containing isoform 1 and 2 of ZRF1 were generously provided by Prof. Elizabeth A. Craig and used as previously described[54]. Yeast cultures (W303 background) were cultured to an OD$_{600}$ of 0.5–0.6 at 30 °C. All yeast cultures were adjusted to OD$_{600}$ of 0.5 and six series of 1:5 dilutions were prepared. From each dilution 5 μl were spotted on a YPD plate and incubated at 30 °C. For UV irradiation cells were irradiated with 15 J m$^{-2}$ UV light after plating. After 2 days the plates were scanned and the growth of strains was compared to estimate growth defects.

## 3′ RNA-seq

RNA from wildtype and ZRF1-KO HeLa T-Rex Flp-In cells, which were untreated or irradiated with UV light (10 J m$^{-2}$), was isolated with TRIzol (Thermo Fisher Scientific) according to the manufacturer's instructions. Libraries were prepared using the QuantSeq 3′-mRNA Library Prep kit (Lexogen) and sequencing was performed on a HiSeq 2500. Obtained sequence reads were aligned to the human reference genome with STAR[83]. Differential expression was determined quantifying enriched reads over the reference genome.

## Quantitative PCR

Quantitative PCR (qPCR) was performed using the iTaq Universal SYBR Green Supermix (BioRad). Fold enrichment of the specific transcript of interest was normalized over the housekeeping transcript RNAU6 and GAPDH. Microsoft Excel and GraphPad Prism 6.2 were used to plot the graphs.

## siRNA transfection

After 24 h of seeding ($1.75 \times 10^5$ cells HeLa wildtype and $2 \times 10^5$ ZRF1-KO cells) HeLa wildtype and ZRF1-KO cells in 6-well plates they were forward transfected with 100 pM siRNA specific for DDB2 (Sigma #SASI_HSo1_00101645) or XPC (Thermo Fisher Scientific #AM16708-139414) and scrambled siRNA using Lipofectamine™ RNAiMAX Transfection Reagent (Thermo Fisher Scientific). Protein knockdown was assessed by western blot of nuclear proteins extracted from cells 48 h post transfection.

## Protein–DNA interaction on native PAGE

After creating serial dilution of the DDB1-DDB2 proteins (heterodimer purified and obtained from the Thomä lab[56]), the complex was incubated with the oligonucleotides pu27_c-Myc_G4 (AAATGGGGAG GGTGGGGAGGGTGGGGAAGG) or pu27_c-Myc_mutG4 (AAATAGCGAG AGTGAGCAGTGTGCGTAAGG) at RT for 30 min. The complex was loaded on a native polyacrylamide (19:1) gel and run at 60 V for 60 min. After a drying step on Whatman paper the gel was exposed on a phosphoimager screen for 16 h. The screen was analyzed using a Typhoon FLA 7000 biomolecular imager (GE Healthcare).

## Flow cytometry

DAPI staining was performed as previously reported[79]. Briefly, HeLa wildtype ($1.75 \times 10^5$ cells) and ZRF1-KO ($2 \times 10^5$ cells) cells were seeded in 6-well plates. 72 h post seeding the cells were irradiated with 10 J m$^{-2}$ UV light and left to recover in DMEM/10% FBS. 24 h post irradiation the cells were harvested, washed with PBS and fixed for 30 min at RT with methanol. Cells were then washed with PBS and stained with a DAPI solution (10 μg ml$^{-1}$ DAPI, 50 μg ml$^{-1}$ RNase in PBS) for 30 min at 37 °C. Cells were finally resuspended in PBS and analyzed by flow cytometry on a BD FACSCanto™ II Cell Analyzer. After data acquisition, the data was analyzed using FlowJo™ (v10.8 Software BD Life Sciences) gating the cell for the size (forward scatter (FSC)) and granularity of the cells (side scatter (SSC)).

## β-galactosidase staining

HeLa wildtype ($1.75 \times 10$ cells) and ZRF1-KO ($2 \times 10$ cells) cells were seeded in a 6-well plate. Post 72 h of seeding the cells were irradiated with 10 J m$^{-2}$ UV light and left to recover in DMEM/10% FBS. 24 h post irradiation cells were washed twice with PBS and fixed for 15 min at RT with fixation solution provided in the staining kit (Cell signaling). Cells were then washed twice with PBS and stained with X-gal staining solution provided in the kit and incubated 24 h at 37 °C in agitation. Slides were visualized on a fluorescence microscope.

## Statistical analyses

Significance was calculated using one-sided Student's t-test and ordinary one-way ANOVA multiple comparison tests. Asterisks indicate

statistical significance in comparison with wildtype cells: *$p < 0.05$, **$p < 0.01$, ***$p < 0.001$, ****$p < 0.0001$. Plotted results were based on the average of at least three biologically independent experiments.

### Reporting summary

Further information on research design is available in the Nature Portfolio Reporting Summary linked to this article.

## Data availability

ChIP-seq data have been uploaded in the National Center for Biotechnology Information (NCBI) Sequencing Read Archive under the reference number PRJNA817435. All data are available upon request from the corresponding author. Source data are provided with this paper.

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

## Acknowledgements

We thank Philipp Schult, Daniel Hilbig, and Lea Sauer for experimental support and Philipp Simon for careful reading of the manuscript. We thank the Thomä lab for providing the purified DDB1-DDB2 heterodimer[56]. We thank the Craig lab for providing us with the plasmid expressing ZRF1[54]. Research in the Paeschke laboratory is funded by the Deutsche Forschungsgemeinschaft (DFG, German Research Foundation) under Germany's Excellence Strategy—EXC2151—390873048, the Fritz Thyssen foundation and „CANTAR". The project "CANTAR" is receiving funding from the programme "Netzwerke 2021", an initiative of the Ministry of Culture and Science of the State of Northrhine Westphalia. A.D.M. postdoctoral fellowship is funded by BONFOR (University Hospital Bonn). D.M. thanks R.H.E. Hudson and J. Mitteaux for their help in deciphering the G4-unfolding ability of PhpC, and the CNRS, FEDER-FSE and Ligue Contre le Cancer for fundings.

## Author contributions

Conceptualization, K.P., A.D.M.; Methodology, A.D.M., M.L. V.M.; Data Analysis, A.D.M., S.J. K.P.; Writing—Original Draft, A.D.M, K.P.; Writing—Review & Editing, A.D.M., M.L., S.J., D.M., K.P.; Funding Acquisition, K.P.; Resources, K.P.; Supervision, K.P.

## Funding

## Competing interests
