## [Peer Review File · Nature Communications]

UV-induced G4 DNA structures recruit ZRF1 which prevents UV-induced senescenceREVIEWER COMMENTS

Reviewer #1 (Remarks to the Author):

In this paper, De Magis and co-authors describe the role of the ZRF1 protein in binding UV-induced G4 structures and subsequently supporting the repair process to prevent senescence. This paper follows a previous one in which De Magis and co-authors identified a protein in yeast, Zuo1, as involved in G4 structure formation and in the mechanism of nucleotide excision repair.

The paper is well written, the data are technically sound, well presented, robust and support conclusions. Significance of the paper is high, since to my knowledge this is the first time a protein is reported to both bind G4 structures and support a repair process in cells.

I have only some major issues that need to be addressed:

- How was UV irradiation performed? Which wavelengths were used? How were the UV doses chosen? Are they comparable to the sun radiation? Please specify.

- The authors demonstrate that the use of "100 μ M PhpC reduced G4s significantly while it had no effect on cell vitality per se". How can the authors be sure that PhpC does not interfere with BG4 binding, and ZRF1 binding as well? How did they demonstrate it? This point has to be clearly proved. Minor points:

- Please describe in the material and methods section how were the plasmids containing the two isoforms of XRF1 obtained.

- Fig. S3 represents "IF staining of WT and ZRF1-KO cells, treated/untreated 48 h with 1 or 2 μ M of PDS": why are there two panels? Please clarify.

- In the legend to Fig. 2 please move "orange asterisks..." from letter e) to letter d).

- In the legend to Fig S5d "G4" and "G4 mut" seem to refer to the black triangle, which indicates the amount of recombinant protein: please make the figure clearer.

- Please check all references: on p. 5, ref. 32 refers to "Miglietta, G., Russo, M., Duardo, R. C. & Capranico, G. Nucleic Acids Res. 49, 6673–6686 (2021)" and not to PhpC.

- Please check all over the text for some sentences that seem like comments that were not removed (Legend of Fig. S2a "as predicted by ??", Legend of Fig. S5 "(The label has moved!)", legend of Fig. S3 "Why two times??").

Reviewer #2 (Remarks to the Author):

Previously, the authors had studied yeast Zuo1 function in relation to formation of G4 structures and nucleotide excision repair (NER) activity. In this manuscript, the authors continue/confirm these studies by focusing on the human Zuo1 ortholog Zrf1. They show that stabilization of G4 structures enhances cell vitality after UV irradiation, which they claim is not due to enhanced DNA repair, and that G4 structures also form after UV irradiation. ZRF1 is shown to binds to G4 structures in cells and to be necessary to prevent genome instability. Human ZRF1 also partially rescues UV sensitivity of yeast Zuo1 mutants, suggesting an evolutionary conserved function. Furthermore, ZRF1 KO leads to enhanced vitality after UV and increased CPD levels. The authors report that upregulation of NER genes DDB2 and XPC in ZRF1 KO cells is responsible for the enhanced vitality of ZRF1 KO cells after UV. Finally, they report that UV-induced senescence induction in ZRF1 KO cells is dependent on DDB2.

The manuscript contains many experiments, with some interesting and striking results such as increased viability and elevated CPD levels in ZRF1 KO cells after UV. However, experiments are often not sufficiently convincing or well executed and there are too many issues that need to be addressed before I can reach the same conclusions as the authors do. Below I describe my main issues that I think need to be addressed before this manuscript is suited for publication anywhere.

On page 5, the authors start by hypothesizing that G4 formation could influence NER because XPB and XPD were reported to bind G4 regions. Therefore, they perform viability MTT assays after UV

irradiation, apparently to investigate NER activity upon G4 formation, as shown in Figures 1A and 3F. My concern with this strategy is that standard assays in the NER field are colony formation assays, in which few hundred cells are plated in a well and colony formation is measured. In this type of assay, cells are irradiated without any liquid (that protects against UV) and much lower UV doses (10x) than the authors use are applied. Because the authors do not use the more common method and do not take along NER deficient controls, it is difficult to evaluate how the UV sensitivity of PDS-treated and ZRF1 KO cells compares to that of truly NER deficient cells. Therefore, it is not really possible to draw any definite conclusions about NER activity based on these experiments. Ideally, the authors should include a NER-deficient or -knockdown control cell line in Figs 1A and 3F. In Fig 5B the authors do knockdown DDB2 and XPC using this assay, but observe no (DDB2) or only mild (XPC) UV sensitivity. This is strange and based on this poor UV sensitivity, I strongly doubt whether this assay in this way is really useful to determine NER capacity or functionality in response to UV irradiation. NER deficient cells are strongly UV sensitive and should not survive even low doses of UV. It would therefore be more convincing if the authors could show similar results using colony formation assays performed with proper controls. Moreover, previously, ZRF1 loss in human cells was reported to lead to similar UV sensitivity as XPC loss (Gracheva et al 2016). How do the authors explain their opposite results?

On the bottom of page 8, the authors conclude that ZRF1 binds 'to G4 structures after UV irradiation where it supports repair'. Also, on page 9 the authors hypothesize that 'without ZRF1, NER repair is not functional'. This conclusion seems premature and cannot be drawn from the results presented. If ZRF1 would support repair, its loss would be expected to result in decrease UV survival, as does the loss of any NER or other protein that supports repair. However, the authors show the opposite, which is that ZRF1 KO cells have increased vitality after UV irradiation. If NER would not be functional, UV irradiation should kill the cells already at very low dose. I understand that the authors may want to make the point that ZRF1 KO cells have increased vitality in spite of having deficient NER, but then they should show that indeed in the absence of functional NER (so knockout of a NER protein) ZRF1 KO cells still have elevated vitality after UV. I strongly doubt whether they will find this based on the XPC knockdown results shown in Fig 4B. These results show that XPC knockdown still leads to lower UV survival, implying that NER must be functional in these cells. Importantly, the authors also have not correctly measured CPD repair, see my comment below, but observe a seemingly impossible increase in CPD levels after UV in 4 hrs. This cannot be used to conclude anything about repair.

Fig 1D. To be able to accurately assess the outcome of this dot blot, which strikingly shows a huge increase in CPD levels in the presence of PDS, the authors should include a loading control (stain the blot with an anti-DNA antibody or equivalent). Also, the experiment should be replicated (at least three times) and quantified (normalized to the DNA signal). Ideally, a control such as a NER deficient cell line should be included to understand what can be expected in the absence of repair. It is important to understand how much more CPDs are detected in Fig 1D in the PDS condition, as this result is unanticipated and opposite of the enhanced survival shown in Fig. 1A (the same is true for Fig 3E and F). I doubt whether the total amount of potential G4 sites in the human genome (700000?), when stabilized, can account for this increase in CPD lesions, as 10 J m⁻² will probably already induce hundred thousands of CPDs in the human genome. The authors should address this and consider that another activity of PDS than G4 stabilization is responsible for the observed increase. Most important is that the assay is not performed with the right recovery time after UV to measure CPD repair. Human CPD repair is slow (~50% repair in 24 hrs; PMID 4010689) and it is therefore necessary to evaluate CPD repair for a period of at least 12 - 24 hrs (which is common in the field). Because the authors measure CPD levels up to only 4 h after UV, also in the untreated condition repair is not visible as weaker CPD signal. The authors should therefore perform this assay correctly, with sufficient recovery time, with loading control, with NER deficient control and with replicates and quantification. The same holds true for Fig 3E (which appears to be performed on two different blots, which should be avoided). Moreover, to substantiate the weird observation that CPD signals are higher after PDS treatment or ZRF loss, the authors should consider to perform similar experiments by CPD immunofluorescence in cells. Also, it would be worthwhile to test if similarly elevated levels are observed for 64PPs, the other abundant type of UV photolesion (which are repaired much faster within

a few hours).

Fig 1E. The authors should explain more clearly how this assay is used to measure ZRF1 binding to G4 structures. I guess this is based on washing away free ZRF1, to determine if ZRF1 is more stably bound to DNA structures, but the authors state that they quantify 'localization' of ZRF1 which confuses. The authors should also show specificity of the antibody in IF, which could for instance be verified by knocking down ZRF1 or do this assay in ZRF1 KO cells. Also, it would be clearer if not only merged images but also the separate green and blue channel are shown.

Supplementary Fig 2D. It is difficult to conclude from these experiments that a 'small but significant enrichment of G4s was detected' as this is not observed with all PDS doses. Rather, the conclusion should be that ZRF1 KO does not really affect G4 levels (at least not, as it seems, in a biological relevant manner).

Page 7. γ H2AX staining can indeed be used as marker for DSB formation, but H2AX is also phosphorylated when single-strand breaks are formed, as for instance also happens in response to UV irradiation when DNA damage is excised by NER (see e.g PMID 17615256; 19797077). γ H2AX signal after UV irradiation is dependent on NER. Therefore, it is not entirely accurate to state that γ H2AX is a known DSB marker and that the increased micronuclei are independent of DSB formation. Apparently, there is no clearly increased or decreased formation of single- or double-strand breaks in ZRF1 KO cells, which should be the conclusion. Please address this issue.

Page 8 and 9. The authors should show and explain which are the genes 'involved in the response to UV-A irradiation' that are differentially expressed in ZRF1 KO cells and what this has to do with NER? The reason I ask this is because NER acts in response to photolesions induced by (mostly) UV-C and UV-B irradiation, but not so much by UV-A irradiation (which would be rather induce oxidative damage and single-strand breaks that require a different repair pathway). It is unclear to me why the authors chose to focus on NER based on this result, also because I can hardly find any NER gene in supplementary Table 1 and NER genes are not typically found to be overexpressed in response to DNA damage induction. Please elaborate on this. Also, it is unclear if these genes are up- or down-regulated.

DDB2 is not an essential NER gene, as suggested by the authors, as is also evident from the lack of effect of siDDB2 in the UV MTT assay of wild type cells shown in Fig 5B. The NER reaction can be reconstituted in vitro without DDB2 and DDB2 is thought to merely stimulate NER of less bulky lesions and in certain chromatin environments. It is therefore questionable whether a transcriptional upregulation of DDB2 by 1.5 fold will majorly affect cell viability after UV through NER activity, also because DDB2 acts in complex with DDB1, Cul4A and RBX1 and its function will depend on the availability of these factors as well. To test their hypothesis, the authors should therefore show upregulation of this entire protein complex (on protein level). Furthermore, the authors hypothesize that 'without ZRF1, NER repair is not functional and CPDs accumulate which further activates DDB2'. However, this is contrary to the generally accepted fact in the field (shown by different labs) that in response to UV irradiation and/or upon accumulation of unrepaired damage (CPDs) DDB2 is auto-ubiquitinated and degraded by the proteasome. The authors find the opposite (no degradation) in WT and ZRF1 KO cells in Fig S5C (again not quantified)/replicated). This figure also shows that in spite of transcriptional upregulation, DDB2 protein levels are not higher in ZRF1 KO cells. Thus, DDB2 does not appear to be expressed at higher levels in unchallenged ZRF1 KO cells, which argues against the proposed hypothesis. These issues should all be addressed and possibly/likely different conclusions should be drawn.

Introduction, page 4. XPC is not an endonuclease. Please correct this error. Also, XPC does not act alone but as part of a heterotrimeric complex. Please accurately describe this. XPC is ubiquitinated by a complex consisting of DDB2, DDB1 and CUL4A and RBX1. Please also accurately describe this. This ubiquitination, however, may not be an 'essential step for NER initiation', as in the absence of DDB2,

XPC is still able, albeit less efficient, to detect many types of DNA damage. Importantly, this DDB2-mediated ubiquitylation is generally accepted in the field and has been confirmed and studied by multiple labs, both in vitro, in cells and in vivo. This is different for the ubiquitination in which ZRF1 is implicated. Involvement of ZRF1 is not (yet) generally accepted in the field and so far only published by one lab (which, I think, does not exist anymore). The authors should therefore describe the proposed/potential involvement of ZRF1 (which involves also another E3 ubiquitin ligase complex) with more care and caution. ZRF1 cannot be claimed to be 'part of the NER complex'.

Fig. 1B. Is there a statistically significant reduction in vitality with 100 μ M PhPC? Why is this experiment (together with the one shown in supplementary Fig 1B) not depicted the same way as in Fig 1A (why only one dose now?). Why is the survival of untreated sample not shown in the same graph? This is confusing.

Fig 2D. It is unclear what is shown in the graph. The legend says fraction but the graph depicts percentage micronuclei per cell. What is a percentage micronuclei per cell? Also, how many cells were quantified per experiment and are the differences statistically significant? Same for supplementary Fig 3A-B. Is the untreated condition in all these graphs from the same experiments? Please indicate this, or show this only in one of the graphs. Also, the text indicates that a range of range: 1-5 μ M PDS is used, but only up to 2 μ M is shown in these figures. Also, what is the difference between these experiments because the legends of all three figures indicate that this was 48 h of PDS exposure, while according to the text the exposure was for 24, 48 and 72 h. It is annoying that the authors have not been accurate in preparing the manuscript. Also, the legend still has the text: 'Why two times??'.

Supplementary Fig 5D. This experiment is poorly explained both in the text and the methods. How were DDB2 and DDB1 purified as this is nowhere shown? Also, the legend and figure are confusing. The legend suggests increasing concentrations of DDB1-DDB2 were used, but the figure suggests as if the G4 DNA dosage was increased. Also, the label was moved and this is still stated as a comment in the legend: 'The label has moved!'. Again, this is an example of inaccuracy in preparing the manuscript (and not the only example). Also, it is unclear why the experiment is performed. Do the authors wish to show that DDB1-DDB2 can bind to G4 structures and can therefore also do this in cells? If so, why don't they show that DDB2 binds to G4 structures in cells as they did also for ZRF1? And why not show whether this depends or happens in concert with ZRF1 in cells?

Page 10, top. The authors write that 'We monitored G4 levels by IF both in WT and ZRF1-KO cells after depletion of DDB2 and after UV irradiation (0-8 h after treatment; Fig. 4c,d).' However, it is unclear what is shown in Figure 4c and d. Is this the siDDB2 condition? This is not described in the legend. But if so, where is the siCtrl to which this is compared? It is not possible for me to judge this result or the conclusions if I don't know what I am looking at. Also, in Fig 4D it is unclear and not described to which statistic comparison is indicated by the asterisks. It would be much cleared if the authors show all statistic similarly, i.e. with connecting lines, so that the reader knows which results are compared.

Page 7, bottom. Please explain the difference between both ZRF1 isoforms (in text or a figure) and also provide some information on the evolutionary conservation between Zuo1 and ZRF1 with regard to amino acid sequence and/or protein motifs. If only the second isoform rescues the UV sensitivity, what makes this isoform different from the first isoform that could explain this difference. Are both isoforms equally expressed, also given the fact that their overexpression causes UV sensitivity in the wild type strain? Also, is the second isoform then more closely related to the yeast Zuo1? Please elaborate on this in the text.

Page 9, bottom. I do not agree with the authors that the survival with DDB2 and XPC siRNAs shows that the enhanced vitality of ZRF1 KO cells is due to DDB2 and XPC upregulation. First of all, the authors have not convincingly shown that DDB2 and XPC protein levels are upregulated in ZRF1 KO cells. Furthermore, it is likely that the same result will be obtained after knockdown of any NER

protein (even if not upregulated), as UV survival/viability depends on NER. Rather, it would be more convincing if the authors could confirm their hypothesis by showing that overexpressing DDB2 leads to XPC overexpression and enhanced UV survival.

Figure 5C. Five images are shown per condition but it is unclear and not described what these represent? Are these different microscopic images from the same experiment? This experiment should be replicated and quantified.

Supplementary Fig S5C. This western blot should be replicated and quantified before any strong conclusions can be drawn from this result.

Supplementary Fig S5F. The quality of these blots is very poor due to the unequal loading as shown by the loading control. These should be properly replicated.

Supplementary Figure S5E. The authors should show efficiency of DDB2 knockdown and also show whether the transcriptional upregulation of XPC indeed leads to higher XPC protein levels. Supplementary Figure S5F suggest that XPC protein levels are not higher in ZRF1 KO cells, which argues against the proposed hypothesis. Like DDB2, XPC acts in a complex with other proteins (HR23B and Centrin2) and its stability and thus protein levels also depend on availability of these proteins. Therefore, the upregulation of this entire complex should be considered and determined.

Page 5. Please explain the BG4 antibody for clarity.

Western blots should be labeled with size markers.

There are minor grammatical errors that should be corrected.

In the legend of supplementary Fig 2A the '??' should be corrected.

In the section entitled 'ZRF1 activity is evolutionary conserved but with an opposite phenotype compared to Zuo1', the experiments switch from yeast to human which is not always clearly indicated. Please do this to avoid confusion.

Reviewer #3 (Remarks to the Author):

This is yet another excellent study generated in the Paeschke lab that further our understanding in the complex relationship between formation of DNA secondary structures (G4s), genomic instability and senescence, which is highly relevant to cancer biology.

In this study De Magis et al. follow up on their recent discovery of Zuo1 being an important regulator of G4-formation in yeast and validate similar observations in human cells on the human analogue ZRF1. What the authors demonstrate is that upon UV irradiation, ZRF1 is recruited at G4s, which are in turn accumulated as a response of the irradiation. This allows ZRF1 to compete with DDB2 for sensing these structures and trigger cell senescence. As all the reports from the Paeschke's lab, this is a thorough study that includes small molecule treatment to prevent or promote G4s, ChIP-Seq data to validate ZRF1 binding sites and RNA-Seq to investigate differentially expressed genes upon different conditions. Based on the novelty of the findings and the wide variety of evidence reported to support the authors' conclusion, I am fully supportive for publication in Nat Commun.

There are, however, a couple of small points that needs addressing prior acceptance:

1) The authors tend to state that G4s are formed as a consequence of UV irradiation, but I would

disagree with this, especially considering that adding PDS helps UV survival to the cells. I would say that this is in agreement with G4s being accumulated rather than formed by the UV, with CPDs locking DNA in certain conformations that do not allow un-folding of G4s. This will freeze G4s formed making them unable to unfold rather than forming G4s that were not there. So changing the terminology to accumulating G4s rather than forming is important, I believe.

2) In the ChIP-Seq experiments, why only MEME analysis and G4-prediction was used and not relying on experimentally validated datasets generated in Chambers et al Nat Biotech 2015? Probably the analysis won't change at all, but it would be good to validate it fully.

3) When referring to reference 30, cPDS increases BG4 staining in the nucleus, citing the seminal paper where this observation was first reported (Biffi et al Nat Chem 2014) would be more appropriate.

Subject to this, I am supportive of publication in Nat Commun.

We thank all referees for their constructive comments, which we have tried to address in their integrality. We thank the referees for their thoughtful comments which we addressed with multiple new experiments, which we quickly list here: Among the main changes, we have performed colony formation assay to confirm, with the standard assay in the NER field, the vitality after UV irradiation. We have performed the assays in WT as well as ZRF1-KO cells, in WT cells treated with G4 stabilizing compound, PDS and a ZRF1 rescue cell line (ZRF1-KO cells stably integrating ZRF1 plasmid). Dot Blots were repeated with 2 additional time points, 0 and 24 h post UV irradiation and a DNA loading control was used for normalizing the CPD signal. To further support strengthen our findings we have performed CPD immunofluorescence (IF). For all the experiments (colony formation assays, dot blots and IFs), a cell line in which a NER Protein (XPC) was downregulated by siRNA was use as a control. We believe that the newly added analysis and experiments strengthen our conclusion that ZRF1 binds to nuclear G4 structures induced by UV light. We propose a model in which ZRF1 targets G4s structures to prevent via DDB2 expression senescence and genome instability. The main changes in the text and new datasets are highlighted in green. We hope that the referee's will find our revision suitable for publication of this manuscript.

Reviewer #1 (Remarks to the Author):

In this paper, De Magis and co-authors describe the role of the ZRF1 protein in binding UV-induced G4 structures and subsequently supporting the repair process to prevent senescence. This paper follows a previous one in which De Magis and co-authors identified a protein in yeast, Zuo1, as involved in G4 structure formation and in the mechanism of nucleotide excision repair.

The paper is well written, the data are technically sound, well presented, robust and support conclusions. Significance of the paper is high, since to my knowledge this it is the first time a protein is reported to both bind G4 structures and support a repair process in cells.

We thank the referee for his positive assessment on our method and have tried to answer his/her concerns in the best possible way.

I have only some major issues that need to be addressed:

- How was UV irradiation performed? Which wavelengths were used? How were the UV doses chosen? Are they comparable to the sun radiation? Please specify.

We have added the information to the method section (page 15). We treat cells with 254 nm (UVC) wavelength using the Stratagene UV Stratalinker 1800 Crosslinker system. The doses of UV were chosen according to the result of the metabolic activity in a dose-response experiment. 10 Jm^{-2} was used in order to avoid

measuring necrosis, apoptosis. The levels of UV reaching the Earth upper atmosphere (based on the solar irradiance curves) seem to be approx :

UVA ~ 85 J/m²

UVB ~ 15 J/m²

UVC ~ 4 J/m²

- The authors demonstrate that the use of “100 μM PhpC reduced G4s significantly while it had no effect on cell vitality per se”. How can the authors be sure that PhpC does not interfere with BG4 binding, and ZRF1 binding as well? How did they demonstrate it? This point has to be clearly proved.

We agree with the referee that from the presented data it is not clear if PhpC is removing G4 or masking G4s and by this prevent the binding of BG4 or ZRF1. However, in cells it is not possible to determine the differences. Also in vitro, using either EMSA or ELISA (standard methods to determine binding of proteins/peptides to G4s) a clear demonstration that will allow us to distinguish masking vs. removing G4s is not possible. However the results published in Mitteaux et al¹ showed that the ability of the G4-recognizing Pif1 is enhanced by PhpC (Figure S27). This demonstrates that PhpC does not ‘mask’ G4 (meaning, does not bind to G4s and prevent its unfolding by Pif1) but does actually destabilize G4s (as the processivity of Pif1 is improved).

Minor points:

- Please describe in the material and methods section how were the plasmids containing the two isoforms of XRF1 obtained.

We have added this information in the section Yeast Spot Assay of the Material and Methods (page 18)

- Fig. S3 represents “IF staining of WT and ZRF1-KO cells, treated/untreated 48 h with 1 or 2 μM of PDS”: why are there two panels? Please clarify.

We have added this information to the Figure legend of S3.

- In the legend to Fig. 2 please move “orange asterisks...” from letter e) to letter d). Thanks for pointing this out, we have corrected it.

- In the legend to Fig S5d “G4” and “G4 mut” seem to refer to the black triangle, which indicates the amount of recombinant protein: please make the figure clearer.

We agree with the referee and improve the labeling of the figure.

- Please check all references: on p. 5, ref. 32 refers to “Miglietta, G., Russo, M., Duardo, R. C. & Capranico, G. Nucleic Acids Res. 49, 6673–6686 (2021)” and not to PhpC.

We are sorry for this mistake; we have controlled all references and corrected them.

- Please check all over the text for some sentences that seem like comments that were not removed (Legend of Fig. S2a “as predicted by ??”, Legend of Fig. S5 “(The label has moved!)”, legend of Fig. S3 “Why two times??”).

We are sorry for this and have updated the legends

Reviewer #2 (Remarks to the Author):

Previously, the authors had studied yeast Zuo1 function in relation to formation of G4 structures and nucleotide excision repair (NER) activity. In this manuscript, the authors continue/confirm these studies by focusing on the human Zuo1 ortholog Zrf1. They show that stabilization of G4 structures enhances cell vitality after UV irradiation, which they claim is not due to enhanced DNA repair, and that G4 structures also form after UV irradiation. ZRF1 is shown to bind to G4 structures in cells and to be necessary to prevent genome instability. Human ZRF1 also partially rescues UV sensitivity of yeast Zuo1 mutants, suggesting an evolutionary conserved function. Furthermore, ZRF1 KO leads to enhanced vitality after UV and increased CPD levels. The authors report that upregulation of NER genes DDB2 and XPC in ZRF1 KO cells is responsible for the enhanced vitality of ZRF1 KO cells after UV. Finally, they report that UV-induced senescence induction in ZRF1 KO cells is dependent on DDB2.

The manuscript contains many experiments, with some interesting and striking results such as increased viability and elevated CPD levels in ZRF1 KO cells after UV. However, experiments are often not sufficiently convincing or well executed and there are too many issues that need to be addressed before I can reach the same conclusions as the authors do. Below I describe my main issues that I think need to be addressed before this manuscript is suited for publication anywhere.

We thank the referee for his/her time and pointing out that we performed many experiments that led to many interesting and striking results.

On page 5, the authors start by hypothesizing that G4 formation could influence NER because XPB and XPD were reported to bind G4 regions. Therefore, they perform viability MTT assays after UV irradiation, apparently to investigate NER activity upon G4 formation, as shown in Figures 1A and 3F. My concern with this strategy is that standard assays in the NER field are colony formation assays, in which few hundred cells are plated in a well and colony formation is measured. In this type of assay, cells are irradiated without any liquid (that protects against UV) and much lower UV doses (10x) than the authors use are applied. Because the authors do not use the more common method and do not take along NER deficient controls, it is difficult to evaluate how the UV sensitivity of PDS-treated and ZRF1 KO cells compares to that of truly NER deficient cells. Therefore, it is not really possible to draw any definite conclusions about NER activity based on these experiments. Ideally, the authors should include a NER-deficient or -knockdown control cell line in Figs 1A and 3F.

We thank the referee for this comment, and we are sorry for the missing clarity. In the MTT assay cells were irradiated without liquid, we have added this information to the materials and methods section (page 15). In order to fit with the standard assay used in the NER field we have performed different colony formation assay with 10 J m^{-2} , UV dose as previously used in several publications²⁻⁴. Further we have

used the XPC-KD as a positive control for these experiments and added in Supplementary Fig. S1a and S4f.

In Fig 5B the authors do knockdown DDB2 and XPC using this assay, but observe no (DDB2) or only mild (XPC) UV sensitivity. This is strange and based on this poor UV sensitivity, I strongly doubt whether this assay in this way is really useful to determine NER capacity or functionality in response to UV irradiation. NER deficient cells are strongly UV sensitive and should not survive even low doses of UV. It would therefore be more convincing if the authors could show similar results using colony formation assays performed with proper controls.

As suggested by the referee, we have performed colony formation assays in these knockdown cell lines. As expected, colony formation leads to a complete death of the cells treated with siRNA against XPC (Supplementary Fig. S1a and S4f) or DDB2 (data not shown). We explain the difference between MTT and colony formation assay based on the different read out and conclusions. We observed as a novel finding that metabolic rates differ in ZRF1-KO and G4 stabilizing conditions compared to growth changes. In MTT metabolic changes were detected. As part of this paper, we demonstrated that although cells stop dividing ZRF1-KO cells are highly metabolic active after UV, which is a consequence that cells enter senescence. In the current version we discuss in detail how cells stop dividing but increase their metabolic activity.

Moreover, previously, ZRF1 loss in human cells was reported to lead to similar UV sensitivity as XPC loss (Gracheva et al 2016). How do the authors explain their opposite results?

In Gracheva et al they have performed colony formation assay using different UV doses ($2-8 \text{ Jm}^{-2}$) and observed a growth defect. In order to demonstrate that indeed ZRF1-KO lead to a growth defect after UV, we have performed colony in our cell line with our selected UV conditions (10 Jm^{-2}). Similar to published work we observed a complete stop in growth of the ZRF1-KO cells. We have included this information to Supplementary Fig. S4f. The results are in line with the previous publication and further support our work that cells activate without ZRF1 efficiently senescence.

On the bottom of page 8, the authors conclude that ZRF1 binds 'to G4 structures after UV irradiation where it supports repair'. Also, on page 9 the authors hypothesize that 'without ZRF1, NER repair is not functional'. This conclusion seems premature and cannot be drawn from the results presented.

We thank the reviewer for this comment. ChIPseq analysis clearly demonstrate that ZRF1 binds specifically to G4 motif after UV irradiation (Fig 3). Further we showed that ZRF1 localization in the cell is dependent on G4 stabilization (Fig 1E). We agree that the conclusion that without ZRF1, NER repair is not functional was overstating the data. We observed that without ZRF1 cells stop growing after UV and that thymine dimers are accumulating. However, cells are still metabolic active. We improved the wording and discuss these findings in a newly created model in which ZRF1 is important during UV induced damage by targeting G4s which is required to prevent the entry into senescence.

If ZRF1 would support repair, its loss would be expected to result in decrease UV survival, as does the loss of any NER or other protein that supports repair.

As mentioned above, we have performed colony formation assay in our cell line after UV irradiation (10 J m^{-2}) and observed a complete loss in cell growth in the ZRF1-KO cells (Figure S4). These results agree with defect in other NER proteins, and suggest a role of ZRF1 in coping with UV-induced DNA damage.

However, the authors show the opposite, which is that ZRF1 KO cells have increased vitality after UV irradiation. If NER would not be functional, UV irradiation should kill the cells already at very low dose. I understand that the authors may want to make the point that ZRF1 KO cells have increased vitality in spite of having deficient NER, but then they should show that indeed in the absence of functional NER (so knockout of a NER protein) ZRF1 KO cells still have elevated vitality after UV. I strongly doubt whether they will find this based on the XPC knockdown results shown in Fig 4B. These results show that XPC knockdown still leads to lower UV survival, implying that NER must be functional in these cells.

The newly added colony formation assay in WT and ZRF1-KO plus UV (10 J m^{-2}) (Supplementary Fig. S4f), show that similar to defects in other NER proteins (and according to our XPC-KD cell line), a complete loss in growing. However, MTT analysis demonstrate that cells lacking ZRF1 are not dead after UV, as they are highly metabolic active. We conclude that ZRF1-KO activate an alternative pathway to prevent cell death. We believe, as suggested by the reviewer, that in ZRF1-KO, based on the XPC knockdown results shown in Fig 4B, NER still works, but cells are not repairing the damage because they are entering senescence. If ZRF1 has a direct impact on NER activity, we do not know and is not part of this manuscript.

Importantly, the authors also have not correctly measured CPD repair, see my comment below, but observe a seemingly impossible increase in CPD levels after UV in 4 hrs. This cannot be used to conclude anything about repair.

We apologize for this and have added experiments to address this point. See below.

Fig 1D. To be able to accurately assess the outcome of this dot blot, which strikingly shows a huge increase in CPD levels in the presence of PDS, the authors should include a loading control (stain the blot with an anti-DNA antibody or equivalent). Also, the experiment should be replicated (at least three times) and quantified (normalized to the DNA signal). Ideally, a control such as a NER deficient cell line should be included to understand what can be expected in the absence of repair. It is important to understand how much more CPDs are detected in Fig 1D in the PDS condition, as this result is unanticipated and opposite of the enhanced survival shown in Fig. 1A (the same is true for Fig 3E and F). I doubt whether the total amount of potential G4 sites in the human genome (700000?), when stabilized, can account for this increase in CPD lesions, as 10 J m^{-2} will probably already induce over hundred thousand of CPDs in the human genome. The authors should address this and consider that another activity of PDS than G4 stabilization is responsible for the observed increase.

We apologize for the missing loading control from the previous experiments. We have added a SYBR Gold staining (Supplementary Fig. S1d, S4c). All the experiments in the manuscript are performed, at least, in triplicates as reported in the figure legends. We have now included a quantification of 3 biological replicates normalized to the DNA signal (Fig. 1d, 3e). We have also included as a control, a NER deficient cell line (HeLa siRNA XPC).

We determined without XPC 2.2-fold more CPDs were detected after 24h. Similar after PDS treatment, we determined also an increase in CPDs after 24h (2.1-fold). In direct comparison of XPC-KD and PDS treatment, we determined a similar increase in CPDs after 24 h. We conclude that G4 stabilization, as well as a lack in a functional NER machinery leads to a loss in repair and an accumulation of CPDs (Fig 1d).

Regarding the second concern about the amount of G4s induced/stabilized by PDS. In the cell there are nearly 1 Million potential G4 forming regions. Multiple cellular changes occur after PDS addition. PDS itself has a nM affinity to G4s⁵. Previous publications have shown that PDS leads to G4 specific gene expression changes, translational changes as well as changes in genome stability. We observed after G4 stabilization by PDS a 1.6-fold increase in CPDs compared to untreated conditions. We explain that PDS causes the increase in CPDs most likely via two cellular changes. First, by modulating ZRF1 localization to the nucleus. Because our presented work showed that ZRF1 translocates to the nucleus and binds to DNA after UV as well as after G4 stabilization. Second, as our data show that although cells are metabolic active after PDS treatment (Fig. 1a), cell have stopped dividing as indicated by the lack in colony formation (Supplementary Fig. S1a). These data suggest that normal NER repair is not functional most likely due to lack in dividing. Combining these changes, we conclude that G4 stabilization by PDS can induce CPDs via direct . We have included this in our discussion.

Most important is that the assay is not performed with the right recovery time after UV to measure CPD repair. Human CPD repair is slow (~50% repair in 24 hrs; PMID 4010689) and it is therefore necessary to evaluate CPD repair for a period of at least 12 - 24 hrs (which is common in the field). Because the authors measure CPD levels up to only 4 h after UV, also in the untreated condition repair is not visible as weaker CPD signal. The authors should therefore perform this assay correctly, with sufficient recovery time, with loading control, with NER deficient control and with replicates and quantification. The same holds true for Fig 3E (which appears to be performed on two different blots, which should be avoided). Moreover, to substantiate the weird observation that CPD signals are higher after PDS treatment or ZRF loss, the authors should consider to perform similar experiments by CPD immunofluorescence in cells. Also, it would be worthwhile to test if similarly elevated levels are observed for 64PPs, the other abundant type of UV photolesion (which are repaired much faster within a few hours).

We thank the referee for the suggested experiment to strengthen our observations. We performed, as suggested, dot blot analysis at time 0 and 24 h after UV irradiation and monitored CPD levels (Fig. 1d, 3e). We have now included a quantification of 3 biological replicates normalized to the DNA signal and a NER deficient cell line (HeLa siRNA XPC). Results confirm, at time 0 (directly after UV), 1.2-fold increase

for ZRF1-KO (Fig. 3e) and 1.6-fold increase for PDS-treated cells compared to WT (Fig. 1d). At 24 h, 1.2-fold increase was detected for ZRF1-KO (Fig. 3e) and 2.1-fold for PDS-treated cells (Fig. 1d). Further, we performed, as suggested, a CPDs staining by IF (using the staining kit from Kamiya Biotech) and observed similar results (Supplementary Fig. S1e, S4d). Note, in the IF only the nucleus was stained, we did not consider the cell cytoplasm (Supplementary Fig. S1e, S4d).

Due to the minor changes in CPDs between WT and ZRF1-KO at 0 and 24 h time points, we excluded a NER deficiency (largely discussed in the new version of the manuscript).

Fig 1E. The authors should explain more clearly how this assay is used to measure ZRF1 binding to G4 structures. I guess this is based on washing away free ZRF1, to determine if ZRF1 is more stably bound to DNA structures, but the authors state that they quantify 'localization' of ZRF1 which confuses. The authors should also show specificity of the antibody in IF, which could for instance be verified by knocking down ZRF1 or do this assay in ZRF1 KO cells. Also, it would be clearer if not only merged images but also the separate green and blue channel are shown.

We are sorry for not explaining the experiment clearly enough in the previous version of the manuscript. In this experiment we aimed to determine if ZRF1 localization after G4 stabilizing is altered and if ZRF1 locates always to the cellular component that harbors more G4s. We used known chemical ligands that either induces G4 formation in both compartment (nucleus and cytoplasm, PDS) or a ligand that induces only G4s in the cytoplasm (cPDS). The results show that there is an increase in ZRF1 occupancy in both compartments after PDS treatment and only in the cytoplasm after cPDS treatment. Based on these findings we conclude that ZRF1 binds to elevated G4 levels both in the cytoplasm and nucleus, if G4s are enriched. In this assay we just add the antibody to already fixed cells and stain for ZRF1 localization, this assay is not providing any indication if ZRF1 binds to DNA or RNA. To address DNA binding, we performed the ChIPseq analysis to monitor globally ZRF1 binding.

To confirm the specificity of ZRF1 antibody, we have performed the experiments in ZRF1-KO cells and no signal was detected. Due to the absence of signal we decide to not include the empty staining in the manuscript. Further we decided to merged images, in order to avoid a too crowded image. In the Supplemental figures we have included here represented images of IFs. The quantification of 278 (WT), 176 (PDS-treated cells), 266 (cPDS-treated cells) cells is included in Figure 1e and demonstrates the robustness of the analysis.

In the ChIPseq experiments (Fig 3), we determine ZRF1 binding to chromatin regions on the genome wide level. We have performed this in WT (plus and minus UV) and PDS-treated cells. We observed that in untreated cells ZRF1 does not bind to DNA regions, which agrees with an absence of ZRF1 in the nucleus in this condition. However, upon PDS addition, G4 are elevated in both nucleus and cytoplasm, ZRF1 binds significantly to G-rich DNA regions that can form G4s. In the ChIPseq experiments, we perform stringent washing conditions to eliminate unbound antibody and we normalize to a no-tag control. All NGS analysis are done in triplicates.

Supplementary Fig 2D. It is difficult to conclude from these experiments that a 'small but significant enrichment of G4s was detected' as this is not observed with all PDS doses. Rather, the conclusion should be that ZRF1 KO does not really affect G4 levels (at least not, as it seems, in a biological relevant manner).

We thank the reviewer for this observation, and we are sorry that in the previous version we overstated it. We have re-written this part as suggested by the referee (page 7).

Page 7. γ H2AX staining can indeed be used as marker for DSB formation, but H2AX is also phosphorylated when single-strand breaks are formed, as for instance also happens in response to UV irradiation when DNA damage is excised by NER (see e.g PMID 17615256; 19797077). γ H2AX signal after UV irradiation is dependent on NER. Therefore, it is not entirely accurate to state that γ H2AX is a known DSB marker and that the increased micronuclei are independent of DSB formation. Apparently, there is no clearly increased or decreased formation of single- or double-strand breaks in ZRF1 KO cells, which should be the conclusion. Please address this issue.

We apologize for stating it not accurately in the previous version and we have modified our conclusion according to the reviewer suggestion (page 7).

Page 8 and 9. The authors should show and explain which are the genes 'involved in the response to UV-A irradiation' that are differentially expressed in ZRF1 KO cells and what this has to do with NER? The reason I ask this is because NER acts in response to photolesions induced by (mostly) UV-C and UV-B irradiation, but not so much by UV-A irradiation (which would be rather induce oxidative damage and single-strand breaks that require a different repair pathway). It is unclear to me why the authors chose to focus on NER based on this result, also because I can hardly find any NER gene in supplementary Table 1 and NER genes are not typically found to be overexpressed in response to DNA damage induction. Please elaborate on this. Also, it is unclear if these genes are up- or down-regulated.

We apologize and fully agree with the referee. Our previous finding was overstating our conclusion. We agree the presented RNAseq results did not fully support this conclusion. Our RNAseq analysis do not show any upregulation or down-regulation of known repair pathways. Moreover, we identified a highly significant upregulation of DDB2 by RNAseq as well as qPCR analysis using specific primer pairs (Figure S5). The finding that NER is not activated fully agrees with our conclusion that UV lesions are not repaired because mainly cell stop dividing. In subsequent analysis we focused on the question why cells lacking ZRF1 stop dividing/growing but show high metabolic activity after UV radiation. Our work shows that ZRF1-KO leads to an upregulation of DDB2 which is causing the entry into senescence. This senescence induction is the cause why cells are unable to repair the lesions. We have re-written this part and focus our attention on DDB2, which is already known to interact with ZRF1 (page 9, 10, Figure S4, Figure 4,5).

Also we have improved the labeling of the RNAseq analysis in Supplemental Table 1-3. All up-regulated genes are now marked in green and the down-regulated in red.

DDB2 is not an essential NER gene, as suggested by the authors, as is also evident from the lack of effect of siDDB2 in the UV MTT assay of wild type cells shown in Fig 5B. The NER reaction can be reconstituted in vitro without DDB2 and DDB2 is thought to merely stimulate NER of less bulky lesions and in certain chromatin environments. It is therefore questionable whether a transcriptional upregulation of DDB2 by 1.5 fold will majorly affect cell viability after UV through NER activity, also because DDB2 acts in complex with DDB1, Cul4A and RBX1 and its function will depend on the availability of these factors as well. To test their hypothesis, the authors should therefore show upregulation of this entire protein complex (on protein level). Furthermore, the authors hypothesize that 'without ZRF1, NER repair is not functional and CPDs accumulate which further activates DDB2'. However, this is contrary to the generally accepted fact in the field (shown by different labs) that in response to UV irradiation and/or upon accumulation of unrepaired damage (CPDs) DDB2 is auto-ubiquitinated and degraded by the proteasome. The authors find the opposite (no degradation) in WT and ZRF1 KO cells in Fig S5C (again not quantified)/replicated). This figure also shows that in spite of transcriptional upregulation, DDB2 protein levels are not higher in ZRF1 KO cells. Thus, DDB2 does not appear to be expressed at higher levels in unchallenged ZRF1 KO cells, which argues against the proposed hypothesis. These issues should all be addressed and possibly/likely different conclusions should be drawn.

We agree that DDB2 acts in complex, for example with DDB1. DDB2 binding to DNA is favored by a decrease in DDB1 levels. In our manuscript we observe enhanced DDB2 and decreased DDB1 gene expression levels (Supplementary Fig. S5a), which are expected to favor DDB2 targeting to DNA. We demonstrated that DDB2/DD1 complex can bind to G4s in vitro. In cells we demonstrate that the upregulation of DDB2 itself is causing the entry into senescence. We speculate that high levels of DDB2, as observed in ZRF1-KO, target G4s in living cells. And that this targeting drives cells into senescence.

As most other NER proteins do not show altered gene expression by RNAseq, we assume that NER is functional. However, due to the entry into senescence NER can not take place. Western Blot for RBX1 confirm that this conclusion no changes in protein levels of RBX1 were detectable. We agree the lack in ZRF1 is not directly impacting the activity of NER, its rather prevents NER because the lack in ZRF1 drives, via DDB2 upregulation, cells into senescence. We have re-written this part of the manuscript (page 9, 10).

Introduction, page 4. XPC is not an endonuclease. Please correct this error. Also, XPC does not act alone but as part of a heterotrimeric complex. Please accurately describe this. XPC is ubiquitylated by a complex consisting of DDB2, DDB1 and CUL4A and RBX1. Please also accurately describe this. This ubiquitination, however, may not be an 'essential step for NER initiation', as in the absence of DDB2, XPC is still able, albeit less efficient, to detect many types of DNA damage. Importantly, this DDB2-mediated ubiquitylation is generally accepted in the field and has been confirmed and studied by multiple labs, both in vitro, in cells and in vivo. This is different for the ubiquitination in which ZRF1 is implicated. Involvement of

ZRF1 is not (yet) generally accepted in the field and so far only published by one lab (which, I think, does not exist anymore). The authors should therefore describe the proposed/potential involvement of ZRF1 (which involves also another E3 ubiquitin ligase complex) with more care and caution. ZRF1 cannot be claimed to be 'part of the NER complex'.

We apologize deeply for not stating correctly the role and function of XPC. We agree that our experiments do not show a direct modulation of the NER machinery. In the new version of the manuscript, we focused, as stated above, on the finding that in ZRF1-KO enhanced metabolic activity but no growth after UV was observed and that this can be rescued by down-regulation of DDB2. Our conclusion focuses on the impact of ZRF1 binding to G4s after UV to prevent DDB2 induced senescence.

Fig. 1B. Is there a statistically significant reduction in vitality with 100 μ M PhPC? Why is this experiment (together with the one shown in supplementary Fig 1B) not depicted the same way as in Fig 1A (why only one dose now?). Why is the survival of untreated sample not shown in the same graph? This is confusing.

We agree that the previous illustration was not clear, we have included the untreated sample in the same graph (Fig. 1b) and removed Supplementary Fig.S1b. Regarding the significance, it is ($p < 0.01$), we have added the information in the graph and added the information to the figure legend.

Fig 2D. It is unclear what is shown in the graph. The legend says fraction but the graph depicts percentage micronuclei per cell. What is a percentage micronuclei per cell? Also, how many cells were quantified per experiment and are the differences statistically significant? Same for supplementary Fig 3A-B. Is the untreated condition in all these graphs from the same experiments? Please indicate this, or show this only in one of the graphs. Also, the text indicates that a range of range: 1-5 μ M PDS is used, but only up to 2 μ M is shown in these figures. Also, what is the difference between these experiments because the legends of all three figures indicate that this was 48 h of PDS exposure, while according to the text the exposure was for 24, 48 and 72 h. It is annoying that the authors have not been accurate in preparing the manuscript. Also, the legend still has the text: 'Why two times??'.

We truly apologize that in the initial manuscript we left the impression that we did not prepare the manuscript with great care. We are sorry for this, we have controlled the text.

Percentage of micronuclei per cells is obtained by counting the number of Micronuclei and dividing for the number or total nuclei (treatment with less events: 117 nuclei and sample with higher number of events: 1212) and transformed in percentage.

Supplementary Fig 5D. This experiment is poorly explained both in the text and the methods. How were DDB2 and DDB1 purified as this is nowhere shown? Also, the legend and figure are confusing. The legend suggests increasing concentrations of DDB1-DDB2 were used, but the figure suggests as if the G4 DNA dosage was increased. Also, the label was moved and this is still stated as a comment in the legend: 'The label has moved!'. Again, this is an example of inaccuracy in preparing

the manuscript (and not the only example). Also, it is unclear why the experiment is performed. Do the authors wish to show that DDB1-DDB2 can bind to G4 structures and can therefore also do this in cells? If so, why don't they show that DDB2 binds to G4 structures in cells as they did also for ZRF1? And why not show whether this depends or happens in concert with ZRF1 in cells?

We truly apologize for the missing clarity. DDB2 and DDB1 were obtained directly as heterodimer from Thomä Lab (we have added this information in the Materials and Methods section (page 19), further the Thomä lab is mentioned in the acknowledgement). As suggested by the reviewer, we explain the rationale by we performed the binding assay of DDB1-DDB2 to G4s. Originally, we aimed to monitor DDB2 binding by ChIP-seq (similar to ZRF1), however ChIPseq gave very poor results which was due to the quality of the commercial available antibodies. In the discussion we state that we predict that DDB2 binds without ZRF1 to G4s which drives the entry into senescence most likely by promoting gene expression changes. It has been shown already that G4 stabilization can induce senescence that multiple genes linked to senescence like CDKN2A⁶ have a G4 in their promoter or coding regions.

Page 10, top. The authors write that 'We monitored G4 levels by IF both in WT and ZRF1-KO cells after depletion of DDB2 and after UV irradiation (0-8 h after treatment; Fig. 4c,d).' However, it is unclear what is shown in Figure 4c and d. Is this the siDDB2 condition? This is not described in the legend. But if so, where is the siCtrl to which this is compared? It is not possible for me to judge this result or the conclusions if I don't know what I am looking at. Also, in Fig 4D it is unclear and not described to which statistic comparison is indicated by the asterisks. It would be much cleared if the authors show all statistic similarly, i.e. with connecting lines, so that the reader knows which results are compared.

We have added the information to the figure legend and labeled the Figure accordingly. The control cell line is a cell line transfected with a scramble siRNA. In order that the reader knows which results are compared we have added an explanation in the figure legends "Significance compared to untreated-WT cells is indicated by asterisks, connecting lines are used when the significance was compared to other samples."

Page 7, bottom. Please explain the difference between both ZRF1 isoforms (in text or a figure) and also provide some information on the evolutionary conservation between Zuo1 and ZRF1 with regard to amino acid sequence and/or protein motifs. If only the second isoform rescues the UV sensitivity, what makes this isoform different from the first isoform that could explain this difference. Are both isoforms equally expressed, also given the fact that their overexpression causes UV sensitivity in the wild type strain? Also, is the second isoform then more closely related to the yeast Zuo1? Please elaborate on this in the text.

We added, as the referee suggested more information on ZRF1 isoforms and evolutionary conservation (page 8). The difference between isoform 1 and isoform 2 is 445 bp. Isoform 2 lacks two alternate in-frame exons, compared to isoform 1,

resulting in a shorter protein compared to isoform 1. The difference between the human ZRF1 and the yeast Zuo1 is restricted to an overhang C-terminus where are localized the SANT domains (as show in the figure, edited from⁷).

Based on this knowledge the isoform 1 of ZRF1 is more similar to Zuo1 and it is hard to explain from a biochemical point why the two isoforms cannot both rescue the yeast defects.

Page 9, bottom. I do not agree with the authors that the survival with DDB2 and XPC siRNAs shows that the enhanced vitality of ZRF1 KO cells is due to DDB2 and XPC upregulation. First of all, the authors have not convincingly shown that DDB2 and XPC protein levels are upregulated in ZRF1 KO cells. Furthermore, it is likely that the same result will be obtained after knockdown of any NER protein (even if not upregulated), as UV survival/viability depends on NER. Rather, it would be more convincing if the authors could confirm their hypothesis by showing that overexpressing DDB2 leads to XPC overexpression and enhanced UV survival.

To address this valid comment we have, as stated above, performed colony formation assays. The results of the colony formation assay show a lack in growth in ZRF1-KO cells after UV. After downregulation of DDB2 by siRNA, XPC levels were also reduced in qPCR (Supplementary Fig. S5d) as well as Centrin 2 in western blot (Figure below). These results underline that DDB2 misregulation causes XPC-complex level changes (as also supported by other publications). Further, we have created a cell line inducing DDB2-overexpressing in cells. Colony formation assay in DDB2 overexpressing cells resulted into no changes in growth after 10 Jm⁻² UV (data not shown). This result was expected as also WT cells are not really sensitive to UV. Further, western blot analysis display that overexpression of DDB2 is not leading to enhanced XPC and Centrin2 levels (data not shown). This could be explained by a sufficient XPC complex activation in WT cells.

Figure 5C. Five images are shown per condition but it is unclear and not described what these represent? Are these different microscopic images from the same experiment? This experiment should be replicated and quantified.

We are sorry for the missing clarity in this figure. All the experiments in the manuscript are performed, at least, in triplicates as reported in the figure legends. We have now included only 3 images, one per each biological replicate. We have also quantified and added a graph with the quantification in the figure. For all quantifications we have at least counted 1615 cells. Significance was determined using an ordinary one-way ANOVA multiple comparison.

Supplementary Fig S5C. This western blot should be replicated and quantified before any strong conclusions can be drawn from this result.

All the experiments in the manuscript were performed, at least, in triplicates as reported in the figure legends. We have also quantified and added a graph with the quantification in the figure.

Supplementary Fig S5F. The quality of these blots is very poor due to the unequal loading as shown by the loading control. These should be properly replicated.

We are sorry for the poor quality of the previous blots; we replaced this with a blot with a greater quality and as well as a quantification. The mean \pm SD are presented in the figure. All the experiments are performed, at least, in triplicates as reported in the figure legends.

Supplementary Figure S5E. The authors should show efficiency of DDB2 knockdown and also show whether the transcriptional upregulation of XPC indeed leads to higher XPC protein levels. Supplementary Figure S5F suggest that XPC protein levels are not higher in ZRF1 KO cells, which argues against the proposed hypothesis. Like DDB2, XPC acts in a complex with other proteins (HR23B and Centrin2) and its stability and thus protein levels also depend on availability of these proteins. Therefore, the upregulation of this entire complex should be considered and determined.

To address this comment, we have included a Western blot showing DDB2 levels after siRNA treatment. Our current manuscript focuses on the consequences of ZRF1-KO after UV to characterize why a lack in UV repair was detected but an increased metabolic rate. We observed no dramatic changes of DDB2 or XPC in ZRF-1-KO by western blot. Also Centrin2 is not different in ZRF1-KO by Western. Because the gene expression is only 1.5-fold different it could be that Western Blot are not sensitive enough to detect minor changes.

Page 5. Please explain the BG4 antibody for clarity.

BG4 is a highly specific antibody fragment that binds specifically to G4 DNA structures. The antibody was originally identified, validated and characterized by the Balasubramanian lab⁸. Since this publication the antibody is now commercially

available and is widely used in the field. Prior each experiment we validate the antibody by ELISA and control Ifs (known conditions that either enhance or deplete G4 formation), for the specificity against G4s. The affinity of BG4 is in the nM range. We also perform control stainings to detect the background level, in those the G4 specific antibody BG4 is missing in the stainings. As suggested by the referee we included more information regarding BG4, its specificity to G4s and the usability of the antibody to the manuscript.

Western blots should be labeled with size markers.

As suggested we included them in all Western blots

There are minor grammatical errors that should be corrected.

Thanks for spotting these mistakes they are corrected.

In the legend of supplementary Fig 2A the '??' should be corrected.

We have controlled all Figures and legend and corrected them

In the section entitled 'ZRF1 activity is evolutionary conserved but with an opposite phenotype compared to Zuo1', the experiments switch from yeast to human which is not always clearly indicated. Please do this to avoid confusion.

We thank the referee for indicating this problem. We have rewritten parts of the introduction to clarify this point (page 8). We hope the referee will be happy with our rewriting and all our editing of the manuscript.

Reviewer #3 (Remarks to the Author):

This is yet another excellent study generated in the Paeschke lab that further our understanding in the complex relationship between formation of DNA secondary structures (G4s), genomic instability and senescence, which is highly relevant to cancer biology.

In this study De Magis et al. follow up on their recent discovery of Zuo1 being an important regulator of G4-formation in yeast and validate similar observations in human cells on the human analogue ZRF1. What the authors demonstrate is that upon UV irradiation, ZRF1 is recruited at G4s, which are in turn accumulated as a response of the irradiation. This allows ZRF1 to compete with DDB2 for sensing these structures and trigger cell senescence. As all the reports from the Paeschke's lab, this is a thorough study that includes small molecule treatment to prevent or promote G4s, ChIP-Seq data to validate ZRF1 binding sites and RNA-Seq to investigate differentially expressed genes upon different conditions. Based on the novelty of the findings and the wide variety of evidence reported to support the authors' conclusion, I am fully supportive for publication in Nat Commun.

We thank the referee for his/her time and for the positive feedback and very nice words on our manuscript.

There are, however, a couple of small points that needs addressing prior acceptance:

1) The authors tend to state that G4s are formed as a consequence of UV irradiation, but I would disagree with this, especially considering that adding PDS helps UV survival to the cells. I would say that this in agreement with G4s being accumulated rather than formed by the UV, with CPDs locking DNA in certain conformations that do not allow un-folding of G4s. This will freeze G4s formed making unable to unfold rather than forming G4s that were not there. So changing the terminology to accumulating G4s rather than forming is important, I believe.

We thank the reviewer for this observation, and we are sorry that in the previous version we used an inappropriate terminology. We have change G4 formation in G4 accumulation.

2) In the ChIP-Seq experiments, why only MEME analysis and G4-prediction was used and not relying on experimentally validated datasets generated in Chambers et al Nat Biotech 2015? Probably the analysis won't change at all, but it would be good to validate it fully.

We agree and add as Supplementary Fig. S2a a genomic overlap of the forward and reverse strands G-quadruplexes predicted by Chambers et al. As stated in the figure S2a, ChIPseq peaks significantly overlap to G4 peaks ($p < 0.001$) published by Chambers et al.

3) When referring to reference 30, cPDS increases BG4 staining in the nucleus, citing the seminal paper where this observation was firstly reported (Biffi et al Nat Chem 2014) would be more appropriate.

We agree and changed the reference.

Subject to this, I am supportive of publication in Nat Commun.

We thank the referee for supporting our work.

References

1. Mitteau, J. *et al.* Identifying G-Quadruplex-DNA-Disrupting Small Molecules. *J. Am. Chem. Soc.* **143**, 12567–12577 (2021).
2. Garinis, G. A. *et al.* Transcriptome analysis reveals cyclobutane pyrimidine dimers as a major source of UV-induced DNA breaks. *EMBO J.* **24**, 3952–3962 (2005).
3. Guha, A., Nag, S. & Ray, P. S. Negative feedback regulation by HuR controls TRIM21

- expression and function in response to UV radiation. *Sci. Rep.* **10**, 1–14 (2020).
4. Gracheva, E. *et al.* ZRF1 mediates remodeling of E3 ligases at DNA lesion sites during nucleotide excision repair. *J. Cell Biol.* (2016). doi:10.1083/jcb.201506099
 5. Müller, S., Kumari, S., Rodriguez, R. & Balasubramanian, S. Small-molecule-mediated G-quadruplex isolation from human cells. *Nat. Chem.* (2010). doi:10.1038/nchem.842
 6. Beltran, M. *et al.* G-tract RNA removes Polycomb repressive complex 2 from genes. *Nat. Struct. Mol. Biol.* **26**, 899–909 (2019).
 7. Craig, E. A. & Marszalek, J. How Do J-Proteins Get Hsp70 to Do So Many Different Things? *Trends Biochem. Sci.* **42**, 355–368 (2017).
 8. Biffi, G., Tannahill, D., McCafferty, J. & Balasubramanian, S. Quantitative visualization of DNA G-quadruplex structures in human cells. *Nat. Chem.* **5**, 182–186 (2013).

REVIEWER COMMENTS

Reviewer #1 (Remarks to the Author):

The authors have addressed most issues and have added new experiments, improving data strength. For these reasons, I recommend the paper to be published in Nature Communications

Reviewer #2 (Remarks to the Author):

In this revised manuscript, the authors have made quite some effort to address my previous concerns. They have added colony survival experiments, which now indeed show that NER- and ZRF1-deficient cells die/arrest after (the relatively high dose of) UV. Also, they have improved and repeated their CPD dot blot experiments, although I think that these are still not useful to show whether repair takes place or not (see concern below). The authors finally come up with an (compared to the previous manuscript) adjusted conclusion/hypothesis, which is that UV irradiation cause G4 DNA accumulation, leading to recruitment of ZRF1 to chromatin to prevent DDB2-mediated senescence. Evidence is presented for parts of this conclusion, but overall, I am still not convinced that the results fully support this interesting idea. In the abstract, the authors conclude that 'loss of ZRF1 as well as high G4 levels lead to overexpression of DDB2' and 'this overexpression of DDB2 drives cells into senescence after UV irradiation'. However, they fail to convincingly show by western blot that DDB2 is overexpressed, which they also admit in the rebuttal letter when they write 'We observed no dramatic changes of DDB2 or XPC in ZRF1-KO by western blot'. Importantly, the mechanism that would connect DDB2, a known DNA repair protein, to senescence induction is unclear and not addressed. Multiple experimental and other issues still remain that should be addressed properly. The manuscript also still contains multiple errors.

My main concerns with this revised version are:

On page 6, the authors perform dot blot analysis of CPD formation and repair to 'address how G4 formation influences cellular pathways after UV irradiation'. I guess with 'cellular pathways' the NER pathway is meant? In answer to my concern in the previous manuscript, CPD repair has now been analyzed 24 h after UV irradiation. However, the experiment can still not be used to evaluate whether repair is affected by PDS treatment or not, because no CPD repair is observed in the untreated control cells 24 h after UV irradiation. Therefore, the authors cannot conclude from this experiment that UV damage is not properly repaired if G4 is stabilized. Also, the authors state that CPD levels are 1.6-fold higher after PDS treatment, but this rather looks like 1.2-fold in the graph in Figure 1D. Also, no individual data points are plotted for the PDS experiments and no statistics are shown.

The authors then write that IF analysis confirmed their observation, but again refer to the same dot blot graph in Fig 1D, while they mean the experiment shown in Supplemental Figure S1e. This experiment indeed suggests that CPD levels are higher after PDS treatment. However, the problem with this experiment is also that no repair of CPDs is found in the untreated control cells. This should be a red flag that something is wrong with the experiment, as the CPD signal should diminish in 24 hours. Also, the legend states that siRNA against XPC was used, but this is not shown. Also, the DAPI staining is not shown. Also, this experiment lacks proper unirradiated control cells. The authors could consider alternative methods to measure NER, such as IF or dot blot for 64PPs or for instance unscheduled DNA synthesis (PMID 25474029).

Line 252. The authors state that enhanced G4 levels by PDS treatment lead to enhanced CPD formation and refer to figure 1D. This is, however, not at all clear from this figure. No statistics or individual data points are plotted for the PDS treated cells and it appears that CPD levels after PDS treatment are not significantly different from untreated cells. Please address this in the text and figure.

Figure 3E. No individual data points are plotted for the ZRF KO cells and not statistics are shown. The authors conclude in line 254 that a moderate increase in CPD levels is displayed, but I suspect that the correct conclusion should be (based on this experiment) that there is no significant difference in CPD levels between wild type and ZRF1 KO cells. As no CPD repair is observed in wild type cells (see concern above, as in Figure 1D, which is, I guess, the same experiment), this experiment cannot be used to tell whether there is repair or not in ZRF1 KO cells (see also conclusion in lines 374-375).

Line 269. The authors conclude that ZRF1 supports DNA repair, but there is no direct evidence shown to support this conclusion. The only evidence shown that ZRF1 might affect DNA repair is the increased genomic instability observed in the form of more micronuclei and the lack of colony formation after UV. This is indeed evidence that ZRF1 might affect DNA repair, but indirect and there can be other reasons for these results in the absence of ZRF1 KO. If the authors wish to draw the conclusion that ZRF1 supports DNA repair, they should provide evidence that DNA repair (which pathway?) is affected. Otherwise, they should more carefully state their conclusions.

The qPCR experiment in Figure S5A suggests that DDB2 mRNA levels are higher in ZRF1 KO cells, but the immunoblot in S5B shows that protein levels (in unirradiated cells) are similar to wild type. Thus, without UV-irradiation, there is not more DDB2 protein in the cell. This is (purposely?) ignored by the authors. How can DDB2, if its protein levels are unchanged, specifically affect XPC mRNA levels in ZRF1 KO cells, as suggested by supplementary Fig S5D? In lines 300 and 309, the authors suggest that XPC is upregulated due to elevated DDB2 expression. If this is true, this must imply a mechanism by which more DDB2 mRNA (but not more protein) leads to more XPC mRNA? How would such a mechanism work and why don't the authors address this? Please address this weird issue, which suggests that more DDB2 mRNA levels regulate XPC transcription depending on ZRF1?

In addition, to support their claim it would be more than logic to show that not only XPC mRNA levels but also XPC protein levels are affected, if the authors wish to conclude that XPC protein levels have any relevance to the observed phenotypes. This could be done in similar experiments as shown in Supplementary Figure S5E where the authors use siRNA against DDB2 and XPC and stain for DDB2 and XPC protein levels. However, they fail to show whether siRNA against DDB2 then indeed affects XPC protein levels in ZRF1 KO cells. This would have been a very obvious experiment, especially since the authors show in the rebuttal letter that they have tested protein levels of the XPC-interacting protein Centrin2. Based on this they claim that 'DDB2 misregulation causes XPC-complex level changes'. However, they also state in the rebuttal letter that 'we observed no dramatic changes of DDB2 and XPC in ZRF-KO by western blot.', indicating that the western blots (for XPC) were done. I think that these western blots should be part of the paper and the fact that DDB2 and XPC protein levels are (largely?) unchanged in ZRF1 KO cells should definitely be considered in the conclusions and discussion of the model.

In the rebuttal letter, the authors show that siRNA against DDB2 leads a substantial downregulation of the XPC-interactor Centrin2. This could indeed be a mechanism by which then also XPC becomes unstable. However, this mechanism would affect XPC on the protein level and not its transcription (on the mRNA level) as shown in supplemental Fig S5D, so this would constitute a different mechanism. However, Centrin2 is expressed in large excess over XPC and the majority of cellular Centrin2 is not in complex with XPC (PMID 15964821). Therefore, downregulation of Centrin2 does not necessarily have to affect XPC stability and this should be shown by staining for XPC itself. The effect of siDDB2 on Centrin2 is striking. I realize that this may be beyond the scope of this manuscript, but it would be wise to test if this downregulation is an off-target effect of the siRNA. Centrin2 is an important component of the centrosome, essential for proper mitosis, and its downregulation by siRNA against DDB2 may therefore be responsible for some of the phenotypes observed in this manuscript.

Supplementary Figure S5B. The quantification suggests that DDB2 protein levels instantly increase 1.5-fold at time point 0 h, which is immediately after UV irradiation. How can this be? Such an instantaneous rise in protein levels, if indeed true, is likely not by transcriptional regulation, but by

regulation of protein stability, and therefore constitute a different mechanism than the transcriptional regulation suggested by the qPCR experiment in supplementary Figure S5A? This must be addressed and not ignored by the authors. But also in this experiment, no statistical tests are shown, so it is not possible to draw the conclusion in lines 292/293 that DDB2 protein levels are upregulated after UV irradiation. DDB2 is well known to be auto-ubiquitylated and degraded after UV irradiation and there is a large amount of literature on this subject. It may appear that ZRF1 KO prevents this DDB2 protein degradation after UV irradiation, rather than affecting its transcription, but to definitely draw this conclusion, more and better controlled experiments are needed. This is something the authors should consider.

The authors observe for cells treated with PDS or with ZRF1 KO a decreased cellular 'growth', as measured with colony forming assay, but enhanced metabolic activity, as measured with MTT assays. It is unclear how these two opposites can be unified and explained. For instance, in line 275, the authors write that the ZRF1 KO cells stop dividing (based on the colony forming assay?) but that their metabolic rate increases after UV irradiation. The difficulty with this conclusion based on the MTT assay is that it is not clear or known if 24 hours after UV irradiation the same number of cells is present in the wells that are compared. The colony formation assay strongly suggests that 24 hours after 10 J/m² UV irradiation, many more wild type cells than ZRF1 KO cells will be present when metabolic rate is measured by MTT absorbance. If the authors wish to compare metabolic rate, I think they should be sure to use the same number of cells at the moment of measurement. This is even more important for the MTT experiments shown in Figure 4A. After UV irradiation, many cells that were treated with siRNA against DDB2 and XPC will stop dividing and/or die (as shown for XPC in the colony formation assay in supplemental Fig S1A). Thus, there will be less of these cells present in the MTT assay and consequently also less total metabolic activity measured. Therefore, this cannot be used to draw the conclusion that (upregulation of) DDB2 or XPC affects metabolic activity.

Line 298. Where is the RNA-seq data shown that XPC is also upregulated in ZRF1 KO cells. I cannot find XPC this in the supplementary Table S1. If XPC, like DDB2, is upregulated, and interacts with ZRF1 (line 298), why did the authors only identify DDB2 in lines 283/284 after an 'in depth analysis of the NER pathway'? Why do they not mention XPC earlier?

Figure 1D, 3E, Supplementary figure S1D and S4C. The dot blot and SYBR gold staining images for untreated and siRNA XPC-treated cells are the same, because this is the same experiment shown in both figures (as indicated by the full dot blots shown in supplementary Fig S7). This should be more clearly indicated in the legends and the figures in 1D, S1D should show not be presented as one single dot blot, as these are rearranged, composite images of the dot blot in Fig S7. Even better, the results of untreated cells, PDS-treated, ZRF-KO and XPC siRNA-treated cells could all be shown in the same figure.

It appears that the colony formation experiments shown in Fig S1A and Fig S4F are also part of the same experiment. Also, this should be clearly indicated, or, better, part of the same figure for clarity.

Line 311. The authors conclude that downregulation of DDB2 did not affect G4 levels significantly. However, it is unclear to which data this is compared. The authors only shown cells treated with siRNA against DDB2 in Figure 4B and C and no control siRNA (performed in the same experiment?) is shown. Similarly, for ZRF1 KO cells, if the authors wish to show that DDB2 downregulation prevents G4 accumulation, they should compare this in the same experiment to ZRF1 KO cells treated with control siRNA.

Lines 312-318. The authors state that in ZRF1 KO cells, DDB2 positively contributes to G4 accumulation after UV irradiation and support this idea by showing that the DDB1-DDB2 complex binds to G4 DNA in vitro. It would have been more convincing if the authors would actually show that DDB2 binds to G4 structures in cells. In the rebuttal, they indicate that they tried this with ChIP experiments, which unfortunately did not work. However, there are multiple other ways to show

enhanced binding to chromatin/DNA and for DDB2 there are multiple papers using techniques like live cell imaging, IF and/or cell fractionation that show its binding to chromatin in cells after UV irradiation. This could be considered by the authors. Please provide a reference for purification of the DDB1-DDB2 complex obtained from the Thomä lab, otherwise its purification and quality control should be described.

Supplementary Figures S2C and S6A. It is unclear how these assays were performed and I cannot find a method where this is described. How many cells were used and how were these assays set up? This is especially important as the result with ZRF1 KO cells after UV appears to differ between the colony formation experiment shown in Figure S4F, where all cells appear to die/arrest one day after UV irradiation and the experiment shown in Figure S6A where 60% of the cells survival one day after UV irradiation. How do the authors explain this discrepancy? Please address this.

Lines 350-352. The authors state that flow cytometry confirmed that ZRF1 KO cells remained in the G0/G1 phase five days after UV irradiation and refer to the results shown in supplementary Figure S6B. Based on these results, it is difficult to draw this conclusion. Basically, the cell cycle distribution in ZRF1 KO cells is the same before UV (Figure 5B) as five days after UV (supplementary Figure S6B). If there is any difference, there are less cells in G0/G1 and more in S/G2. This does not necessarily mean that more cells 'remain' in G0/G1 or are 'quiescent', especially not after UV, as this type of experiment cannot distinguish between slow proliferation, transient cell cycle arrest or quiescence. Cell proliferation is apparently slower, as already shown in Supplemental Fig S2C (which the authors mention in line 199 is only a 'minor change in growth').

Minor concerns:

Supplemental Fig S1A. This is a poor example of a proper colony formation assay, as the number of colonies in unirradiated condition differs widely between cell lines. All conditions should start with the same number of colonies, but here there are much less cells treated with siRNA against XPC compared to control cells, making it difficult to draw definite conclusions (even though this outcome is as expected).

Figure 2D and supplementary Figure 3. I find the term 'micronuclei per cell' still confusing. Apparently, the authors mean micronucleus frequency or simply percentage micronuclei (but not per cell). The authors call this experiment 'IF staining', but no antibodies ('immuno') are used. This is just DAPI staining.

Supplementary Figure S4D. This experiment lacks unirradiated cells as control, to test if the CPD signal observed is specific. Also, the legend states that siRNA against XPC is used, but this is not shown. Also, the DAPI staining is not shown.

Line 60. Different kinds of DNA damage are summarized but 'UV irradiation' is not a kind of DNA damage.

Line 90. DDB1 and DDB2 together are not an E3 ubiquitin ligase. The complex also comprises RBX1 and Cul4, which are the actual E3 ligase components that are targeted by DDB2 to certain types of DNA damage. In line 177, the authors refer to this same complex with UV-DDB-CUL4A, which is inconsistent. Only, in line 388 do the authors correctly mention all the proteins of this complex, but this should of course already been introduced in the introduction. Nowadays, this complex is usually referred to as CRL4-DDB2, as many other types of CRL4 complexes exist, each with a different substrate-targeting protein like DDB2. For instance, in transcription-coupled NER, another CRL4 complex acts, referred to as CRL4-CSA.

Line 89. XPC ubiquitination is not an essential step for NER, as in the absence of ubiquitylation, repair can still take place although less efficient.

Line 96. What is meant with 'the NER complex'?

Line 176. Which 'E3 ligase at lesion sites' is meant. Please be accurate.

Line 220/221. SSB and DSB should be SSBs and DSBs

Line 285 significantly should be significantly

Line 287. It is unclear why (a priori) the authors would assume that an upregulation of DDB2 leads to an increase in metabolic activity. What does DDB2 have to do with metabolic activity?

Line 306 'were' should be 'was'

Line 310. 'Weather' should be 'Whether'. Also, this is not a complete correct English sentence.

Line 311. Fig 5c,d should be Fig 4c,d

Line 323 NER repair is repetitive. The 'R' in NER already means repair.

Lines 340/341 is a repetition of what is already mentioned in lines 331-333.

Line 331 'were' should be 'was'.

Line 332 states 'p16^{INK4a}' but line 338 and the figure states 'p16^{INK4a}'

Line 393. Please explain how gene expression changes that drive cells into senescence can cause lack in repair? This is not logical, as these are not DNA repair proteins.

Line 398. The authors state that no proteins that control or protect genome stability by binding to G4s during repair processes have been identified. However, the DNA repair helicase FANCD1 is well known to recognize, bind and resolve G4 structures to prevent genome instability. Is this not what the authors mean? Please address this.

Line 487. It is not explained in this method when cells were irradiated with UV. I guess this is after plating of the siRNA treated cells (which is common)

I am not particularly enthusiastic about the quality of some of the western blots. I doubt how accurate a western blot such as shown in supplemental fig S5B can be quantified. The bands are not straight, of equal size, and there are bubbles in the bands. I leave it up to the editor to decide if this is acceptable.

There are grammatical errors and typos in the manuscript. Please correct these.

Reviewer #3 (Remarks to the Author):

The authors have addressed my comments in full and I am supportive for publication of the revised manuscript.

We thank all reviewers for their positive feedback and taking time to read and evaluate our manuscript again. In response to the reviewers' comments, we have performed different new experiments and changes to the manuscript that are outlined below in the point-to-point answers. In particular we have corrected inconsistent labeling of the NER components in agreement with the current literature. Further we included additional stainings and western blot analysis to confirm siRNA treatment of NER components and changes in NER after G4 stainings (e.g., 6-4PPs immunofluorescence analysis). Furthermore, by using a pull-down approach, we have confirmed that in living cells DDB2 is binding to G4 structures both before and after UV irradiation. All this data is either added to the manuscript or are part of the data source file, which is also uploaded as part of this submission. All major changes are highlighted in green in the manuscript. We thank the reviewers for this constructive review process. All suggested experiments shaped and strengthen the conclusion of our manuscript.

Reviewer #1 (Remarks to the Author):

The authors have addressed most issues and have added new experiments, improving data strength.

For these reasons, I recommend the paper to be published in Nature Communications

We thank Reviewer #1 for his positive feedback.

Reviewer #2 (Remarks to the Author):

In this revised manuscript, the authors have made quite some effort to address my previous concerns. They have added colony survival experiments, which now indeed show that NER- and ZRF1-deficient cells die/arrest after (the relatively high dose of) UV. Also, they have improved and repeated their CPD dot blot experiments, although I think that these are still not useful to show whether repair takes place or not (see concern below). The authors finally come up with an (compared to the previous manuscript) adjusted conclusion/hypothesis, which is that UV irradiation cause G4 DNA accumulation, leading to recruitment of ZRF1 to chromatin to prevent DDB2-mediated senescence. Evidence is presented for parts of this conclusion, but overall, I am still not convinced that the results fully support this interesting idea. In the abstract, the authors conclude that 'loss of ZRF1 as well as high G4 levels lead to overexpression of DDB2' and 'this overexpression of DDB2 drives cells into senescence after UV irradiation'. **However, they fail to convincingly show by western blot that DDB2 is overexpressed**, which they also admit in the rebuttal letter when they write 'We observed no dramatic changes of DDB2 or XPC in ZRF-1-KO by western blot'. Importantly, the mechanism that would connect DDB2, a known DNA repair protein, to senescence induction is unclear and not addressed. Multiple experimental and other issues still remain that should be addressed properly. The manuscript also still contains multiple errors.

We thank reviewer #2 for carefully reading our manuscript again and for acknowledging the amount of work that we included in the revised manuscript. In the new revised version, we believe we have addressed all comments and concerns of reviewer #2.

My main concerns with this revised version are:

On page 6, the authors perform dot blot analysis of CPD formation and repair to ‘address how G4 formation influences cellular pathways after UV irradiation’. I guess with ‘cellular pathways’ the NER pathway is meant? In answer to my concern in the previous manuscript, CPD repair has now been analyzed 24 h after UV irradiation. However, the experiment can still not be used to evaluate whether repair is affected by PDS treatment or not, because no CPD repair is observed in the untreated control cells 24 h after UV irradiation. Therefore, the authors cannot conclude from this experiment that UV damage is not properly repaired if G4 is stabilized. Also, the authors state that CPD levels are 1.6-fold higher after PDS treatment, but this rather looks like 1.2-fold in the graph in Figure 1D. Also, no individual data points are plotted for the PDS experiments and no statistics are shown.

To address the concern of reviewer# 2 we measured CPD formation directly after UV irradiation (0 h) and 24 h after UV irradiation in WT cells. As a control siRNA against XPC was used (below, left figure). This data clearly shows that initially after UV irradiation CPDs form which are resolved 24 h after UV irradiation (see Figure below). We have included this data into the quantification of Fig. 1d of the manuscript. Taken all WT data together the trend shows that in WT cells CPDs are starting to be repaired after 24 h. Overall these quantifications show an increase in CPD formation after G4 stabilization by PDS treatment compared to untreated cells (see below, right figure). This is true directly after UV irradiation (0 h and 24 h). Control cells treated with a siRNA targeting XPC showed similar changes in CPDs.

We have changed “cellular pathway” with “NER pathway” as the reviewer suggested and rewrote the conclusion regarding the role of G4 stabilization on UV-induced DNA damage. It now states: “These results indicate that cells treated with the G4-stabilizer PDS cannot response properly to the UV-induced DNA damage (Fig. 1d).”

The authors then write that IF analysis confirmed their observation, but again refer to the same dot blot graph in Fig 1D, while they mean the experiment shown in Supplemental Figure S1e. This experiment indeed suggests that CPD levels are higher after PDS treatment. However, the problem with this experiment is also that no repair of CPDs is found in the untreated control cells. This should be a red flag that something is wrong with the experiment, as the CPD signal should diminish in 24 hours. Also, the legend states that siRNA against XPC was used, but this is not shown. Also, the DAPI staining is not shown. Also, this experiment lacks proper unirradiated control cells. The authors could consider alternative methods to measure NER, such as IF or dot blot for 64PPs or for instance

unscheduled DNA synthesis (PMID 25474029).

We thank the reviewer for the comments. Of course, DAPI stainings and western blot analyses were done for all IF analysis. Because we found that the main Figures and the Supplementary data are already extremely crowded, we did not include them in the previous manuscript. We indicated DAPI stainings as a white circled line of the nucleus, which we obtained by overlaying DAPI stainings with IF images. We have included western blot analysis as well as an example of a DAPI staining in the Data source folder as well as here (see below). We have included more information on these stainings and how the DAPI overlap was done in the Figure legends. All experiments show an unirradiated control, but as no CPDs are detected and the IF is empty we refrained from showing this data. We have included this control and show the lack in staining in the assay also here below.

To further address the reviewers' comment we have also performed IF for 6-4PPs and these new results show a repair in WT cells 4 h after treatment. In contrast, after PDS-treatment ZRF1-KO cell still exhibit intense 6-4PPs levels which are in intensity similar to the 6-4PPs levels directly after UV irradiation (time 0 h). As a control we used an siRNA against XPC that showed, as expected, no repair after UV irradiation. The data (IF and quantifications of images) are included here in the letter and are also added to the data source file.

Line 252. The authors state that enhanced G4 levels by PDS treatment lead to enhanced CPD formation and refer to figure 1D. This is, however, not at all clear from this figure. No statistics or individual data points are plotted for the PDS treated cells and it appears that CPD levels after PDS treatment are not significantly different from untreated cells. Please address this in the text and figure. Figure 3E. No individual data points are plotted for the ZRF KO cells and not statistics are shown. The authors conclude in line 254 that a moderate increase in CPD levels is displayed, but I suspect that the correct conclusion should be (based on this experiment) that there is no significant difference in CPD levels between wild type and ZRF1 KO cells. As no CPD repair is observed in wild type cells (see concern above, as in Figure 1D, which is, I guess, the same experiment), this experiment cannot be used to tell whether there is repair or not in ZRF1 KO cells (see also conclusion in lines 374-375).

As mentioned above, we are sorry about the export problems with Prism. In the new quantification of Fig. 1d a clear increase in CPD formation after PDS in comparison to WT is documented, which is supporting our conclusion.

We have also included all data points in Fig. 3e and statistics (no treatment is significant, due to the nature of this semi-quantitative experiment). We have included this information in the Figure legend.

We have also rewritten our conclusions as suggested by reviewer #2 and carefully rephrased our conclusions why CPDs remain in the ZRF1-KO cells or after PDS treatment.

Line 269. The authors conclude that ZRF1 supports DNA repair, but there is no direct evidence shown to support this conclusion. The only evidence shown that ZRF1 might affect DNA repair is the increased genomic instability observed in the form of more micronuclei and the lack of colony formation after UV. This is indeed evidence that ZRF1 might affect DNA repair, but indirect and there can be other reasons for these results in the absence of ZRF1 KO. If the authors wish to draw the conclusion that ZRF1 supports DNA repair, they should provide evidence that DNA repair (which pathway?) is affected. Otherwise, they should more carefully state their conclusions.

We fully agree that we cannot draw such a global conclusion based on our findings. According to the reviewers' comment we have carefully rephrased this statement. It now reads: "Taken together our results show that I.) ZRF1 binds to G4s after UV irradiation where it prevents genome instability and II.) cells lacking ZRF1 show increased metabolic activity after UV irradiation."

The qPCR experiment in Figure S5A suggests that DDB2 mRNA levels are higher in ZRF1 KO cells, but the immunoblot in S5B shows that protein levels (in unirradiated cells) are similar to wild type. Thus, without UV-irradiation, there is not more DDB2 protein in the cell. This is (purposely?) ignored by the authors. How can DDB2, if its protein levels are unchanged, specifically affect XPC mRNA levels in ZRF1 KO cells, as suggested by supplementary Fig S5D? In lines 300 and 309, the authors suggest that XPC is upregulated due to elevated DDB2 expression. If this is true, this must imply a mechanism by which more DDB2 mRNA (but not more protein) leads to more XPC mRNA? How would such a mechanism work and why don't the authors address this? Please address this weird issue, which suggests that more DDB2 mRNA levels regulate XPC transcription depending on ZRF1?

The reviewer is fully correct that this conclusion is not fully supported in the previous manuscript. We have re-quantified all (n=4, all data points are shown) DDB2 western blot experiments, see below for an example. Quantifications show an increase in DDB2 protein levels in ZRF1-KO cells. We conclude that increased DDB2 protein levels affect XPC mRNA abundance and probably protein levels (western blot analysis, as shows below, shows a XPC upregulation but due to the poor quality of the gels we have decided to remove the XPC protein levels from the manuscript).

In addition, to support their claim it would be more than logic to show that not only XPC mRNA levels but also XPC protein levels are affected, if the authors wish to conclude that XPC protein levels have any relevance to the observed phenotypes. This could be done in similar experiments as shown in Supplementary Figure S5E where the authors use siRNA against DDB2 and XPC and stain for DDB2 and XPC protein levels. However, they fail to show whether siRNA against DDB2 then indeed affects XPC protein levels in ZRF1 KO cells. This would have been a very obvious experiment, especially since the authors show in the rebuttal letter that they have tested protein levels of the XPC-interacting protein Centrin2. Based on this they claim that 'DDB2 misregulation causes XPC-complex level changes'. However, they also state in the rebuttal letter that 'we observed no dramatic changes of DDB2 and XPC in ZRF-KO by western blot.', indicating that the western blots (for XPC) were done. I think that these western blots should be part of the paper and the fact that DDB2 and XPC protein levels are (largely?) unchanged in ZRF1 KO cells should definitely be considered in the conclusions and discussion of the model.

We agree that the phenotype and all cellular changes are hard to explain just on mRNA expression changes. However, the quantification of western blot analyses is added to the revised manuscript and supports our hypothesis that DDB2 protein upregulation induced XPC mRNA expression and maybe protein levels (see western blot above). We have previously performed the suggested western blot analysis after siRNA against DDB2 (see example below) and this shows minor changes in XPC protein levels. As stated above, we have carefully rephrased our conclusion that ZRF1 mainly impacts DDB2 expression and protein levels.

In the rebuttal letter, the authors show that siRNA against DDB2 leads a substantial downregulation of the XPC-interactor Centrin2. This could indeed be a mechanism by which then also XPC becomes unstable. However, this mechanism would affect XPC on the protein level and not its transcription (on the mRNA level) as shown in supplemental Fig S5D, so this would constitute a different mechanism. However, Centrin2 is expressed in large excess over XPC and the majority of cellular Centrin2 is not in complex with XPC (PMID 15964821). Therefore, downregulation of Centrin2 does not necessarily have to affect XPC stability and this should be shown by staining for XPC itself. The effect of siDDB2 on Centrin2 is striking. I realize that this may be beyond the scope of this manuscript, but it would be wise to test if this downregulation is an off-target effect of the siRNA. Centrin2 is an important component of the centrosome, essential for proper mitosis, and its downregulation by siRNA against DDB2 may therefore be responsible for some of the phenotypes observed in this manuscript.

We fully agree that the downregulation of Centrin2 as a consequence of siDDB2 is very interesting and worth further analysis. We agree that DDB2 changes after Centrin2 downregulation and that the impact of Centrin2 on G4 structures, metabolic changes and the entry into senescence is in need of further experiments, but this is beyond the scope of this manuscript and will be addressed in subsequent works of our group. To test if Centrin2 downregulation is a direct consequence of siRNA treatment we tested Centrin2 levels after other siRNA controls but did not observe a similar effect. Furthermore, we would like to point out that the siRNA we used in this study was used previously also in other publications¹⁻⁴. As stated above, it could well be that DDB2 impacts Centrin2 which in turn has no strong impact on XPC protein levels as indicated by the western blot above. We have carefully rewritten all statements regarding the relevance of XPC in this ZRF1-DDB2 driven senescence activation.

Supplementary Figure S5B. The quantification suggests that DDB2 protein levels instantly increase 1.5-fold at time point 0 h, which is immediately after UV irradiation. How can this be? Such an instantaneous rise in protein levels, if indeed true, is likely not by transcriptional regulation, but by regulation of protein stability, and therefore constitute a different mechanism than the transcriptional regulation suggested by the qPCR experiment

in supplementary Figure S5A? This must be addressed and not ignored by the authors. But also in this experiment, no statistical tests are shown, so it is not possible to draw the conclusion in lines 292/293 that DDB2 protein levels are upregulated after UV irradiation. DDB2 is well known to be auto-ubiquitylated and degraded after UV irradiation and there is a large amount of literature on this subject. It may appear that ZRF1 KO prevents this DDB2 protein degradation after UV irradiation, rather than affecting its transcription, but to definitely draw this conclusion, more and better controlled experiments are needed. This is something the authors should consider.

We agree that an instant upregulation of DDB2 protein is too fast for protein expression changes. As mentioned in the Figure legends the upregulation is not significant (due to the nature of immunoblots, a semi-quantitative experiment). However, the trend for the different time points is similar, suggesting an upregulation of DDB2 protein in ZRF1-KO cells after UV irradiation. Furthermore, the reviewer is correct that higher levels of a mRNA does not automatically imply that this is due to elevated gene expression. In a previous publication we showed that G4 structures regulated by DHX36 directly impact mRNA stability⁵. As the ZRF1-KO leads to more G4s, it could be that this also impacts mRNA stability. Instant high levels of DDB2 protein could be due to different changes e.g. ubiquitylation or due to other degradation pathways. In the results section we state that DDB2 protein levels are upregulated directly after UV irradiation. We have rewritten this part in the discussion.

The authors observe for cells treated with PDS or with ZRF1 KO a decreased cellular 'growth', as measured with colony forming assay, but enhanced metabolic activity, as measured with MTT assays. It is unclear how these two opposites can be unified and explained. For instance, in line 275, the authors write that the ZRF1 KO cells stop dividing (based on the colony forming assay?) but that their metabolic rate increases after UV irradiation. The difficulty with this conclusion based on the MTT assay is that it is not clear or known if 24 hours after UV irradiation the same number of cells is present in the wells that are compared. The colony formation assay strongly suggests that 24 hours after 10 J/m² UV irradiation, many more wild type cells than ZRF1 KO cells will be present when metabolic rate is measured by MTT absorbance. If the authors wish to compare metabolic rate, I think they should be sure to use the same number of cells at the moment of measurement. This is even more important for the MTT experiments shown in Figure 4A. After UV irradiation, many cells that were treated with siRNA against DDB2 and XPC will stop dividing and/or die (as shown for XPC in the colony formation assay in supplemental Fig S1A). Thus, there will be less of these cells present in the MTT assay and consequently also less total metabolic activity measured. Therefore, this cannot be used to draw the conclusion that (upregulation of) DDB2 or XPC affects metabolic activity.

We agree that these findings are interesting and at a first sign counter intuitive. We have performed colony formation assays, growth curves and MTT analyses to better understand these findings (Fig. 1, 4; Supplementary Fig. 1, 5). We would like to point out that the MTT assays were performed 24 h after UV irradiation and colony formation 2 weeks after UV irradiation. These data, as well as the flow cytometry data (Fig. 5, Supplementary Fig. 6) show that ZRF1-KO cells are arresting in the G0/G1 cell cycle phase and stop dividing. These assays, including also all data presented in Fig. 5, show that they stop dividing because of

cellular senescence. Senescent cells are known to show elevated metabolic rates^{6,7}. We show that ZRF1-KO cells are not growing but have higher metabolic activity than WT cells that did start re-growing (as confirmed by the growth curves performed in Supplementary Fig. 6a). We speculate, based on the β -galactosidase levels in ZRF1-KO cells (24 h after UV irradiation) that the opposite results between MTT and colony formation assay are due to the senescent status. The subsequent mortality (displayed 3 days after UV irradiation in Supplementary Fig. 6a) in turn leads to the lack of growth in the colony formation assay. In order to fully rule out that cell numbers are causing the effect after siRNA treatment targeting DDB2 or XPC we carefully counted cells before each experiment.

Line 298. Where is the RNA-seq data shown that XPC is also upregulated in ZRF1 KO cells. I cannot find XPC this in the supplementary Table S1. If XPC, like DDB2, is upregulated, and interacts with ZRF1 (line 298), why did the authors only identify DDB2 in lines 283/284 after an 'in depth analysis of the NER pathway'? Why do they not mention XPC earlier?

This is a very valid question; the presented table shows only those RNA-seq changes that have a p-value in all triplicates of $p < 0.05$. However, XPC had in the triplicates one p-value higher than our stringent cut-off criteria. We have added this information to the Supplementary Table 1 as well as to the main text p10 lane 299.

Figure 1D, 3E, Supplementary Figure S1D and S4C. The dot blot and SYBR gold staining images for untreated and siRNA XPC-treated cells are the same, because this is the same experiment shown in both figures (as indicated by the full dot blots shown in supplementary Fig S7). This should be more clearly indicated in the legends and the figures in 1D, S1D should show not be presented as one single dot blot, as these are rearranged, composite images of the dot blot in Fig S7. Even better, the results of untreated cells, PDS-treated, ZRF-KO and XPC siRNA-treated cells could all be shown in the same figure.

We have included in all relevant Figure legend the information that data were obtained in one single dot blot, which had the benefit to improve direct comparison. In order to avoid misunderstanding of information presented in the figures, we did not include in Fig. 1 a single file showing all dot blots. We believe it is more intuitive for the reader to present one by one relevant information.

It appears that the colony formation experiments shown in Fig S1A and Fig S4F are also part of the same experiment. Also, this should be clearly indicated, or, better, part of the same figure for clarity.

The reviewer is correct that Supplementary Fig. 4c is the DNA loading control (SYBR gold) of the same dot blot. We have included this information in the Figure legend.

Line 311. The authors conclude that downregulation of DDB2 did not affect G4 levels significantly. However, it is unclear to which data this is compared. The authors only shown cells treated with siRNA against DDB2 in Figure 4B and C and no control siRNA (performed in the same experiment?) is shown. Similarly, for ZRF1 KO cells, if the authors wish to show that DDB2 downregulation prevents G4 accumulation, they should compare this in the same experiment to ZRF1 KO cells treated with control siRNA.

We are sorry for the missing clarity; we used in all assays a control siRNA. All the data were normalized over cells treated with a scrambled siRNA. We have avoided to show the scrambled (scr) siRNA data that we obtained in the same experiment because it displayed the same results as for untreated cells. DDB2 downregulation did not affect G4 levels significantly compared to untreated as well as to the scr siRNA-treated cells. We have added this information to the text on p10 lane 313.

Lines 312-318. The authors state that in ZRF1 KO cells, DDB2 positively contributes to G4 accumulation after UV irradiation and support this idea by showing that the DDB1-DDB2 complex binds to G4 DNA in vitro. It would have been more convincing if the authors would actually show that DDB2 binds to G4 structures in cells. In the rebuttal, they indicate that they tried this with ChIP experiments, which unfortunately did not work. However, there are multiple other ways to show enhanced binding to chromatin/DNA and for DDB2 there are multiple papers using techniques like live cell imaging, IF and/or cell fractionation that show its binding to chromatin in cells after UV irradiation. This could be considered by the authors. Please provide a reference for purification of the DDB1-DDB2 complex obtained from the Thomä lab, otherwise its purification and quality control should be described.

We have included the reference of the Thomä lab for protein purification. In order to show enhanced binding to chromatin/G4-DNA we have performed a pull down following the protocol described in⁸, avoiding the proteinase K treatment. Shortly, similar to G4 ChIP-seq analysis we incubated cell lysate with the G4-specific antibody and pulled down G4-containing DNA regions. In contrast to a ChIP analysis, we have avoided a proteinase K digestion. After stringent washing steps we analyzed by western blot the presence of DDB2. We performed two control experiments: first we have used only the anti-Flag antibody and second only the Protein G beads. Western blot analysis against DDB2 and H3 were performed (due to the high abundance of histone proteins we expected to see H3 also in the negative control). When we immunoprecipitated G4 regions with the G4 antibody, DDB2 as an interacting protein could be clearly identified in untreated and UV- irradiated conditions as shown in the immunoblot below.

Supplementary Figures S2C and S6A. It is unclear how these assays were performed and I cannot find a method where this is described. How many cells were used and how were these assays set up? This is especially important as the result with ZRF1 KO cells after UV appears to differ between the colony formation experiment shown in Figure S4F, where all cells appear to die/arrest one day after UV irradiation and the experiment shown in Figure S6A where 60% of the cells survive one day after UV irradiation. How do the authors explain this discrepancy? Please address this.

We are sorry for not including a protocol for these assays, we have now added this to the new manuscript in the materials and methods section, p15. In Supplementary Fig. 2c and 6a, cells were seeded in 24 wells (25000 per well). 24 h after seeding PDS treatment or UV irradiation was performed. 24 h post PDS treatment or UV irradiation cells were trypsinized in triplicate and counted using a Bürker chamber. Furthermore, in Supplementary Fig. 2c are only PDS-treated cells without any UV irradiation, whereas in Supplementary Fig. 6a we considered only UV-irradiated cells and the results show that ZRF1-KO cells slowly die few days after treatment. These results agree with the colony formation in Supplementary Fig. 4f where no colonies were detected 2 weeks after UV irradiation.

Lines 350-352. The authors state that flow cytometry confirmed that ZRF1 KO cells remained in the G0/G1 phase five days after UV irradiation and refer to the results shown in supplementary Figure S6B. Based on these results, it is difficult to draw this conclusion. Basically, the cell cycle distribution in ZRF1 KO cells is the same before UV (Figure 5B) as five days after UV (supplementary Figure S6B). If there is any difference, there are less cells in G0/G1 and more in S/G2. This does not necessarily mean that more cells 'remain' in G0/G1 or are 'quiescent', especially not after UV, as this type of experiment cannot distinguish between slow proliferation, transient cell cycle arrest or quiescence. Cell proliferation is apparently slower, as already shown in Supplemental Fig S2C (which the authors mention in line 199 is only a 'minor change in growth').

We agree this experiment alone cannot make this conclusion and we are sorry if we phrased this too strong in the previous version. However, flow cytometry data together with the data presented in Fig. 5 (activation of senescent genes, the lack in growth, the positive signal obtained from the lysosomal- β galactosidase) and the high metabolic rate support our conclusion that ZRF1-KO cells enter senescence after UV irradiation and that this is DDB2 dependent. Regarding Supplementary Fig. 2c the growth curves were performed without UV irradiation and cannot be compared with the cell cycle distributions performed after UV irradiation.

Minor concerns:

Supplemental Fig S1A. This is a poor example of a proper colony formation assay, as the number of colonies in unirradiated condition differs widely between cell lines. All conditions should start with the same number of colonies, but here there are much less cells treated with siRNA against XPC compared to control cells, making it difficult to draw definite conclusions (even though this outcome is as expected).

Originally, we started with the same numbers of cells, but the prolonged siRNA treatment induced an increase in mortality due to the essential role of XPC in DNA repair.

Figure 2D and supplementary Figure 3. I find the term 'micronuclei per cell' still confusing. Apparently, the authors mean micronucleus frequency or simply percentage micronuclei (but not per cell). The authors call this experiment 'IF staining', but no antibodies ('immuno') are used. This is just DAPI staining.

The reviewer is correct that DAPI is not an IF staining, we have reworded this part.

Supplementary Figure S4D. This experiment lacks unirradiated cells as control, to test if the CPD signal observed is specific. Also, the legend states that siRNA against XPC is used, but this is not shown. Also, the DAPI staining is not shown.

We did not include the DAPI staining in Supplementary Fig. 4d because of space limitations, but as mentioned before unirradiated cells show no signal. We did not include an empty staining. See the figure above for an example of CPD staining in untreated cells.

Line 60. Different kinds of DNA damage are summarized but 'UV irradiation' is not a kind of DNA damage.

We agree and have reworded this part.

Line 90. DDB1 and DDB2 together are not an E3 ubiquitin ligase. The complex also comprises RBX1 and Cul4, which are the actual E3 ligase components that are targeted by DDB2 to certain types of DNA damage. In line 177, the authors refer to this same complex with UV-DDB-CUL4A, which is inconsistent. Only, in line 388 do the authors correctly mention all the proteins of this complex, but this should of course already been introduced in the introduction. Nowadays, this complex is usually referred to as CRL4-DDB2, as many other types of CRL4 complexes exist, each with a different substrate-targeting protein like DDB2.

For instance, in transcription-coupled NER, another CRL4 complex acts, referred to as CRL4-CSA.

We have now introduced the full complex in the introduction and named it CRL4-DDB2 as suggested by the reviewer.

Line 89. XPC ubiquitination is not an essential step for NER, as in the absence of ubiquitylation, repair can still take place although less efficient.

We thank the reviewer and have rewritten this part.

Line 96. What is meant with 'the NER complex'?

We apologize for not being scientific specific enough here. We have rewritten this part.

Line 176. Which 'E3 ligase at lesion sites' is meant. Please be accurate.

We reworded this part.

Line 220/221. SSB and DSB should be SSBs and DSBs

We corrected this mistake.

Line 285 significantly should be significantly

We corrected this mistake.

Line 287. It is unclear why (a priori) the authors would assume that an upregulation of DDB2 leads to an increase in metabolic activity. What does DDB2 have to do with metabolic activity?

We agree we make this conclusion based on experiments in Fig. 4a, where we observed that upon DDB2 downregulation by siRNA in the ZRF1-KO cells metabolic levels are significantly reduced compared to ZRF1-KO cells. We believe that DDB2 is not directly driving metabolic changes, but that rather without higher levels of DDB2 gene expression changes do not occur that drive cells into senescence.

Line 306 'were' should be 'was'

We corrected this mistake.

Line 310. 'Weather' should be 'Whether'. Also, this is not a complete correct English sentence.

We corrected this mistake.

Line 311. Fig 5c,d should be Fig 4c,d

We corrected this mistake.

Line 323 NER repair is repetitive. The 'R' in NER already means repair

We corrected this mistake.

Lines 340/341 is a repetition of what is already mentioned in lines 331-333.

We corrected this mistake.

Line 331 'were' should be 'was'.

We corrected this mistake.

Line 332 states 'p16IN4a' but line 338 and the figure states 'p16INK4a'

We corrected this mistake.

Line 393. Please explain how gene expression changes that drive cells into senescence can cause lack in repair? This is not logical, as these are not DNA repair proteins.

We agree that we argued too hasty. We believe if a cell is not actively replicating it cannot repair lesions. We have reworded this part.

Line 398. The authors state that no proteins that control or protect genome stability by binding to G4s during repair processes have been identified. However, the DNA repair helicase FANCI is well known to recognize, bind and resolve G4 structures to prevent genome instability. Is this not what the authors mean? Please address this.

We apologize that we were not specific enough. Of course, there are many helicases that unwind G4s and by this prevent the loss of genome stability. However here we identified a mechanism of a protein that binds to a folded G4 and is not unwinding it. We added this information to the discussion.

Line 487. It is not explained in this method when cells were irradiated with UV. I guess this is after plating of the siRNA treated cells (which is common)

We have included this information.

I am not particularly enthusiastic about the quality of some of the western blots. I doubt how accurate a western blot such as shown in supplemental fig S5B can be quantified. The bands are not straight, of equal size, and there are bubbles in the bands. I leave it up to the editor to decide if this is acceptable.

There are grammatical errors and typos in the manuscript. Please correct these.

We corrected these mistakes.

Reviewer #3 (Remarks to the Author):

The authors have addressed my comments in full and I am supportive for publication of the revised manuscript.

We thank Reviewer #3 for his positive feedback.

References

1. Matsunuma, R. *et al.* UV Damage-Induced Phosphorylation of HBO1 Triggers CRL4. *Mol. Cell. Biol.* **36**, 394–406 (2016).
2. Nishimoto, K. *et al.* HDAC3 is required for XPC recruitment and nucleotide excision repair of DNA damage induced by UV irradiation. *Mol. Cancer Res.* **18**, 1367–1378 (2020).
3. He, Y. H. *et al.* ER α determines the chemo-resistant function of mutant p53 involving the switch between lincRNA-p21 and DDB2 expressions. *Mol. Ther. - Nucleic Acids* **25**, 536–553 (2021).
4. Niida, H. *et al.* Phosphorylated HBO1 at UV irradiated sites is essential for nucleotide excision repair. *Nat. Commun.* **8**, (2017).
5. Sauer, M. *et al.* DHX36 prevents the accumulation of translationally inactive mRNAs with G4-structures in untranslated regions. *Nat. Commun.* (2019). doi:10.1038/s41467-019-10432-5
6. Wiley, C. D. & Campisi, J. The metabolic roots of senescence: mechanisms and opportunities for intervention. *Nat. Metab.* **3**, 1290–1301 (2021).
7. Sabbatinelli, J. *et al.* Where Metabolism Meets Senescence: Focus on Endothelial Cells. *Front. Physiol.* **10**, 1–17 (2019).
8. Hänsel-Hertsch, R., Spiegel, J., Marsico, G., Tannahill, D. & Balasubramanian, S. Genome-wide mapping of endogenous G-quadruplex DNA structures by chromatin immunoprecipitation and high-throughput sequencing. *Nat. Protoc.* (2018). doi:10.1038/nprot.2017.150

REVIEWER COMMENTS

Reviewer #2 (Remarks to the Author):

The authors have (again) made considerable effort to address my additional concerns and corrected several mistakes. I appreciate this effort and I think that the manuscript has technically improved because of this. Unfortunately, there are still some issues (and several errors/mistakes) that I have detailed below, which dampen my enthusiasm.

I appreciate that the authors have performed an additional slot blot experiment as shown in the rebuttal letter. The slot blot indeed suggests complete repair in WT cells after 24 hours, but it is unclear to me why in this experiment the PDS treatment (or ZRF1 KO cells) is not included? This is, after all, what the authors are investigating? The quantification of the slot blot experiment on the right (and in Fig 1D) is then apparently not from experiments in which control and treatment conditions were performed in the same experiment? And, if both WT_1 and WT_2 of the blot on the left are quantified, these are not independent replicate measurements/experiments.

The most important issue is that, although I agree that the PDS treatment condition looks different from WT, this is not a statistically significant difference. This is admitted in the legend, but not clearly indicated in the figure or in main text. Thus, based on this the authors should conclude that there is no difference in CPD repair compared to WT, or (at the least) that this is still inconclusive. The same is true for Figure 3E. In describing this experiment on page 9, the authors do write that the CPD levels in ZRF1 KO cells are comparable to wild type, and on page 11 that PDS treatment or ZRF1 KO does not lead to an efficient NER. However, they also write in the discussion section that ZRF1 binding to G4 sites is 'essential for an efficient UV damage repair' and that 'in the absence of ZRF1 CPDs ... are not resolved within the first 24 hours'. Also, they write that there is a delay in repair in ZRF1-KO cells and that without ZRF1, UV lesions are not repaired. Also, in the abstract they write that the absence of ZRF1 triggers improper UV lesions repair. Thus, the authors contradict themselves in different sections, which is very confusing. Clear and conclusive evidence for a defect in CPD repair is still lacking, but this is ignored by the authors, while they draw major conclusions from these inconclusive experiments.

I appreciate that the authors performed the IFs with 6-4PP antibody. This data may be more convincing than the repair assays shown with CPD antibody, as in this experiment (and quantification) the control cells do show repair (albeit minor), which seems absent after PDS treatment or in ZRF1 KO cells. Why is this experiment not shown more prominently and hidden away in the data source file? And why do the authors not base their conclusions on these results instead of the CPD results?

The authors have re-quantified western blots to evaluate if DDB2 protein levels are higher in ZRF1-KO cells (shown in Fig S5B). This is very important as some of their main conclusions, such as that increased DDB2 levels induce senescence in these cells, are based on these supposed increased DDB2 protein levels. The quantified data, however, show that there is no statistically significant difference between WT and ZRF1 KO cells. This is admitted in the legend but not clearly indicated in the figure or main text. Moreover, this quantification shows a wide spread in DDB2 protein levels between experiments, making it unclear what DDB2 protein levels would really signify. Also, the mRNA levels of DDB2 appear not to be significantly different in ZRF1-KO cells compared to WT, in Fig S6A, or at least no statistical tests are performed/shown. Also, in the western blots shown in Figure S5C clearly no increased DDB2 protein expression in ZRF1 KO cells is observed. This lack of conclusive and convincing data is ignored by the authors, and it is still stated in the abstract and throughout the text that the loss of ZRF1 leads to upregulation of DDB2.

The authors similarly draw conclusions based on quantified data shown in supplementary Fig S5C. Also in this experiment, the differences shown are not statistically different, as indicated in the legend. This is also ignored in the main text.

The authors still fail to show that XPC protein levels (rather than mRNA levels) are upregulated in ZRF1 KO cells and regulated by DDB2. Even so, they do conclude that DDB2 upregulation leads to upregulation of XPC. If protein levels are not determined, it remains unclear if this can have any functional relevance.

I still think that it cannot be ruled out that the reduced metabolic activity observed after siDDB2 and siXPC in the MTT assays of Figure 4A are caused a lower number of living cells present after 24 hrs. The authors write that they carefully counted cells, but I assume this is before and not 24 hours after irradiation? Or did they first count cells, to have the same number of cells, and then directly do the MTT measurement? If there are less of cells present in the MTT assay, consequently also less total metabolic activity will be measured.

Page 6. For the IF experiment with CPD staining, the authors still wrongly refer to Fig 1D, instead of Figure S1e. Also, it is unclear where the authors observe a 1.4 fold and where a 1.6 fold increase, as levels after PDS treatment at 0 and 24 hrs appear similar and not different.

Page 8. Is it correct that co-treatment of UV and PDS resulted in 34934 ZRF1 binding sites, meaning ~10 times more sites than after UV or PDS treatment alone?

Page 9, line 3. What does the 1.2 fold refer to? To the 0 or the 24 hr time point?

Page 9. The authors write that 'the rescue conditions had similar ZRF1 protein levels', but these levels are very clearly much lower than those in WT, as is evident from the blot shown in Fig S4E. This should be accurately stated. From this western blot, I would conclude that possibly not all rescue cells stably express ZRF1. How can the authors be sure of this based on this experiment?

The legend of figure 4 refers to the wrong panels. The BG4 IP is shown in panel d, not in b. The IF is shown in panel b, not in c. The quantification is shown in panel c, not in d. Also, the new BG4 immunoprecipitation is not properly explained. The authors should indicate in the text that (according to their methods), this is an IP after crosslinking. This is important to estimate if DDB2 binding to G4s is specific or not, as anything binding to DNA (at or near G4 sites) will also be crosslinked to DNA and thus immunoprecipitated. It is therefore doubtful how specific the DDB2 binding to G4s is based on this experiment, as DDB2 is enriched in chromatin. Moreover, it is not explained what the numbers 1, 2, 3 in the figure stand for. Why is the input not shown and how do the authors know if the vague band is DDB2? Why is the vague band higher in the samples labeled with '2'? The histone 3 staining is overstained and therefore not informative at all.

Page 13, the authors claim that lack of UV damage repair is due to entry into senescence, where DNA repair is not active. Is this a known fact or speculation? What is the evidence that NER is not active in senescent cells? Please indicate this or refer to papers that show this.

Please correct the following:

Page 2, 'senescence has a two roles' should be without the 'a'

Page 2, 'the entry into senescent' should be senescence.

Page 4 . 'due to its negative impact' should be 'their negative impact'

Page 6 'the NER' should be 'the NER pathway' or 'NER'

Page 8 'is not due to SSBs or DSBs' should be 'is not due to more SSBs or DSBs'

Page 8 'and and' should be 'and'

Page 9 'enhance' should be 'enhanced'

Page 13 'this finding suggest' should be 'suggests'

Page 14 'DDB2 is upregulated immediate' should be 'immediately'

Page 14. 'treatment with the G4 ligand 20A does not induces' should be 'induce'.

Figure S5A. Expression should be Expression. The legend states that at least n=3 but many more data

points are shown.

Thank you very much for your support and sending us the reviewer comments. Please see below our responses.

The authors have (again) made considerable effort to address my additional concerns and corrected several mistakes. I appreciate this effort and I think that the manuscript has technically improved because of this. Unfortunately, there are still some issues (and several errors/mistakes) that I have detailed below, which dampen my enthusiasm.

We thank the referee for noting the considerable effort to address his/her comments which significantly improved the manuscript. We are sorry that we still could not address all concerns of the referee in the previous version. We have in response to his/her comments made further modifications to the manuscript. Please see details below:

I appreciate that the authors have performed an additional slot blot experiment as shown in the rebuttal letter. The slot blot indeed suggests complete repair in WT cells after 24 hours, but it is unclear to me why in this experiment the PDS treatment (or ZRF1 KO cells) is not included? This is, after all, what the authors are investigating? The quantification of the slot blot experiment on the right (and in Fig 1D) is then apparently not from experiments in which control and treatment conditions were performed in the same experiment? And, if both WT_1 and WT_2 of the blot on the left are quantified, these are not independent replicate measurements/experiments.

We have performed more slot blots as requested. In the previous reviewer comment he/she requested to show that DNA damage in WT cells is repaired after 24 h. In the previous version we could show this. He/she is correct that the quantification presented shows the summary of at least three independent experiments. These experiments were not performed on the same day. We do not think that another experiment would change the conclusion as the quantification of all experiments is clear and is clearly supported by the IF of CPD and 6-4PP products (see next comment).

The most important issue is that, although I agree that the PDS treatment condition looks different from WT, this is not a statistically significant difference. This is admitted in the legend, but not clearly indicated in the figure or in main text. Thus, based on this the authors should conclude that there is no difference in CPD repair compared to WT, or (at the least) that this is still inconclusive.

We disagree with the referee. In our opinion our data are clear and supporting our conclusion. We included observed changes and the corresponding statistics, both, in the figure and in the main text. All experiments were performed at least in three different, independent experiments. Further to show that PDS treatment (= G4 stabilization) leads, similar than XPC-deficient cells, to a lack in UV damage repair we performed three different experiments: First, we performed spot assays to monitor CPD formation, these dot blots are semi-quantitative and showed a small change of CPD products after PDS treatment. This change was not significant. Second, to strengthen our data we have performed IF analysis of CPD product formation. These analyses clearly showed that upon UV irradiation cells accumulated CPD products directly after the exposure (timepoint 0) and showed elevated CPD levels 24 h after irradiation. WT cells showed normal levels of CPD 24 h after exposure. These changes were significant and indicated as such in the main text and Figure. Third, we quantified 6-4PP products by IF. We obtained similar results also by this analysis: upon PDS treatment cells showed no repair and 6-4PP products accumulate. These changes were also significant. These three different, independent experiments all show that G4 stabilization by PDS leads to an accumulation of UV-induced DNA damage which is not repaired in a timely manner (e.g. CPD and 6-4PP). As suggested by the referee we included the 6-4PP data in the main figure and move the CPD quantification by IF and dot blot to the supplement. Regardless in the Figure legend statistics are listed. Significances are indicated by an asterisk symbol.

The same is true for Figure 3E. In describing this experiment on page 9, the authors do write that the CPD levels in ZRF1 KO cells are comparable to wild type, and on page 11 that PDS treatment or ZRF1 KO does not lead to an efficient NER. However, they also write in the discussion section that ZRF1 binding to G4 sites is 'essential for an efficient UV damage repair' and that 'in the absence of ZRF1 CPDs.... are not resolved within the first 24 hours'. Also, they write that there is a delay in repair in ZRF1-KO cells and that without ZRF1, UV lesions are not repaired. Also, in the abstract they write that the absence of ZRF1 triggers improper UV lesions repair. Thus, the authors contradict themselves in different sections, which is very confusing. Clear and conclusive evidence for a defect in CPD repair is still lacking, but this is ignored by the authors, while they draw major conclusions from these inconclusive experiments.

Similar to Figure 1d we performed three independent experiments to quantify changes in NER in ZRF1-KO cells. We performed quantified CPDs by spot assays and IF and determined by IF the formation/levels of 6-4PP product. All three experiments showed that in ZRF1-KO cells UV damage products are elevated and are not repaired. Observed changes in ZRF1-KO cells by both IF analyses (CPD and 6-4PP) were significant, as indicated by asterisks and in the figure legend. We agree that our spot assays is not significant but nevertheless showed elevated levels of CPDs with agree with IF stainings. We believe that spot assays are similar to western blot analyses not a reliable source for absolute quantifications – due to normalization issues. However, both IF analyses (CPD and 6-4PP) showed significant changes in ZRF1-KO compared to WT cells. These data clearly demonstrate that without ZRF1 the cells cannot efficiently repair UV lesion. Similar to Figure 1, we included the 6-4PPs staining in the main figure and moved both CPD quantifications to the supplement. Given data clearly demonstrate that ZRF1-KO cells accumulate UV induced DNA damage products.

I appreciate that the authors performed the IFs with 6-4PP antibody. This data may be more convincing than the repair assays shown with CPD antibody, as in this experiment (and quantification) the control cells do show repair (albeit minor), which seems absent after PDS treatment or in ZRF1 KO cells. Why is this experiment not shown more prominently and hidden away in the data source file? And why do the authors not base their conclusions on these results instead of the CPD results?

We are happy that the reviewer acknowledges these experiments. We have replaced the spot assay of CPDs with the IF of 6-4PP as suggested.

The authors have re-quantified western blots to evaluate if DDB2 protein levels are higher in ZRF1-KO cells (shown in Fig S5B). This is very important as some of their main conclusions, such as that increased DDB2 levels induce senescence in these cells, are based on these supposed increased DDB2 protein levels. The quantified data, however, show that there is no statistically significant difference between WT and ZRF1 KO cells. This is admitted in the legend but not clearly indicated in the figure or main text. Moreover, this quantification shows a wide spread in DDB2 protein levels between experiments, making it unclear what DDB2 protein levels would really signify. Also, the mRNA levels of DDB2 appear not to be significantly different in ZRF1-KO cells compared to WT, in Fig S6A, or at least no statistical tests are performed/shown. Also, in the western blots shown in Figure S5C clearly no increased DDB2 protein expression in ZRF1 KO cells is observed. This lack of conclusive and convincing data is ignored by the authors, and it is still stated in the abstract and throughout the text that the loss of ZRF1 leads to upregulation of DDB2.

We have performed multiple western and mRNA expression analyses, and all of these showed that DDB2 mRNA and protein levels are upregulated upon UV irradiation in ZRF1-KO cells. We apologies that our data presentation was not clear. Before UV irradiation we see elevated DDB2 protein levels (Figure S5B). This increase is clear albeit not significant (as indicated due

to the absence of asterisk in the figure). After UV irradiation, when most ZRF1 protein is recruited to the nucleus and bound to chromatin, DDB2 levels are in ZRF1-KO cells highly elevated (Figure S5C). All western blot analyses were performed at least in triplicates. As standard western blot analyses are per se semi-quantitative we performed all western blot analyses of WT and ZRF1-KO cells always on the same gel and normalized protein levels to a reference protein; high error bars are expected. We normalized in our western blot analyses (Figure S5C) all values over untreated, therefore WT and ZRF1-KO cells are both set to 1. In this analysis a clear trend was detected: in ZRF1-KO cells more DDB2 is detected after UV irradiation whereas in WT cells no changes in DDB2 levels are observed after UV irradiation. If required, we can plot the quantification of WT over ZRF1-KO cells in a chart, if this would help to understand our conclusion that both, mRNA and protein levels, are elevated. We have stated in the manuscript that western blot analyses did not yield a significant change. We have added the statistics of mRNA expression changes to Figure S5A. For DDB2 the p-value is 0.000297. We strongly believe that our data is conclusive. Without ZRF1 cells show higher levels of DDB2 mRNA expression and protein.

The authors similarly draw conclusions based on quantified data shown in supplementary Fig S5C. Also in this experiment, the differences shown are not statistically different, as indicated in the legend. This is also ignored in the main text.

See previous comment.

The authors still fail to show that XPC protein levels (rather than mRNA levels) are upregulated in ZRF1 KO cells and regulated by DDB2. Even so, they do conclude that DDB2 upregulation leads to upregulation of XPC. If protein levels are not determined, it remains unclear if this can have any functional relevance.

Based on the reviewer comments we reduced data on XPC in the past version. However, we do not want to eliminate it completely as it is a nice result and confirms previous work from other groups showing that DDB2 levels impact XPC levels and its translocation to DNA damage sites (see for example Wang QE et al 2007). We do not make major conclusion if and how XPC is involved in the senescence phenotype as this is beyond the scope of this paper in which we focus on ZRF1 and DDB2. If required, we could delete the XPC siRNA knock-down experiments shown in Figure S5E as they are nice to have but not mandatory for our manuscript.

I still think that it cannot be ruled out that the reduced metabolic activity observed after siDDB2 and siXPC in the MTT assays of Figure 4A are caused a lower number of living cells present after 24 hrs. The authors write that they carefully counted cells, but I assume this is before and not 24 hours after irradiation? Or did they first count cells, to have the same number of cells, and then directly do the MTT measurement? If there are less of cells present in the MTT assay, consequently also less total metabolic activity will be measured.

We acknowledge the concerns of the reviewer but want to argue against it. First, we counted cells before and after UV irradiation. Second, ZRF1-KO cells, as well as XPC siRNA cells do not grow, so difference in MTT rates are not due to changes in cell numbers as both cells are not growing. All tested conditions had similar cell numbers before and after UV irradiation. We have added more information to the methods and main text.

Page 6. For the IF experiment with CPD staining, the authors still wrongly refer to Fig 1D, instead of Figure S1e. Also, it is unclear where the authors observe a 1.4 fold and where a 1.6 fold increase, as levels after PDS treatment at 0 and 24 hrs appear similar and not different.

We apologize and have corrected this.

Page 8. Is it correct that co-treatment of UV and PDS resulted in 34934 ZRF1 binding sites, meaning ~10 times more sites than after UV or PDS treatment alone?

Yes, this is correct.

Page 9, line 3. What does the 1.2 fold refer to? To the 0 or the 24 hr time point?

it refers to 0 hr, after 24 hr its 1.9. fold. We have clarified this in the text.

Page 9. The authors write that ‘the rescue conditions had similar ZRF1 protein levels’, but these levels are very clearly much lower than those in WT, as is evident from the blot shown in Fig S4E. This should be accurately stated. From this western blot, I would conclude that possibly not all rescue cells stably express ZRF1. How can the authors be sure of this based on this experiment?

We agree that the rescue conditions did not fully recover WT levels of ZRF1. The main purpose of the rescue was to check if CRISPR-Cas9 knockout was specific and if there were off-site effects. We could rescue the ZRF1-KO phenotype, indicating that the ZRF1-KO was specific and there were no off-target sites. The reduced levels did not affect this. We have stated this in the text in more detail.

The legend of figure 4 refers to the wrong panels. The BG4 IP is shown in panel d, not in b. The IF is shown in panel b, not in c. The quantification is shown in panel c, not in d. Also, the new BG4 immunoprecipitation is not properly explained. The authors should indicate in the text that (according to their methods), this is an IP after crosslinking. This is important to estimate if DDB2 binding to G4s is specific or not, as anything binding to DNA (at or near G4 sites) will also be crosslinked to DNA and thus immunoprecipitated. It is therefore doubtful how specific the DDB2 binding to G4s is based on this experiment, as DDB2 is enriched in chromatin. Moreover, it is not explained what the numbers 1, 2, 3 in the figure stand for. Why is the input not shown and how do the authors know if the vague band is DDB2? Why is the vague band higher in the samples labeled with ‘2’? The histone 3 staining is overstained and therefore not informative at all.

We will correct this and explain in more detail this experiment. The experiment was performed with a G4-specific antibody (BG4). BG4 specifically targets only DNA regions that form a G4 structure. This antibody has been used multiple times in high-ranking journals (including Nature and Nature Comm.) by us and other scientists in Chromatin Immunoprecipitation (ChIP) experiments coupled to either specific qPCRs or sequencing techniques (e.g.: Hansel Hertsch et al 2016, Lago et al 2021, De Magis et al 2020). We used similar incubation times and washing conditions to ensure specific pull-down. The DNA is sheared prior to the pull-down to an average of 200 bp. Due to the specific nature of the BG4 antibody to G4 structures, the robust and proven ChIP quality of this antibody and the small fragment size precipitated (an average G4 being 30- 50 nt) we believe a specific interaction of a protein to G4 was detected. DDB2 specific binding was monitored after the pull-down by Western blot analysis using a DDB2 specific antibody. However, the referee is correct, as G4s are part of the chromatin we cannot fully exclude this. Regardless, due to band shift changes in EMSA analysis we believe that DDB2 binds to G4s. To address the referee comments, we have included more information on the protocol in the main text and further discussed these findings in more detail.

The numbers “1-2-3” refer to replicate 1, 2 and 3. We added this information to the legend. We have removed the Histone 3 data from the Figure and figure legend as it is not important for the conclusion or the quality of the data.

Page 13, the authors claim that lack of UV damage repair is due to entry into senescence,

where DNA repair is not active. Is this is known fact or speculation? What is the evidence that NER is not active in senescent cells? Please indicate this or refer to papers that show this.

It is known that cells can enter senescence due to many different aspects. One of those is the accumulation of DNA damage (e.g. see for review: Fabrizio d'Adda Di Fagagna Nature Reviews Cancer 2008 or Kumari and Jat 2021 Front. Cell Dev.).

We detected that in ZRF1-KO cells, after UV irradiation, that CPDs and 6-4PPs are accumulating and are not repaired. Further we determined that all essential NER proteins, but also other genes of other repair pathways, are expressed (based on our RNAseq data). Indicating that not per se NER is malfunctioning rather that something prevent NER from binding and repairing the lesions. We speculate that NER is not efficiently recruited to the lesions or that the cells can not complete the repair as they become stuck in G0 phase. We have discussed this further in this manuscript.

Please correct the following:

Page 2, 'senescence has a two roles' should be without the 'a'

Page 2, 'the entry into senescent' should be senescence.

Page 4 . 'due to its negative impact' should be 'their negative impact'

Page 6 'the NER' should be 'the NER pathway' or 'NER'

Page 8 'is not due to SSBs or DSBs' should be 'is not due to more SSBs or DSBs'

Page 8 'and and' should be 'and'

Page 9 'enhance' should be 'enhanced'

Page 13 'this finding suggest' should be 'suggests'

Page 14 'DDB2 is upregulated immediate' should be 'immediately'

Page 14. 'treatment with the G4 ligand 20A does not induces' should be 'induce'.

Figure S5A. Espression should be Expression. The legend states that at least n=3 but many more data points are shown.

We have corrected these mistakes and thank the reviewer for pointing them out.